# When Do Graph Neural Networks Help with Node Classification? Investigating the Impact of Homophily Principle on Node Distinguishability

**Sitao Luan**[1,2], **Chenqing Hua**[1,2], **Minkai Xu**[4], **Qincheng Lu**[1], **Jiaqi Zhu**[1],
**Xiao-Wen Chang**[1,†], **Jie Fu**[2,5,†], **Jure Leskovec**[4,†], **Doina Precup**[1,2,3,†]
{sitao.luan@mail, chenqing.hua@mail, qincheng.lu@mail, jiaqi.zhu@mail, chang@cs,
dprecup@cs}.mcgill.ca, {minkai, jure}@cs.stanford.edu, jiefu@ust.hk
[1]McGill University; [2] Mila - Quebec Artificial Intelligence Institute; [3]Google DeepMind;
[4]Stanford University; [5]HKUST; [†] Corresponding Authors

## Abstract

Homophily principle, *i.e.,* nodes with the same labels are more likely to be connected, has been believed to be the main reason for the performance superiority of Graph Neural Networks (GNNs) over Neural Networks on node classification tasks. Recent research suggests that, even in the absence of homophily, the advantage of GNNs still exists as long as nodes from the same class share similar neighborhood patterns [38]. However, this argument only considers intra-class Node Distinguishability (ND) but neglects inter-class ND, which provides incomplete understanding of homophily on GNNs. In this paper, we first demonstrate such deficiency with examples and argue that an ideal situation for ND is to have smaller intra-class ND than inter-class ND. To formulate this idea and study ND deeply, we propose Contextual Stochastic Block Model for Homophily (CSBM-H) and define two metrics, Probabilistic Bayes Error (PBE) and negative generalized Jeffreys divergence, to quantify ND. With the metrics, we visualize and analyze how graph filters, node degree distributions and class variances influence ND, and investigate the combined effect of intra- and inter-class ND. Besides, we discovered the mid-homophily pitfall, which occurs widely in graph datasets. Furthermore, we verified that, in real-work tasks, the superiority of GNNs is indeed closely related to both intra- and inter-class ND regardless of homophily levels. Grounded in this observation, we propose a new hypothesis-testing based performance metric beyond homophily, which is non-linear, feature-based and can provide statistical threshold value for GNNs' the superiority. Experiments indicate that it is significantly more effective than the existing homophily metrics on revealing the advantage and disadvantage of graph-aware modes on both synthetic and benchmark real-world datasets.

## 1 Introduction

Graph Neural Networks (GNNs) have gained popularity in recent years as a powerful tool for graph-based machine learning tasks. By combining graph signal processing and convolutional neural networks, various GNN architectures have been proposed [27, 19, 50, 36, 23], and have been shown to outperform traditional neural networks (NNs) in tasks such as node classification (**NC**), graph classification, link prediction and graph generation. The success of GNNs is believed to be rooted in the homophily principle (assumption) [42], which states that connected nodes tend to have similar attributes [18], providing extra useful information to the aggregated features over the original node features. Such relational inductive bias is thought to be a major contributor to the superiority of GNNs over NNs on various tasks [4]. On the other hand, the lack of homophily, *i.e.,* heterophily, is considered as the main cause of the inferiority of GNNs on heterophilic graphs, because nodes from different classes are connected and mixed, which can lead to indistinguishable node embeddings, making the classification task more difficult for GNNs [57, 55, 37]. Numerous models have been proposed to address the heterophily challenge lately [46, 57, 55, 37, 5, 32, 7, 54, 21, 34, 31, 51, 35].

37th Conference on Neural Information Processing Systems (NeurIPS 2023).

Recently, both empirical and theoretical studies indicate that the relationship between homophily principle and GNN performance is much more complicated than "homophily wins, heterophily loses" and the existing homophily metrics cannot accurately indicate the superiority of GNNs [38, 35, 37]. For example, the authors in [38] stated that, as long as nodes within the same class share similar neighborhood patterns, their embeddings will be similar after aggregation. They provided experimental evidence and theoretical analysis, and concluded that homophily may not be necessary for GNNs to distinguish nodes. The paper [35] studied homophily/heterophily from post-aggregation node similarity perspective and found that heterophily is not always harmful, which is consistent with [38]. Besides, the authors in [37] have proposed to use high-pass filter to address some heterophily cases, which is adopted in [7, 5] as well. They have also proposed aggregation homophily, which is a linear feature-independent performance metric and is verified to be better at revealing the performance advantages and disadvantages of GNNs than the existing homophily metrics [46, 57, 32]. Moreover, [6] has investigated heterophily from a neighbor identifiable perspective and stated that heterophily can be helpful for NC when the neighbor distributions of intra-class nodes are identifiable.

Inspite that the current literature on studying homophily principle provide the profound insights, they are still deficient: 1. [38, 6] only consider intra-class node distinguishability (**ND**), but ignore inter-class ND; 2. [35] does not show when and how high-pass filter can help with heterophily problem; 3. There is a lack of a non-linear, feature-based performance metric which can leverage richer information to provide an **accurate statistical threshold value** to indicate whether GNNs are really needed on certain task or not.

To address those issues, in this paper: 1. We show that, to comprehensively study the impact of homophily on ND, one needs to consider both intra- and inter-class ND and an ideal case is to have smaller intra-class ND than inter-class ND; 2. To formulate this idea, we propose Contextual Stochastic Block Model for Homophily (CSBM-H) as the graph generative model. It incorporates an explicit parameter to manage homophily levels, alongside class variance parameters to control intra-class ND, and node degree parameters which are important to study homophily [38, 54]; 3. To quantify ND of CSBM-H, we propose and compute two ND metrics, Probabilistic Bayes Error (**PBE**) and Negative Generalized Jeffreys Divergence ($D_{\text{NGJ}}$), for the optimal Bayes classifier of CSBM-H. Based on the metrics, we can analytically study how intra- and inter-class ND impact ND together. We visualize the relationship between PBE, $D_{\text{NGJ}}$ and homophily levels and discuss how different graph filters (full-, low- and high-pass filters), class variances and node degree distributions will influence ND in details; 4. In practice, we verify through hypothesis testing that the performance superiority of GNNs is indeed related to whether intra-class ND is smaller than inter-class ND, regardless of homophily levels. Based on this conclusion and the p-values of hypothesis testing, we propose Classifier-based Performance Metric (CPM), a new non-linear feature-based metric that can provide statistical threshold values. Experiments show that CPM is significantly more effective than the existing homophily metrics on predicting the superiority of graph-aware models over graph-agnostic.

## 2 Preliminaries

We use **bold** font for vectors (*e.g.,* $\boldsymbol{v}$) and define a connected graph $\mathcal{G} = (\mathcal{V}, \mathcal{E})$, where $\mathcal{V}$ is the set of nodes with a total of $N$ elements, $\mathcal{E}$ is the set of edges without self-loops. $A$ is the symmetric adjacency matrix with $A_{i,j} = 1$ if there is an edge between nodes $i$ and $j$, otherwise $A_{i,j} = 0$. $D$ is the diagonal degree matrix of the graph, with $D_{i,i} = d_i = \sum_j A_{i,j}$. The neighborhood set $\mathcal{N}_i$ of node $i$ is defined as $\mathcal{N}_i = \{j : e_{ij} \in \mathcal{E}\}$. A graph signal is a vector in $\mathbb{R}^N$, whose $i$-th entry is a feature of node $i$. Additionally, we use $X \in \mathbb{R}^{N \times F_h}$ to denote the feature matrix, whose columns are graph signals and $i$-th row $X_{i,:} = \boldsymbol{x}_i^\top$ is the feature vector of node $i$ (*i.e.,* the full-pass (FP) filtered signal). The label encoding matrix is $Z \in \mathbb{R}^{N \times C}$, where $C$ is the number of classes, and its $i$-th row $Z_{i,:}$ is the one-hot encoding of the label of node $i$. We denote $z_i = \arg\max_j Z_{i,j} \in \{1, 2, \ldots C\}$. The indicator function $\mathbf{1}_B$ equals 1 when event $B$ happens and 0 otherwise.

For nodes $i, j \in \mathcal{V}$, if $z_i = z_j$, then they are considered as *intra-class nodes*; if $z_i \neq z_j$, then they are considered to be *inter-class nodes*. Similarly, an edge $e_{i,j} \in \mathcal{E}$ is considered to be an *intra-class edge* if $z_i = z_j$, and an *inter-class edge* if $z_i \neq z_j$.

### 2.1 Graph-aware Models and Graph-agnostic Models

A network that includes the feature aggregation step according to graph structure is called graph-aware (**G-aware**) model, *e.g.,* GCN [27], SGC [53]; A network that does not use graph structure information is called graph-agnostic (**G-agnostic**) model, such as Multi-Layer Perceptron with 2

layers (MLP-2) and MLP-1. A G-aware model is often coupled with a G-agnostic model because when we remove the aggregation step in G-aware model, it becomes exactly the same as its coupled G-agnostic model, *e.g.,* GCN is coupled with MLP-2 and SGC-1 is coupled with MLP-1 as below,

**GCN:** $Y = \text{Softmax}(\hat{A}_{\text{sym}} \text{ReLU}(\hat{A}_{\text{sym}} X W_0) W_1)$, **MLP-2:** $Y = \text{Softmax}(\text{ReLU}(X W_0) W_1)$

**SGC-1:** $Y = \text{Softmax}(\hat{A}_{\text{sym}} X W_0)$, **MLP-1:** $Y = \text{Softmax}(X W_0)$

$$(1)$$

where $\hat{A}_{\text{sym}} = \tilde{D}^{-1/2} \tilde{A} \tilde{D}^{-1/2}$, $\tilde{A} \equiv A + I$ and $\tilde{D} \equiv D + I$; $W_0 \in \mathbb{R}^{F_0 \times F_1}$ and $W_1 \in \mathbb{R}^{F_1 \times O}$ are learnable parameter matrices. For simplicity, we denote $y_i = \arg\max_j Y_{i,j} \in \{1, 2, \ldots C\}$. The random walk renormalized matrix $\hat{A}_{\text{rw}} = \tilde{D}^{-1} \tilde{A}$ can also be applied to GCN, which is essentially a mean aggregator commonly used in some spatial-based GNNs [19]. To bridge spectral and spatial methods, we use $\hat{A}_{\text{rw}}$ in the theoretical analysis, but **self-loops are not added to the adjacency matrix** to maintain consistency with previous literature [38, 35].

To address the heterophily challenge, high-pass (HP) filter [14], such as $I - \hat{A}_{\text{rw}}$, is often used to replace low-pass (LP) filter [39] $\hat{A}_{\text{rw}}$ in GCN [5, 7, 35]. In this paper, we use $\hat{A}_{\text{rw}}$ and $I - \hat{A}_{\text{rw}}$ as the LP and HP operators, respectively. The LP and HP filtered feature matrices are represented as $H = \hat{A}_{\text{rw}} X$ and $H^{\text{HP}} = (I - \hat{A}_{\text{rw}}) X$. For simplicity, we denote $\boldsymbol{h}_i = (H_{i,:})^\top$, $\boldsymbol{h}_i^{\text{HP}} = (H_{i,:}^{\text{HP}})^\top$.

**To measure how likely the G-aware model can outperform its coupled G-agnostic model before training them** (*i.e.,* if the aggregation step according to graph structure is helpful for node classification or not), a lot of homophily metrics have been proposed and we will introduce the most commonly used ones in the following subsection.

## 2.2 Homophily Metrics

The homophily metric is a way to describe the relationship between node labels and graph structure. We introduce five commonly used homophily metrics: edge homophily [1, 57], node homophily [46], class homophily [32], generalized edge homophily [26] and aggregation homophily [35], adjusted homophily [47] and label informativeness [47] as follows:

$$H_{\text{edge}}(\mathcal{G}) = \frac{\left| \{ e_{uv} | e_{uv} \in \mathcal{E}, Z_{u,:} = Z_{v,:} \} \right|}{|\mathcal{E}|}, \; H_{\text{node}}(\mathcal{G}) = \frac{1}{|\mathcal{V}|} \sum_{v \in \mathcal{V}} H_{\text{node}}^v = \frac{1}{|\mathcal{V}|} \sum_{v \in \mathcal{V}} \frac{\left| \{ u | u \in \mathcal{N}_v, Z_{u,:} = Z_{v,:} \} \right|}{d_v},$$

$$H_{\text{class}}(\mathcal{G}) = \frac{1}{C-1} \sum_{k=1}^C \left[ h_k - \frac{\left| \{ v | Z_{v,k} = 1 \} \right|}{N} \right]_+, \; \text{where } h_k = \frac{\sum_{v \in \mathcal{V}, Z_{v,k}=1} \left| \{ u | u \in \mathcal{N}_v, Z_{u,:} = Z_{v,:} \} \right|}{\sum_{v \in \{ v | Z_{v,k}=1 \}} d_v},$$

$$H_{\text{GE}}(\mathcal{G}) = \frac{\sum_{(i,j) \in \mathcal{E}} \cos(\boldsymbol{x}_i, \boldsymbol{x}_j)}{|\mathcal{E}|}, \; H_{\text{agg}}(\mathcal{G}) = \frac{1}{|\mathcal{V}|} \times \left| \left\{ v \, | \, \text{Mean}_u \left( \{ S(\hat{A}, Z)_{v,u}^{Z_{u,:} = Z_{v,:}} \} \right) \geq \text{Mean}_u \left( \{ S(\hat{A}, Z)_{v,u}^{Z_{u,:} \neq Z_{v,:}} \} \right) \right\} \right|,$$

$$H_{\text{adj}} = \frac{H_{\text{edge}} - \sum_{c=1}^C \bar{p}_c^2}{1 - \sum_{k=1}^C \bar{p}_c^2}, \; \text{LI} = -\frac{\sum_{c_1, c_2} p_{c_1, c_2} \log \frac{p_{c_1, c_2}}{\bar{p}_{c_1} \bar{p}_{c_2}}}{\sum_c \bar{p}_c \log \bar{p}_c} = 2 - \frac{\sum_{c_1, c_2} p_{c_1, c_2} \log p_{c_1, c_2}}{\sum_c \bar{p}_c \log \bar{p}_c}$$

$$(2)$$

where $H_{\text{node}}^v$ is the local homophily value for node $v$; $[a]_+ = \max(0, a)$; $h_k$ is the class-wise homophily metric [32]; $\text{Mean}_u (\{\cdot\})$ takes the average over $u$ of a given multiset of values or variables and $S(\hat{A}, Z) = \hat{A} Z (\hat{A} Z)^\top$ is the post-aggregation node similarity matrix; $D_c = \sum_{v:z_v=c} d_v, \bar{p}_c = \frac{D_c}{2|\mathcal{E}|}, p_{c_1,c_2} = \sum_{(u,v) \in \mathcal{E}} \frac{\mathbf{1}\{z_u=c_1, z_v=c_2\}}{2|\mathcal{E}|}, c, c_1, c_2 \in \{1, \ldots, C\}$.

These metrics all fall within the range of $[0, 1]$, with a value closer to 1 indicating strong homophily and implying that G-aware models are more likely to outperform its coupled G-agnostic model, and vice versa. However, the current homophily metrics are almost all linear, feature-independent metrics which cannot provide a threshold value [35] for the superiority of G-aware model and fail to give an accurate measurement of node distinguishability (ND) . In the following section, we focus on quantifying the ND of graph models with with homophily levels and analyzing their relations.

# 3 Analysis of Homophily on Node Distinguishability (ND)

## 3.1 Motivation

**The Problem in Current Literature**    Recent research has shown that heterophily does not always negatively impact the embeddings of intra-class nodes, as long as their neighborhood patterns "corrupt in the same way" [38, 6]. For example, in Figure 1, nodes {1,2} are from class blue and both have the same heterophilic neighborhood patterns. As a result, their aggregated features will still be similar and they can be classified into the same class.

However, this is only partially true for ND if we forget to discuss inter-class ND, *e.g.,* node 3 in Figure 1 is from class green and has the same neighborhood pattern (1/3 orange, 1/3 yellow and 1/3 green) as nodes {1,2}, which means the inter-class ND will be lost after aggregation. This highlights the necessity for careful consideration of both intra- and inter-class ND when evaluating the impact of homophily on the performance of GNNs and an ideal case for NC would be node {1,2,4}, where we have smaller intra-class "distance" than inter-class "distance". We will formulate the above idea in this section and verify if it really relates to the performance of GNNs in section 4. In the following subsection, we will propose a toy graph model, on which we can study the relationship between homophily and ND directly and intuitively, and analyze intra- and inter-class ND analytically.

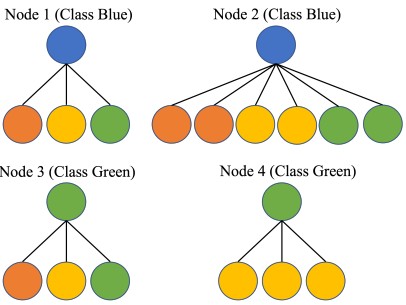

Figure 1: Example of intra- and inter-class node distinguishability.

## 3.2 CSBM-H and Optimal Bayes Classifier

In order to have more control over the assumptions on the node embeddings, we consider the Contextual Stochastic Block Model (CSBM) [11]. It is a generative model that is commonly used to create graphs and node features, and it has been widely adopted to study the behavior of GNNs [49, 3, 52]. To investigate the impact of homophily on ND, the authors in [38] simplify CSBM to the two-normal setting, where the node features $X$ are assumed to be sampled from two normal distributions and intra- and inter-class edges are generated according to two separate parameters. This simplification does not lose much information about CSBM, but 1. it does not include an explicit homophily parameter to study homophily directly and intuitively; 2. it does not include class variance parameters to study intra-class ND; 3. the authors do not rigorously quantify ND.

In this section, we introduce the Contextual Stochastic Block Model for Homophily/Heterophily (CSBM-H), which is a variation of CSBM that incorporates an explicit homophily parameter $h$ for the two-normal setting and also has class variance parameters $\sigma_0^2, \sigma_1^2$ to describe the intra-class ND. We then derive the optimal Bayes classifier (CL$_{\text{Bayes}}$) and negative generalized Jeffreys divergence for CSBM-H, based on which we can quantify and investigate ND for CSBM-H.

**CSBM-H($\boldsymbol{\mu}_0, \boldsymbol{\mu}_1, \sigma_0^2 I, \sigma_1^2 I, d_0, d_1, h$)**[1]    The generated graph consists of two disjoint sets of nodes, $i \in \mathcal{C}_0$ and $j \in \mathcal{C}_1$, corresponding to the two classes. The features of each node are generated independently, with $\boldsymbol{x}_i$ generated from $N(\boldsymbol{\mu}_0, \sigma_0^2 I)$ and $\boldsymbol{x}_j$ generated from $N(\boldsymbol{\mu}_1, \sigma_1^2 I)$, where $\boldsymbol{\mu}_0, \boldsymbol{\mu}_1 \in \mathbb{R}^{F_h}$ and $F_h$ is the dimension of the embeddings. The degree of nodes in $\mathcal{C}_0$ and $\mathcal{C}_1$ are $d_0, d_1 \in \mathbb{N}$ respectively. For $i \in \mathcal{C}_0$, its neighbors are generated by independently sampling from $h \cdot d_0$ intra-class nodes and $(1 - h) \cdot d_0$ inter-class nodes [2]. The neighbors of $j \in \mathcal{C}_1$ are generated in the same way. As a result, the FP (full-pass), LP and HP filtered features are generated as follows,

$$
\begin{aligned}
i \in \mathcal{C}_0 : \boldsymbol{x}_i \sim N(\boldsymbol{\mu}_0, \sigma_0^2 I); \ \boldsymbol{h}_i \sim N(\tilde{\boldsymbol{\mu}}_0, \tilde{\sigma}_0^2 I), \ \boldsymbol{h}_i^{\text{HP}} \sim N\left(\tilde{\boldsymbol{\mu}}_0^{\text{HP}}, (\tilde{\sigma}_0^{\text{HP}})^2 I\right), \\
j \in \mathcal{C}_1 : \boldsymbol{x}_j \sim N(\boldsymbol{\mu}_1, \sigma_1^2 I); \ \boldsymbol{h}_j \sim N(\tilde{\boldsymbol{\mu}}_1, \tilde{\sigma}_1^2 I), \ \boldsymbol{h}_j^{\text{HP}} \sim N\left(\tilde{\boldsymbol{\mu}}_1^{\text{HP}}, (\tilde{\sigma}_1^{\text{HP}})^2 I\right),
\end{aligned}
\tag{3}
$$

where $\tilde{\boldsymbol{\mu}}_0 = h(\boldsymbol{\mu}_0 - \boldsymbol{\mu}_1) + \boldsymbol{\mu}_1$, $\tilde{\boldsymbol{\mu}}_1 = h(\boldsymbol{\mu}_1 - \boldsymbol{\mu}_0) + \boldsymbol{\mu}_0$, $\tilde{\boldsymbol{\mu}}_0^{\text{HP}} = (1 - h)(\boldsymbol{\mu}_0 - \boldsymbol{\mu}_1)$, $\tilde{\boldsymbol{\mu}}_1^{\text{HP}} = (1 - h)(\boldsymbol{\mu}_1 - \boldsymbol{\mu}_0)$, $\tilde{\sigma}_0^2 = \frac{(h(\sigma_0^2 - \sigma_1^2) + \sigma_1^2)}{d_0}$, $\tilde{\sigma}_1^2 = \frac{(h(\sigma_1^2 - \sigma_0^2) + \sigma_0^2)}{d_1}$, $(\tilde{\sigma}_0^{\text{HP}})^2 = \sigma_0^2 + \frac{(h(\sigma_0^2 - \sigma_1^2) + \sigma_1^2)}{d_0}$, $(\tilde{\sigma}_1^{\text{HP}})^2 = \sigma_1^2 + \frac{(h(\sigma_1^2 - \sigma_0^2) + \sigma_0^2)}{d_1}$. If $\sigma_0^2 < \sigma_1^2$, we refer to $\mathcal{C}_0$ as the low variation class and $\mathcal{C}_1$ as the high variation class. The variance of each class can reflect the intra-class ND. We abuse the notation $\boldsymbol{x}_i \in \mathcal{C}_0$ for $i \in \mathcal{C}_0$ and $\boldsymbol{x}_j \in \mathcal{C}_1$ for $j \in \mathcal{C}_1$.

To quantify the ND of CSBM-H, we first compute the optimal Bayes classifier in the following theorem. The theorem is about the original features, but the results are applicable to the filtered features when the parameters are replaced according to equation 3.

**Theorem 1.** Suppose $\sigma_0^2 \neq \sigma_1^2$ and $\sigma_0^2, \sigma_1^2 > 0$, the prior distribution for $\boldsymbol{x}$ is $\mathbb{P}(\boldsymbol{x} \in \mathcal{C}_0) = \mathbb{P}(\boldsymbol{x} \in \mathcal{C}_1) = 1/2$, then the optimal Bayes Classifier (CL$_{\text{Bayes}}$) for CSBM-H ($\boldsymbol{\mu}_0, \boldsymbol{\mu}_1, \sigma_0^2 I, \sigma_1^2 I, d_0, d_1, h$) is

---

[1]This implies that we generate undirected graphs. See Appendix E.1 for the discussion of directed vs. undirected graphs. See E.2 for the discussion on how to extend CSBM-H to more general settings.

[2]To avoid unnecessary confusion: we relax $hd_0$ and $(1-h)d_0$ to be continuous values so that the visualization in the following sections are more readable and intuitive, especially to show the intersections of the curves.

$$\text{CL}_{\text{Bayes}}(\boldsymbol{x}) = \begin{cases} 1, & \eta(\boldsymbol{x}) \geq 0.5 \\ 0, & \eta(\boldsymbol{x}) < 0.5 \end{cases}, \ \eta(\boldsymbol{x}) = \mathbb{P}(z=1|\boldsymbol{x}) = \frac{1}{1+\exp\left(Q(\boldsymbol{x})\right)},$$

where $Q(\boldsymbol{x}) = a\boldsymbol{x}^\top \boldsymbol{x} + \boldsymbol{b}^\top \boldsymbol{x} + c$, $a = \frac{1}{2}\left(\frac{1}{\sigma_1^2} - \frac{1}{\sigma_0^2}\right)$, $\boldsymbol{b} = \frac{\boldsymbol{\mu}_0}{\sigma_0^2} - \frac{\boldsymbol{\mu}_1}{\sigma_1^2}$, $c = \frac{\boldsymbol{\mu}_1^\top \boldsymbol{\mu}_1}{2\sigma_1^2} - \frac{\boldsymbol{\mu}_0^\top \boldsymbol{\mu}_0}{2\sigma_0^2} + \ln\left(\frac{\sigma_1^{F_h}}{\sigma_0^{F_h}}\right)$

[3]. See the proof in Appendix A.

**Advantages of $\text{CL}_{\text{Bayes}}$ Over the Fixed Linear Classifier in [38]**    The decision boundary in [38] is defined as $P = \{\boldsymbol{x}|\boldsymbol{w}^\top \boldsymbol{x} - \boldsymbol{w}^\top(\boldsymbol{\mu}_0 + \boldsymbol{\mu}_1)/2\}$ where $\boldsymbol{w} = (\boldsymbol{\mu}_0 - \boldsymbol{\mu}_1)/\|\boldsymbol{\mu}_0 - \boldsymbol{\mu}_1\|_2$ is a fixed parameter. This classifier only depends on $\boldsymbol{\mu}_0, \boldsymbol{\mu}_1$ and is independent of $h$. However, as $h$ changes, the "separability" of the two normal distributions should be different. The fixed classifier cannot capture this difference, and thus is not qualified to measure ND for different $h$. Besides, we cannot investigate how variances $\sigma_0^2, \sigma_1^2$ and node degrees $d_0, d_1$ affect ND with the fixed classifier in [38].

In the following subsection, we will define two methods to quantify ND of CSBM-H, one is based on $\text{CL}_{\text{Bayes}}$, which is a precise measure but hard to be explainable; another is based on KL-divergence, which can give us more intuitive understanding of how intra- and inter-class ND will impact ND at different homophily levels. These two measurements can be used together to analyze ND.

### 3.3 Measure Node Distinguishability of CSBM-H

The Bayes error rate (BE) of the data distribution is the probability of a node being mis-classified when the true class probabilities are known given the predictors [20]. It can be used to measure the distinguishability of node embeddings and the BE for $\text{CL}_{\text{Bayes}}$ is defined as follows,

**Definition 1** (Bayes Error Rate). *The Bayes error rate [20] for* $\text{CL}_{\text{Bayes}}$ *is defined as*

$$\text{BE} = \mathbb{E}_{\boldsymbol{x}}[\mathbb{P}(z|\text{CL}_{\text{Bayes}}(\boldsymbol{x}) \neq z)] = \mathbb{E}_{\boldsymbol{x}}[1 - \mathbb{P}(\text{CL}_{\text{Bayes}}(\boldsymbol{x}) = z|\boldsymbol{x})]$$

Specifically, the BE for CSBM-H can be written as

$$\text{BE} = \mathbb{P}\left(\boldsymbol{x} \in \mathcal{C}_0\right)\left(1 - \mathbb{P}(\text{CL}_{\text{Bayes}}(\boldsymbol{x}) = 0|\boldsymbol{x} \in \mathcal{C}_0)\right) + \mathbb{P}\left(\boldsymbol{x} \in \mathcal{C}_1\right)\left(1 - \mathbb{P}(\text{CL}_{\text{Bayes}}(\boldsymbol{x}) = 1|\boldsymbol{x} \in \mathcal{C}_1)\right). \quad (4)$$

To estimate the above value, we compute Probabilistic Bayes Error (PBE) for CSBM-H as follows.

**Probabilistic Bayes Error (PBE)**    The random variable in each dimension of $\boldsymbol{x}$ is independently normally distributed. As a result, $Q(\boldsymbol{x})$ defined in Theorem 1 follows a generalized $\chi^2$ distribution [9, 10](See the calculation in Appendix C). Specifically,

$$\text{For } \boldsymbol{x}_i \in \mathcal{C}_0, \ Q(\boldsymbol{x}_i) \sim \tilde{\chi}^2(w_0, F_h, \lambda_0) + \xi; \ \boldsymbol{x}_j \in \mathcal{C}_1, \ Q(\boldsymbol{x}_j) \sim \tilde{\chi}^2(w_1, F_h, \lambda_1) + \xi$$

where $w_0 = a\sigma_0^2, w_1 = a\sigma_1^2$, the degree of freedom is $F_h$, $\lambda_0 = (\frac{\boldsymbol{\mu}_0}{\sigma_0} + \frac{\boldsymbol{b}}{2a\sigma_0})^\top(\frac{\boldsymbol{\mu}_0}{\sigma_0} + \frac{\boldsymbol{b}}{2a\sigma_0})$, $\lambda_1 = (\frac{\boldsymbol{\mu}_1}{\sigma_1} + \frac{\boldsymbol{b}}{2a\sigma_1})^\top(\frac{\boldsymbol{\mu}_1}{\sigma_1} + \frac{\boldsymbol{b}}{2a\sigma_1})$ and $\xi = c - \frac{\boldsymbol{b}^\top \boldsymbol{b}}{4a}$ . Then, by using the Cumulative Distribution Function (CDF) of $\tilde{\chi}^2$, we can calculate the predicted probabilities directly as,

$$\mathbb{P}(\text{CL}_{\text{Bayes}}(\boldsymbol{x}) = 0|\boldsymbol{x} \in \mathcal{C}_0) = 1 - \text{CDF}_{\tilde{\chi}^2(w_0, F_h, \lambda_0)}(-\xi), \ \mathbb{P}(\text{CL}_{\text{Bayes}}(\boldsymbol{x}) = 1|\boldsymbol{x} \in \mathcal{C}_1) = \text{CDF}_{\tilde{\chi}^2(w_1, F_h, \lambda_1)}(-\xi).$$

Suppose we have a balanced prior distribution $\mathbb{P}(\boldsymbol{x} \in \mathcal{C}_0) = \mathbb{P}(\boldsymbol{x} \in \mathcal{C}_1) = 1/2$. Then, PBE is,

$$\frac{\text{CDF}_{\tilde{\chi}^2(w_0, F_h, \lambda_0)}(-\xi) + \left(1 - \text{CDF}_{\tilde{\chi}^2(w_1, F_h, \lambda_1)}(-\xi)\right)}{2}$$

To investigate the impact of homophily on the ND for LP and HP filtered embeddings, we just need to replace $\left(\boldsymbol{\mu}_0, \sigma_0^2, \boldsymbol{\mu}_1, \sigma_1^2\right)$ by $\left(\tilde{\boldsymbol{\mu}}_0, \tilde{\sigma}_0^2, \tilde{\boldsymbol{\mu}}_1, \tilde{\sigma}_1^2\right)$ and $\left(\tilde{\boldsymbol{\mu}}_0^{\text{HP}}, (\tilde{\sigma}_0^{\text{HP}})^2, \tilde{\boldsymbol{\mu}}_1^{\text{HP}}, (\tilde{\sigma}_1^{\text{HP}})^2\right)$ as equation 3.

PBE can be numerically calculated and visualized to show the relationship between $h$ and ND precisely. However, we do not have an analytic expression for PBE, which makes it less explainable and intuitive. To address this issue, we define another metric for ND in the following paragraphs.

**Generalized Jeffreys Divergence**    The KL-divergence is a statistical measure of how a probability distribution $P$ is different from another distribution $Q$ [8]. It offers us a tool to define an explainable ND measure, generalized Jeffreys divergence.

**Definition 2** (Generalized Jeffreys Divergence). *For a random variable $\boldsymbol{x}$ which has either the distribution $P(\boldsymbol{x})$ or the distribution $Q(\boldsymbol{x})$, the generalized Jeffreys divergence [4] is defined as*

$$D_{GJ}(P, Q) = \mathbb{P}(\boldsymbol{x} \sim P)\mathbb{E}_{\boldsymbol{x} \sim P}\left[\ln \frac{P(\boldsymbol{x})}{Q(\boldsymbol{x})}\right] + \mathbb{P}(\boldsymbol{x} \sim Q)\mathbb{E}_{\boldsymbol{x} \sim Q}\left[\ln \frac{Q(\boldsymbol{x})}{P(\boldsymbol{x})}\right]$$

---

[3] The Bayes classifier for multiple categories ($> 2$) can be computed by stacking multiple expectation terms using similar methods as in [12, 15]. We do not discuss the more complicated settings in this paper.

[4] Jeffreys divergence [25] is originally defined as $D_{\text{KL}}(P\|Q) + D_{\text{KL}}(Q\|P)$

With $\mathbb{P}(\boldsymbol{x} \sim P) = \mathbb{P}(\boldsymbol{x} \sim Q) = 1/2$ [5], the negative generalized Jeffreys divergence for the two-normal setting in CSBM-H can be computed by (See Appendix B for the calculation)

$$D_{\text{NGJ}}(\text{CSBM-H}) = \underbrace{-d_X^2(\frac{1}{4\sigma_1^2} + \frac{1}{4\sigma_0^2})}_{\text{Negative Normalized Distance}} \underbrace{-\frac{F_h}{4}(\rho^2 + \frac{1}{\rho^2} - 2)}_{\text{Negative Variance Ratio}} \qquad (5)$$

where $d_X^2 = (\boldsymbol{\mu}_0 - \boldsymbol{\mu}_1)^\top(\boldsymbol{\mu}_0 - \boldsymbol{\mu}_1)$ is the squared Euclidean distance between centers; $\rho = \frac{\sigma_0}{\sigma_1}$ and since we assume $\sigma_0^2 < \sigma_1^2$, we have $0 < \rho < 1$. For $\boldsymbol{h}$ and $\boldsymbol{h}^{\text{HP}}$, we have $d_H^2 = (2h-1)^2 d_X^2$, $d_{\text{HP}}^2 = 4(1-h)^2 d_X^2$. The smaller $D_{\text{NGJ}}$ the CSBM-H has, the more distinguishable the embeddings are.

From equation 5, we can see that $D_{\text{NGJ}}$ implies that ND relies on two terms, Expected Negative Normalized Distance (ENND) and the Negative Variance Ratio (NVR): 1. ENND depends on how large is the inter-class ND $d_X^2$ compared with the normalization term $\frac{1}{4\sigma_1^2} + \frac{1}{4\sigma_0^2}$, which is determined by intra-class ND (variances $\sigma_0, \sigma_1$); NVR depends on how different the two intra-class NDs are, *i.e.,* when the intra-class ND of high-variation class is significantly larger than that of low-variation class ($\rho$ is close to 0), NVR is small which means the nodes are more distinguishable and vice versa.

Now, we can investigate the impact of homophily on ND through the lens of PBE and $D_{\text{NGJ}}$ [6]. Specifically, we set the standard CSBM-H as $\boldsymbol{\mu}_0 = [-1,0], \boldsymbol{\mu}_1 = [0,1], \sigma_0^2 = 1, \sigma_1^2 = 2, d_0 = 5, d_1 = 5$. And as shown in Figure 2, its PBE and $D_{\text{NGJ}}$ curves for LP filtered feature $\boldsymbol{h}$ are bell-shaped [7]. This indicates that, contrary to the prevalent belief that heterophily has the most negative impact on ND, a medium level of homophily actually has a more detrimental effect on ND than extremely low levels of homophily. We refer to this phenomenon as the **mid-homophily pitfall**.

The PBE and $D_{\text{NGJ}}$ curves for $\boldsymbol{h}^{\text{HP}}$ are monotonically increasing, which means that the high-pass filter works better in heterophily areas than in homophily areas. Moreover, it is observed that $\boldsymbol{x}, \boldsymbol{h}$, and $\boldsymbol{h}^{\text{HP}}$ will get the lowest PBE and $D_{\text{NGJ}}$ in different homophily intervals, which we refer to as the "FP regime *(black)*", "LP regime *(green)*", and "HP regime *(red)*" respectively. This indicates that LP filter works better at very low and very high homophily intervals (two ends), HP filter works better at low to medium homophily interval [8], the original (*i.e.,* full-pass or FP filtered) features works betters at medium to high homophily area.

Researchers have always been interested in exploring how node degree relates to the effect of homophily [38, 54]. In the upcoming subsection, besides node degree, we will also take a deeper look at the impact of class variances via the homophily-ND curves and the FP, LP and HP regimes.

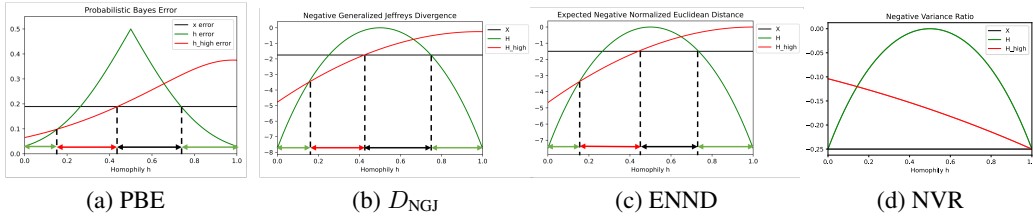

| (a) PBE | (b) $D_{\text{NGJ}}$ | (c) ENND | (d) NVR |

Figure 2: Visualization of CSBM-H $\left(\boldsymbol{\mu}_0 = [-1,0], \boldsymbol{\mu}_1 = [0,1], \sigma_0^2 = 1, \sigma_1^2 = 2, \ d_0 = 5, d_1 = 5\right)$

### 3.4 Ablation Study on CSBM-H

**Increase the Variance of High-variation Class** ($\sigma_0^2 = 1, \sigma_1^2 = 5$) From Figure 3, it is observed that as the variance in $\mathcal{C}_1$ increases and the variance between $\mathcal{C}_0$ and $\mathcal{C}_1$ becomes more imbalanced, the PBE and $D_{\text{NGJ}}$ of the three curves all go up which means the node embeddings become less distinguishable under HP, LP and FP filters. The significant shrinkage of the HP regimes and the expansion of the FP regime indicates that the original features are more robust to imbalanced variances

---

[5]We provide an open-ended discussion of imbalanced prior distributions in Appendix D.

[6]See two more metrics, negative squared Wasserstein distance and Hellinger distance, in Appendix E.3.

[7]This is consistent with the empirical results found in [35] that the relationship between the prediction accuracy of GNN and homophily value is a U-shaped curve.

[8]This verifies the conjecture made in [35] saying that high-pass filter cannot address all kinds of heterophily and only works well for certain heterophily cases.

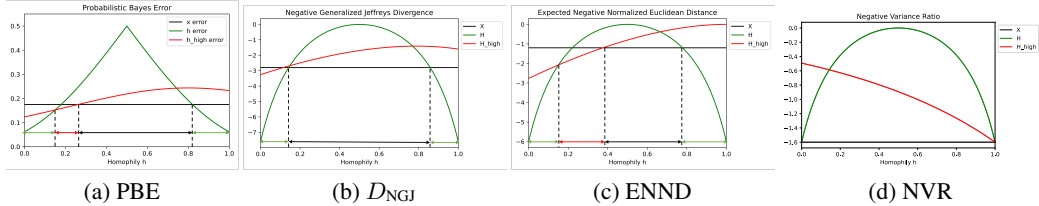

(a) PBE  (b) $D_{\mathrm{NGJ}}$  (c) ENND  (d) NVR

Figure 3: Comparison of CSBM-H with $\sigma_0^2 = 1, \sigma_1^2 = 5$.

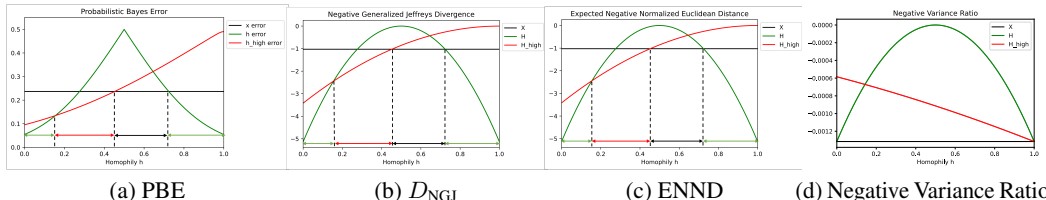

(a) PBE  (b) $D_{\mathrm{NGJ}}$  (c) ENND  (d) Negative Variance Ratio

Figure 4: Comparison of CSBM-H with $\sigma_0^2 = 1.9, \sigma_1^2 = 2$.

especially in the low homophily area. From Figure 3 (d), we can see that the main cause is that the NVR of the 3 curves all move down but the HP curve moves less in low homophily area than other 2 curves. This implies that the HP curve exhibits less sensitivity to $\rho$ within the area of low homophily.

**Increase the Variance of Low-variation Class** ($\sigma_0^2 = 1.9, \sigma_1^2 = 2$)  As shown in Figure 4, when the variance in $\mathcal{C}_0$ increases and the variance between $\mathcal{C}_0$ and $\mathcal{C}_1$ becomes more balanced, PBE and $D_{\mathrm{NGJ}}$ curves go up, which means the node embeddings become less distinguishable. The LP, HP and the FP regimes almost stays the same because the magnitude of NVR becomes too small that it almost has no effect to ND as shown in Figure 4 (d).

Interestingly, we found the change of variances cause less differences of the 3 regimes in ENND than that in NVR [9] and HP filter is less sensitive to $\rho$ changes in low homophily area than LP and FP filters. This insensitivity will have significant impact to the 3 regimes when $\rho$ is close to 0 and have trivial effect when $\rho$ is close to 1 because the magnitude of NVR is too small.

**Increase the Node Degree of High-variation Class** ($d_0 = 5, d_1 = 25$)  From Figure 5, it can be observed that as the node degree of the high-variation class increases, the PBE and $D_{\mathrm{NGJ}}$ curves of FP and HP filters almost stay the same while the curves of LP filters go down with a large margin. This leads to a substantial expansion of LP regime and shrinkage of FP and HP regime. This is mainly due to the decrease of ENND of LP filters and the decrease of its NVR in low homophily area also plays an important role.

**Increase the Node Degree of Low-variation Class** ($d_0 = 25, d_1 = 5$)  From Figure 6, we have the similar observation as when we increase the node degree of high-variation class. The difference is that the expansion of LP regime and shrinkage of FP and HP regimes are not as significant as before.

From $\tilde{\sigma}_0^2$, $\tilde{\sigma}_1^2$ we can see that increasing node degree can help LP filter reduce variances of the aggregated features so that the ENND will decrease, especially for high-variation class while HP filter is less sensitive to the change of variances and node degree.

### 3.5 More General Theoretical Analysis

Besides the toy example, in this subsection, we aim to gain a deeper understanding of how LP and HP filters affect ND in a broader context beyond the two-normal settings. To be consistent with previous literature, we follow the assumptions outlined in [38], which are: 1. The features of node $i$ are sampled from distribution $\mathcal{F}_{z_i}$, *i.e.,*, $\boldsymbol{x}_i \sim \mathcal{F}_{z_i}$, with mean $\boldsymbol{\mu}_{z_i} \in \mathbb{R}^{F_h}$; 2. Dimensions of $\boldsymbol{x}_i$ are independent to each other; 3. Each dimension in feature $\boldsymbol{x}_i$ is bounded, *i.e.,* $a \leq \boldsymbol{x}_{i,k} \leq b$; 4. For node $i$, the labels of its neighbors are independently sampled from neighborhood distribution

---

[9]To verify this, we increase $\sigma_0^2$ and $\sigma_1^2$ proportionally. From Figure 10 in Appendix F, relative sizes of the FP, LP, and HP areas remain similar.

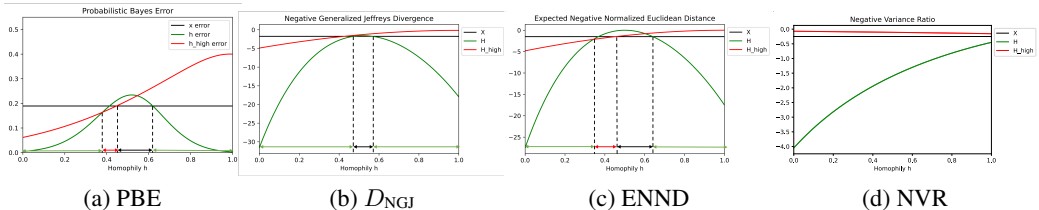

(a) PBE       (b) $D_{\text{NGJ}}$       (c) ENND       (d) NVR

Figure 5: Comparison of CSBM with different $d_0 = 5, d_1 = 25$ setups.

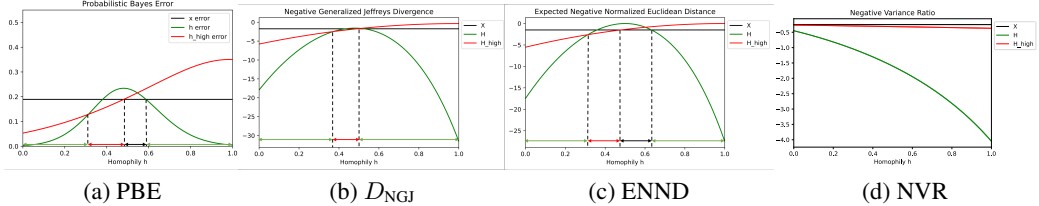

(a) PBE       (b) $D_{\text{NGJ}}$       (c) ENND       (d) NVR

Figure 6: Comparison of CSBM with different $d_0 = 25, d_1 = 5$ setups.

$\mathcal{D}_{z_i}$ and repeated for $d_i$ times. We refer to a graph that follows the above assumptions as $\mathcal{G} = \{\mathcal{V}, \mathcal{E}, \{\mathcal{F}_c, c \in \mathcal{C}\}, \{\mathcal{D}_c, c \in \mathcal{C}\}\}, \mathcal{C} = \{1, \ldots, C\}$ and $(b-a)^2$ reflects how variation the features are. The authors in [38] analyze the distance between the aggregated node embedding and its expectation, *i.e.,* $\|\boldsymbol{h}_i - \mathbb{E}(\boldsymbol{h}_i)\|_2$, which only considers the intra-class ND and has been shown to be inadequate for a comprehensive understanding of ND. Instead, we investigate **how significant the intra-class embedding distance is smaller than the inter-class embedding distance** in the following theorem, which is a better way to understand ND.

**Theorem 2.** Suppose a graph $\mathcal{G} = \{\mathcal{V}, \mathcal{E}, \{\mathcal{F}_c, c \in \mathcal{C}\}, \{\mathcal{D}_c, c \in \mathcal{C}\}\}$ meets all the above assumptions (1-4). For nodes $i, j, v \in \mathcal{V}$, suppose $z_i \neq z_j$ and $z_i = z_v$, then for constants $t_x, t_h, t_{\text{HP}}$ that satisfy $t_x \geq \sqrt{F_h} D_x(i,j)$, $t_h \geq \sqrt{F_h} D_h(i,j)$, $t_{\text{HP}} \geq \sqrt{F_h} D_{\text{HP}}(i,j)$ we have

$$\mathbb{P}\left(\|\boldsymbol{x}_i - \boldsymbol{x}_j\|_2 \geq \|\boldsymbol{x}_i - \boldsymbol{x}_v\|_2 + t_x\right) \leq 2 F_h \exp\left(-\frac{(D_x(v,j) - \frac{t_x}{\sqrt{F_h}})^2}{V_x(v,j)}\right),$$

$$\mathbb{P}(\|\boldsymbol{h}_i - \boldsymbol{h}_j\|_2 \geq \|\boldsymbol{h}_i - \boldsymbol{h}_v\|_2 + t_h) \leq 2 F_h \exp\left(-\frac{(D_h(v,j) - \frac{t_h}{\sqrt{F_h}})^2}{V_h(v,j)}\right), \quad (6)$$

$$\mathbb{P}(\|\boldsymbol{h}_i^{\text{HP}} - \boldsymbol{h}_j^{\text{HP}}\|_2 \geq \|\boldsymbol{h}_i^{\text{HP}} - \boldsymbol{h}_v^{\text{HP}}\|_2 + t_{\text{HP}}) \leq 2 F_h \exp\left(-\frac{\left(D_{\text{HP}}(v,j) - \frac{t_{\text{HP}}}{\sqrt{F_h}}\right)^2}{V_{\text{HP}}(v,j)}\right),$$

where $D_x(v,j) = \|\boldsymbol{\mu}_{z_v} - \boldsymbol{\mu}_{z_j}\|_2$, $V_x(v,j) = (b-a)^2$, $D_h(v,j) = \|\tilde{\boldsymbol{\mu}}_{z_v} - \tilde{\boldsymbol{\mu}}_{z_j}\|_2$, $V_h(v,j) = \left(\frac{1}{2d_v} + \frac{1}{2d_j}\right)(b-a)^2$, $D_{\text{HP}}(v,j) = \left\|\boldsymbol{\mu}_{z_v} - \tilde{\boldsymbol{\mu}}_{z_v} - \left(\boldsymbol{\mu}_{z_j} - \tilde{\boldsymbol{\mu}}_{z_j}\right)\right\|_2$, $V_{\text{HP}}(v,j) = \left(1 + \frac{1}{2d_v} + \frac{1}{2d_j}\right)(b-a)^2$, $\tilde{\boldsymbol{\mu}}_{z_v} = \sum_{u \in \mathcal{N}(v)} \mathbb{E}_{\substack{z_u \sim \mathcal{D}_{z_v} \\ \mathbf{x}_u \sim \mathcal{F}_{z_u}}}\left[\frac{1}{d_v}\boldsymbol{x}_u\right]$.

See the proof in Appendix G.

We can see that, the probability upper bound mainly depends on a distance term (inter-class ND) and normalized variance term (intra-class ND). The normalized variance term of HP filter is less sensitive to the changes of node degree than that of LP filter because there is an additional 1 in the constant term. Moreover, we show that the distance term of HP filter actually depends on the **relative center distance**, which is a novel discovery. As shown in Figure 7, when homophily decreases, the aggregated centers will move away from the original centers, and the relative center distance (purple) will get larger which means the embedding distance of nodes from different classes will have larger probability to be big. This explains how HP filter work for some heterophily cases. Overall, in a more general setting with weaker assumptions, we can see that ND is also described by the intra- and inter-class ND terms together rather than intra-class ND only, which is consistent with CSBM-H.

## 4 Empirical Study of Node Distinguishability

| | | Cornell | Wisconsin | Texas | Film | Chameleon | Squirrel | Cora | CiteSeer | PubMed |
|---|---|---|---|---|---|---|---|---|---|---|
| Baseline Homophily Metrics | $H_{edge}$ | 0.5669 | 0.4480 | 0.4106 | 0.3750 | 0.2795 | 0.2416 | 0.8100 | 0.7362 | 0.8024 |
| | $H_{node}$ | 0.3855 | 0.1498 | 0.0968 | 0.2210 | 0.2470 | 0.2156 | 0.8252 | 0.7175 | 0.7924 |
| | $H_{class}$ | 0.0468 | 0.0941 | 0.0013 | 0.0110 | 0.0620 | 0.0254 | 0.7657 | 0.6270 | 0.6641 |
| | $H_{agg}$ | 0.8032 | 0.7768 | 0.694 | 0.6822 | 0.61 | 0.3566 | 0.9904 | 0.9826 | 0.9432 |
| | $H_{GE}$ | 0.31 | 0.34 | 0.35 | 0.16 | 0.0152 | 0.0157 | 0.17 | 0.19 | 0.27 |
| | $H_{adj}$ | 0.1889 | 0.0826 | 0.0258 | 0.1272 | 0.0663 | 0.0196 | 0.8178 | 0.7588 | 0.7431 |
| | LI | 0.0169 | 0.1311 | 0.1923 | 0.0002 | 0.048 | 0.0015 | 0.5904 | 0.4508 | 0.4093 |
| Classifier-based Performance Metrics | $KR_{NNGP}$ | 0.00 | 0.00 | 0.00 | 0.00 | 1.00 | 1.00 | 1.00 | 1.00 | 1.00 |
| | GNB | 0.00 | 0.00 | 0.00 | 0.00 | 1.00 | 1.00 | 1.00 | 1.00 | 1.00 |
| SGC v.s. MLP-1 | p-value | 0.00 | 0.00 | 0.00 | 0.00 | 1.00 | 1.00 | 1.00 | 1.00 | 0.00 |
| | ACC SGC | 70.98 ± 8.39 | 70.38 ± 2.85 | 83.28 ± 5.43 | 25.26 ± 1.18 | 64.86 ± 1.81 | 47.62 ± 1.27 | 85.12 ± 1.64 | 79.66 ± 0.75 | 85.5 ± 0.76 |
| | ACC MLP-1 | 93.77 ± 3.34 | 93.87 ± 3.33 | 93.77 ± 3.34 | 34.53 ± 1.48 | 45.01 ± 1.58 | 29.17 ± 1.46 | 74.3 ± 1.27 | 75.51 ± 1.35 | 86.23 ± 0.54 |
| | **Diff Acc** | -22.79 | -23.49 | -10.49 | -9.27 | 19.85 | 18.45 | 10.82 | 4.15 | -0.73 |
| GCN v.s. MLP-2 | p-value | 0.00 | 0.00 | 0.00 | 0.00 | 1.00 | 1.00 | 1.00 | 1.00 | 0.00 |
| | ACC GCN | 82.46 ± 3.11 | 75.5 ± 2.92 | 83.11 ± 3.2 | 35.51 ± 0.99 | 64.18 ± 2.62 | 44.76 ± 1.39 | 87.78 ± 0.96 | 81.39 ± 1.23 | 88.9 ± 0.32 |
| | ACC MLP-2 | 91.30 ± 0.70 | 93.87 ± 3.33 | 92.26 ± 0.71 | 38.58 ± 0.25 | 46.72 ± 0.46 | 31.28 ± 0.27 | 76.44 ± 0.30 | 76.25 ± 0.28 | 86.43 ± 0.13 |
| | **Diff Acc** | -8.84 | -18.37 | -9.15 | -3.07 | 17.46 | 13.48 | 11.34 | 5.14 | 2.47 |

Table 1: P-values, homophily values and classifier-based performance metrics on 9 real-world benchmark datasets. Cells marked by grey are incorrect results for both SGC v.s. MLP-1 and GCN v.s. MLP-2 and cells marked by blue are incorrect for 1 of the 2 tests. We use 0.5 as the threshold value of the homophily metrics.

Besides theoretical analysis, in this section, we will conduct experiments to verify whether the effect of homophily on the performance of GNNs really relates to its effect on ND. If a strong relation can be verified, then it indicates that we can design new training-free ND-based performance metrics beyond homophily metrics, to evaluate the superiority and inferiority of G-aware models against its coupled G-agnostic models.

Figure 7: Demonstration of how HP filter captures the relative center distance.

### 4.1 Hypothesis Testing on Real-world Datasets

To test whether "intra-class embedding distance is smaller than the inter-class embedding distance" strongly relates to the superiority of G-aware models to their coupled G-agnostic models in practice, we conduct the following hypothesis testing [10].

**Experimental Setup** We first train two G-aware models GCN, SGC-1 and their coupled G-agnostic models MLP-2 and MLP-1 with fine-tuned hyperparameters provided by [35]. For each trained model, we calculate the pairwise Euclidean distance of the node embeddings in output layers. Next, we compute the proportion of nodes whose intra-class node distance is significantly smaller than inter-class node distance [11] *e.g.,* we obtain Prop(GCN) for GCN. We use Prop to quantify ND and we train the models multiple times for samples to conduct the following hypothesis tests:

$H_0$ : Prop(G-aware model) $\geq$ Prop(G-agnostic model); $H_1$ : Prop(G-aware model) $<$ Prop(G-agnostic model)

Specifically, we compare GCN vs. MLP-2 and SGC-1 vs. MLP-1 on 9 widely used benchmark datasets with different homophily values for 100 times. In each time, we randomly split the data into training/validation/test sets with a ratio of 60%/20%/20%. For the 100 samples, we conduct *T-test for the means of two independent samples of scores*, and obtain the corresponding p-values. The test results and model performance comparisons are shown in Table 1 (See more experimental tests on state-of-the-art model in Appendix H).

---

[10]Authors in [33] also conduct hypothesis testing to find out when to use GNNs for node classification, but they test the differences between connected nodes and unconnected nodes instead of intra- and inter-class nodes.

[11]A node is considered as "significantly smaller" when the p-value for its intra-class node distance being smaller than inter-class node distance is smaller than 0.05. In other words, this node is considered as significantly distinguishable. This second statistical test is necessary to avoid noisy nodes. In practice, we noticed that the ratio of intra-class node distance to inter-class node distance is roughly 1 for lots of nodes. This is particularly evident when the labels are sparse and when we use sampling method. It will not only cause instability of the outputs, but also result in false results sometimes. Thus, we don't want to take account these "marginal nodes" into the comparison of Prop values and we found that using another hypothesis test would be helpful.

It is observed that, in most cases (except for GCN vs. MLP-2 on *PubMed* [12]), when $H_1$ significantly holds, G-aware models will underperform the coupled G-agnostic models and vice versa. This supports our claim that the performance of G-aware models is closely related to "intra-class vs. inter-class node embedding distances", no matter the homophily levels. It reminds us that the p-value can be a better performance metric for GNNs beyond homophily. Moreover, the p-value can provide a statistical threshold, such as $p \leq 0.05$. This property is not present in existing homophily metrics.

However, it is required to train and fine-tune the models to obtain the p-values, which make it less practical because of computational costs. To overcome this issue, in the next subsection, we propose a classifier-based performance metric that can provide p-values without training.

### 4.2 Towards A Better Metric Beyond Homophily: Classifier-based Performance Metric

A qualified classifier should not require iterative training. In this paper, we choose Gaussian Naïve Bayes (GNB)[20] and Kernel Regression (KR) with Neural Network Gaussian Process (NNGP) [30, 2, 16, 41] to capture the **feature-based linear or non-linear** information.

To get the p-value efficiently, we first randomly sample 500 labeled nodes from $\mathcal{V}$ and splits them into 60%/40% as "training" and "test" data. The original features $X$ and aggregated features $H$ of the sampled training and test nodes can be calculated and are then fed into a given classifier. The predicted results and prediction accuracy of the test nodes will be computed directly with feedforward method. We repeat this process for 100 times to get 100 samples of prediction accuracy for $X$ and $H$. Then, for the given classifier, we compute the p-value of the following hypothesis testing,

$$H_0 : \text{Acc}(\text{Classifier}(H)) \geq \text{Acc}(\text{Classifier}(X)); \ H_1 : \text{Acc}(\text{Classifier}(H)) < \text{Acc}(\text{Classifier}(X))$$

The p-value can provide a statistical threshold value, such as 0.05, to indicate whether $H$ is significantly better than $X$ for node classification. As seen in Table 1, KR and GNB based metrics significantly outperform the existing homophily metrics, reducing the errors from at least 5 down to just 1 out of 18 cases. Besides, we only need a small set of the labels to calculate the p-value, which makes it better for sparse label scenario. Table 2 summarizes its advantages over the existing metrics. (See Appendix H for more details on classifier-based performance metrics, experiments on synthetic datasets, more detailed comparisons on small-scale and large-scale datasets, discrepancy between linear and non-linear models, results for symmetric renormalized affinity matrix and running time.)

## 5 Conclusions

In this paper, we provide a complete understanding of homophily by studying intra- and inter-class ND together. To theoretically investigate ND, we study the PBE and $D_{\text{NGJ}}$ of the proposed CSBM-H and analyze how graph filters, class variances and node degree distributions will influence the PBE and $D_{\text{NGJ}}$ curves and the FP, LP, HP regimes. We extend the investigation to broader settings with weaker assumptions and theoretically prove that ND is indeed affected by both intra- and inter-class ND. We also discover that the effect of HP filter depends on the relative center distance.

| Performance Metrics | Linear or Non-linear | Feature Dependency | Sparse Labels | Statistical Threshold |
|---|---|---|---|---|
| $H_{\text{node}}$ | linear | ✗ | ✗ | ✗ |
| $H_{\text{edge}}$ | linear | ✗ | ✗ | ✗ |
| $H_{\text{class}}$ | linear | ✗ | ✗ | ✗ |
| $H_{\text{agg}}$ | linear | ✗ | ✓ | ✗ |
| $H_{\text{GE}}$ | linear | ✓ | ✓ | ✗ |
| $H_{\text{adj}}$ | linear | ✗ | ✗ | ✗ |
| LI | linear | ✗ | ✗ | ✗ |
| Classifier | both | ✓ | ✓ | ✓ |

Table 2: Property comparisons of performance metrics

Empirically, through hypothesis testing, we corroborate that the performance of GNNs versus NNs is closely related to whether intra-class node embedding "distance" is smaller than inter-class node embedding "distance". We find that the p-value is a much more effective performance metric beyond homophily metrics on revealing the advantage and disadvantage of GNNs. Based on this observation, we propose classifier-based performance metric, which is a non-linear feature-based metric and can provide statistical threshold value.

---

[12]We discuss this special case in Appendix H.4, together with some similar inconsistency instances found on large-scale datasets.

## 6 Reproducibility and Blogs

- Code: https://github.com/SitaoLuan/When-Do-GNNs-Help.
- Blog in English (on Medium): https://medium.com/SitaoLuan/when-should-we-use-graph-neural-networks-for-node-classification-8ce77a772085.
- Blog in Chinese (on Zhihu): https://zhuanlan.zhihu.com/p/653631858.

## 7 Acknowledgements

This work was partially supported by the Natural Sciences and Engineering Research Council of Canada (NSERC) Grant RGPIN-2023-04125, RGPIN 2389 and Canadian Institute for Advanced Research (CIFAR) Grant CIFAR FS20-126, CIFAR 10450. Minkai Xu thanks the generous support of Sequoia Capital Stanford Graduate Fellowship.

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
