# A  Proof of Theorem 1

**Theorem 1.** Suppose $\sigma_0^2 \neq \sigma_1^2$ and $\sigma_0^2, \sigma_1^2 > 0$, the prior distribution for $x_i$ is $\mathbb{P}(x_i \in \mathcal{C}_0) = \mathbb{P}(x_i \in \mathcal{C}_1) = 1/2$, then the optimal Bayes Classifier ($\mathrm{CL}_{\mathrm{Bayes}}$) for CSBM-H $(\mu_0, \mu_1, \sigma_0^2 I, \sigma_1^2 I, d_0, d_1, h)$ is

$$\mathrm{CL}_{\mathrm{Bayes}}(x_i) = \begin{cases} 1, & \eta(x_i) \geq 0.5 \\ 0, & \eta(x_i) < 0.5 \end{cases}, \quad \eta(x_i) = \mathbb{P}(z_i = 1|x_i) = \frac{1}{1 + \exp\left(Q(x_i)\right)},$$

where $Q(x_i) = a x_i^\top x_i + b^\top x_i + c$, $a = \frac{1}{2}\left(\frac{1}{\sigma_1^2} - \frac{1}{\sigma_0^2}\right), b = \frac{\mu_0}{\sigma_0^2} - \frac{\mu_1}{\sigma_1^2}, c = \frac{\mu_1^\top \mu_1}{2\sigma_1^2} - \frac{\mu_0^\top \mu_0}{2\sigma_0^2} + \ln\left(\frac{\sigma_1^{F_h}}{\sigma_0^{F_h}}\right)$.

*Proof.* Since the prior distribution for $\mathrm{CL}_{\mathrm{Bayes}}$ is

$$\mathbb{P}(x_i \in \mathcal{C}_0) = \mathbb{P}(x_i \in \mathcal{C}_1) = \frac{1}{2}$$

$$\mathrm{CL}_{\mathrm{Bayes}}(z_i = 1|x_i) = \frac{\mathbb{P}(z_i = 1, x_i)}{\mathbb{P}(x_i)} = \frac{\mathbb{P}(z_i = 1)\mathbb{P}(x_i|z_i = 1)}{\mathbb{P}(z_i = 0)\mathbb{P}(x_i|z_i = 0) + \mathbb{P}(z_i = 1)\mathbb{P}(x_i|z_i = 1)}$$

$$= \frac{1}{1 + \frac{\mathbb{P}(z_i=0)\mathbb{P}(x_i|z_i=0)}{\mathbb{P}(z_i=1)\mathbb{P}(x_i|z_i=1)}} = \frac{1}{1 + \frac{(2\pi)^{-F_h/2}\det\left(\sigma_0^2 I\right)^{-1/2}\exp\left(-\frac{1}{2\sigma_0^2}(x_i-\mu_0)^\top(x_i-\mu_0)\right)}{(2\pi)^{-F_h/2}\det\left(\sigma_1^2 I\right)^{-1/2}\exp\left(-\frac{1}{2\sigma_1^2}(x_i-\mu_1)^\top(x_i-\mu_1)\right)}}$$

$$= \frac{1}{1 + \frac{\sigma_0^{-F_h}}{\sigma_1^{-F_h}}\exp\left(-\frac{1}{2\sigma_0^2}(x_i-\mu_0)^\top(x_i-\mu_0) + \frac{1}{2\sigma_1^2}(x_i-\mu_1)^\top(x_i-\mu_1)\right)}$$

$$= \frac{1}{1 + \frac{\sigma_0^{-F_h}}{\sigma_1^{-F_h}}\exp\left(-\frac{1}{2\sigma_0^2}(x_i^\top x_i - 2\mu_0^\top x_i + \mu_0^\top \mu_0) + \frac{1}{2\sigma_1^2}(x_i^\top x_i - 2\mu_1^\top x_i + \mu_1^\top \mu_1)\right)}$$

$$= \frac{1}{1 + \exp\left((\frac{1}{2\sigma_1^2} - \frac{1}{2\sigma_0^2})x_i^\top x_i + (\frac{\mu_0}{\sigma_0^2} - \frac{\mu_1}{\sigma_1^2})^\top x_i + \frac{\mu_1^\top \mu_1}{2\sigma_1^2} - \frac{\mu_0^\top \mu_0}{2\sigma_0^2} + \ln\left(\frac{\sigma_1^{F_h}}{\sigma_0^{F_h}}\right)\right)}$$

For the more general case where $\mathbb{P}(x_i \in \mathcal{C}_0) = \frac{n_0}{n_0+n_1}$, $\mathbb{P}(x_i \in \mathcal{C}_1) = \frac{n_1}{n_0+n_1}$, the results for $a, b$ are the same and $c = \frac{\mu_1^\top \mu_1}{2\sigma_1^2} - \frac{\mu_0^\top \mu_0}{2\sigma_0^2} + \ln\left(\frac{n_0\sigma_1^{F_h}}{n_1\sigma_0^{F_h}}\right)$. $\square$

# B  Generalized Jeffreys Divergence

Suppose we have

$$P(x) = N(\mu_0, \sigma_0^2 I), \quad Q(x) = N(\mu_1, \sigma_1^2 I)$$

Then, the KL-divergence between $P(x)$ and $Q(x)$ is

$$D_{\mathrm{KL}}(P||Q) = \int P(x) \ln\frac{P(x)}{Q(x)} dx = \mathbb{E}_{x \sim P(x)} \ln\frac{P(x)}{Q(x)}$$

$$= \mathbb{E}_{x \sim P(x)} \ln\left(\frac{\sigma_1^{F_h}}{\sigma_0^{F_h}}\exp\left(-\frac{1}{2}\sigma_0^{-2}(x-\mu_0)^\top(x-\mu_0) + \frac{1}{2}\sigma_1^{-2}(x-\mu_1)^\top(x-\mu_1)\right)\right)$$

$$= F_h \ln\frac{\sigma_1}{\sigma_0} + \mathbb{E}_{x \sim P(x)}\left(-\frac{1}{2}\sigma_0^{-2}(x-\mu_0)^\top(x-\mu_0) + \frac{1}{2}\sigma_1^{-2}(x-\mu_1)^\top(x-\mu_1)\right)$$

$$= F_h \ln\frac{\sigma_1}{\sigma_0} - \frac{F_h}{2} + \mathbb{E}_{x \sim P(x)}\left(\frac{1}{2}\sigma_1^{-2}(x-\mu_1)^\top(x-\mu_1)\right)$$

$$= F_h \ln\frac{\sigma_1}{\sigma_0} - \frac{F_h}{2} + F_h\frac{\sigma_0^2}{2\sigma_1^2} + \frac{(\mu_0 - \mu_1)^\top(\mu_0 - \mu_1)}{2\sigma_1^2}$$

$$= F_h \ln\frac{\sigma_1}{\sigma_0} - \frac{F_h}{2} + F_h\frac{\sigma_0^2}{2\sigma_1^2} + \frac{d_X^2}{2\sigma_1^2}$$

where $d_X^2$ is the squared Euclidean distance. In the same way, we have

$$D_{KL}(Q||P) = F_h \ln \frac{\sigma_0}{\sigma_1} - \frac{F_h}{2} + F_h \frac{\sigma_1^2}{2\sigma_0^2} + \frac{d_X^2}{2\sigma_0^2}$$

Suppose $\mathbb{P}(\boldsymbol{x} \sim P) = \mathbb{P}(\boldsymbol{x} \sim Q) = \frac{1}{2}$, then we have

$$D_{NGJ}(\text{CSBM-H}) = -\mathbb{P}(\boldsymbol{x} \sim P)\mathbb{E}_{\boldsymbol{x} \sim P}\left[\ln \frac{P(\boldsymbol{x})}{Q(\boldsymbol{x})}\right] - \mathbb{P}(\boldsymbol{x} \sim Q)\mathbb{E}_{\boldsymbol{x} \sim Q}\left[\ln \frac{Q(\boldsymbol{x})}{P(\boldsymbol{x})}\right]$$

$$= -\frac{F_h}{4}\left(\rho^2 + \frac{1}{\rho^2} - 2\right) - d_X^2\left(\frac{1}{4\sigma_1^2} + \frac{1}{4\sigma_0^2}\right)$$

## C   Calculation of Probabilistic Bayes Error (PBE)

### C.1   An Introduction

**Noncentral $\chi^2$ distribution**   Let $(X_1, X_2, \ldots, X_i, \ldots, X_k)$ be $k$ independent, normally distributed random variables with means $\mu_i$ and unit variances. Then the random variable

$$\sum_{i=1}^{k} X_i^2 \sim \chi'^2(k, \lambda)$$

is distributed according to the noncentral $\chi^2$ distribution. It has two parameters $(k, \lambda)$: $k$ which specifies the number of degrees of freedom (*i.e.,* the number of $X_i$), and $\lambda$ which is the sum of the squared mean of the random variables $X_i$:

$$\lambda = \sum_{i=1}^{k} \mu_i^2.$$

$\lambda$ is sometimes called the noncentrality parameter.

**Generalized $\chi^2$ distribution**   The generalized $\chi^2$ variable can be written as a linear sum of independent noncentral $\chi^2$ variables and a normal variable:

$$\xi = \sum_i w_i Y_i + X, \quad Y_i \sim \chi'^2(k_i, \lambda_i), \quad X \sim N(m, s^2)$$

Here the parameters are the weights $w_i$, the degrees of freedom $k_i$ and non-centralities $\lambda_i$ of the constituent $\chi^2$, and the normal parameters $m$ and $s$.

### C.2   Quadratic Function

For $i \in \mathcal{C}_0$, we rewrite $\boldsymbol{x}_i = \sigma_0 \boldsymbol{y}_i + \boldsymbol{\mu}_0$ where $\boldsymbol{y}_i$ is the standard normal variable. The quadratic function of $Q(\boldsymbol{x}_i)$ satisfies

$$\begin{aligned}
Q(\boldsymbol{x}_i) &= a\boldsymbol{x}_i^\top \boldsymbol{x}_i + \boldsymbol{b}^\top \boldsymbol{x}_i + c \\
&= a\left(\boldsymbol{x}_i + \frac{\boldsymbol{b}}{2a}\right)^\top \left(\boldsymbol{x}_i + \frac{\boldsymbol{b}}{2a}\right) + c - \frac{\boldsymbol{b}^\top \boldsymbol{b}}{4a} \\
&= a\left(\sigma_0 \boldsymbol{y}_i + \boldsymbol{\mu}_0 + \frac{\boldsymbol{b}}{2a}\right)^\top \left(\sigma_0 \boldsymbol{y}_i + \boldsymbol{\mu}_0 + \frac{\boldsymbol{b}}{2a}\right) + c - \frac{\boldsymbol{b}^\top \boldsymbol{b}}{4a} \\
&= a\sigma_0^2\left(\boldsymbol{y}_i + \frac{\boldsymbol{\mu}_0}{\sigma_0} + \frac{\boldsymbol{b}}{2a\sigma_0}\right)^\top \left(\boldsymbol{y}_i + \frac{\boldsymbol{\mu}_0}{\sigma_0} + \frac{\boldsymbol{b}}{2a\sigma_0}\right) + c - \frac{\boldsymbol{b}^\top \boldsymbol{b}}{4a} \\
&= w_0 y_i' + c - \frac{\boldsymbol{b}^\top \boldsymbol{b}}{4a} \sim \tilde{\chi}^2(w_0, F_h, \lambda_0) + c - \frac{\boldsymbol{b}^\top \boldsymbol{b}}{4a}
\end{aligned}$$

where $y_i' = (\boldsymbol{y}_i + \frac{\boldsymbol{\mu}_0}{\sigma_0} + \frac{\boldsymbol{b}}{2a\sigma_0})^\top(\boldsymbol{y}_i + \frac{\boldsymbol{\mu}_0}{\sigma_0} + \frac{\boldsymbol{b}}{2a\sigma_0}) \sim \chi'^2(F_h, \lambda_0)$ is distributed as non-central $\chi^2$ distribution, the degree of freedom is $F_h$, $\lambda_0 = (\frac{\boldsymbol{\mu}_0}{\sigma_0} + \frac{\boldsymbol{b}}{2a\sigma_0})^\top(\frac{\boldsymbol{\mu}_0}{\sigma_0} + \frac{\boldsymbol{b}}{2a\sigma_0})$, the weight $w_0 = a\sigma_0^2$. Then, the cumulative distribution function (CDF) of $Q(\boldsymbol{x}_i)$ can be calculated as follows,

$$\text{CDF}(x) = \mathbb{P}(Q(\boldsymbol{x}_i) \leq x) = \mathbb{P}(\tilde{\chi}^2(w_0, F_h, \lambda_0) \leq x - c + \frac{\boldsymbol{b}^\top \boldsymbol{b}}{4a}) = \text{CDF}_{\tilde{\chi}^2(w_0, F_h, \lambda_0)}(x - \xi)$$

where $\xi = c - \frac{\boldsymbol{b}^\top \boldsymbol{b}}{4a}$. For $j \in \mathcal{C}_1$ and $\boldsymbol{h}_i, \boldsymbol{h}_j$, we can apply the same computation. And since

$$\mathbb{P}(\text{CL}_{\text{Bayes}}(\boldsymbol{x}) = 0 | \boldsymbol{x} \in \mathcal{C}_0) = \mathbb{P}(Q(\boldsymbol{x}) > 0 | \boldsymbol{x} \in \mathcal{C}_0) = 1 - \text{CDF}_{\tilde{\chi}^2(w_0, F_h, \lambda_0)}(-\xi),$$
$$\mathbb{P}(\text{CL}_{\text{Bayes}}(\boldsymbol{x}) = 1 | \boldsymbol{x} \in \mathcal{C}_1) = \mathbb{P}(Q(\boldsymbol{x}) \leq 0 | \boldsymbol{x} \in \mathcal{C}_1) = \text{CDF}_{\tilde{\chi}^2(w_1, F_h, \lambda_1)}(-\xi).$$

where $w_1 = a\sigma_1^2$, $\lambda_1 = (\frac{\boldsymbol{\mu}_1}{\sigma_1} + \frac{\boldsymbol{b}}{2a\sigma_1})^\top(\frac{\boldsymbol{\mu}_1}{\sigma_1} + \frac{\boldsymbol{b}}{2a\sigma_1})$. Then from equation 4 and with $\mathbb{P}(\boldsymbol{x} \sim P) = \mathbb{P}(\boldsymbol{x} \sim Q) = 1/2$, the PBE for the two normal settings can be calculated as,

$$\frac{\text{CDF}_{\tilde{\chi}^2(w_0, F_h, \lambda_0)}(-\xi) + \left(1 - \text{CDF}_{\tilde{\chi}^2(w_1, F_h, \lambda_1)}(-\xi)\right)}{2}$$

For a special case where $\sigma_0^2 = \sigma_1^2 \neq 0$, we have $a = 0$ and $Q(\boldsymbol{x}_i) = \boldsymbol{b}^\top \boldsymbol{x}_i + c$ follows a normal distribution:

$$Q(\boldsymbol{x}_i) \sim N(\boldsymbol{b}^\top \boldsymbol{\mu}_0 + c, \sigma_0^2 \boldsymbol{b}^\top \boldsymbol{b})$$

Then

$$\text{CDF}(x) = \mathbb{P}(Q(\boldsymbol{x}_i) \leq x) = \mathbb{P}(\boldsymbol{b}^\top \boldsymbol{\mu}_0 + c + \sqrt{\sigma_0^2 \boldsymbol{b}^\top \boldsymbol{b}} x' \leq x)$$
$$= \mathbb{P}(x' \leq \frac{x - (\boldsymbol{b}^\top \boldsymbol{\mu}_0 + c)}{\sqrt{\sigma_0^2 \boldsymbol{b}^\top \boldsymbol{b}}})$$

where $x' \sim N(0,1)$ and the PBE becomes

$$\frac{\text{CDF}_{N(0,1)}\left(\frac{-(\boldsymbol{b}^\top \boldsymbol{\mu}_0 + c)}{\sqrt{\sigma_0^2 \boldsymbol{b}^\top \boldsymbol{b}}}\right) + \left(1 - \text{CDF}_{N(0,1)}\left(\frac{-(\boldsymbol{b}^\top \boldsymbol{\mu}_1 + c)}{\sqrt{\sigma_1^2 \boldsymbol{b}^\top \boldsymbol{b}}}\right)\right)}{2}$$

## D   A Discussion of (Imbalanced) Prior Distribution

In this section, we provide an open-ended discussion on how the prior distribution (*i.e.,* imbalanced datasets) will influence the ND of CSBM-H, which can possibly lead to some interesting future works.

Let $\mathbb{P}(\boldsymbol{x} \sim \mathcal{C}_0) = \frac{n_0}{n_1 + n_0} = p_0$, $\mathbb{P}(\boldsymbol{x} \sim \mathcal{C}_1) = \frac{n_1}{n_1 + n_0} = p_1, p_0 + p_1 = 1$ and $\rho = \frac{\sigma_0}{\sigma_1}$, which is the ratio of standard deviation and $0 \leq \rho \leq 1$, then $D_{\text{NGJ}}$ is

$$D_{\text{NGJ}}(\text{CSBM-H}) = -p_0 D_{\text{KL}}(P||Q) - p_1 D_{\text{KL}}(Q||P)$$
$$= F_h \ln \rho(p_1 - p_0) + \frac{F_h}{2}(p_0 \rho^2 + \frac{p_1}{\rho^2} - 1) + d_X^2(\frac{p_0}{2\sigma_1^2} + \frac{p_1}{2\sigma_0^2})$$

where $d_X^2 = (\boldsymbol{\mu}_0 - \boldsymbol{\mu}_1)^\top (\boldsymbol{\mu}_0 - \boldsymbol{\mu}_1)$, which is the square Euclidean distance between the means of the two distributions.

And the PBE of CSBM-H becomes

$$\frac{n_0 \text{CDF}_{\tilde{\chi}^2(w_0, F_h, \lambda_0)}(-\xi) + n_1 \left(1 - \text{CDF}_{\tilde{\chi}^2(w_1, F_h, \lambda_1)}(-\xi)\right)}{n_0 + n_1}$$

From Figure 8 (b) we can find that, as the size of the low-variation class increases, the LP regime expands and HP regime shrinks at the low homophily area in terms of $D_{\text{NGJ}}$. This is because in ENND, the normalization term $\frac{p_0}{2\sigma_1^2}$ gets higher weight, making the curve for LP filters move down and HP filter move up, which leads to the expansion of LP regime.

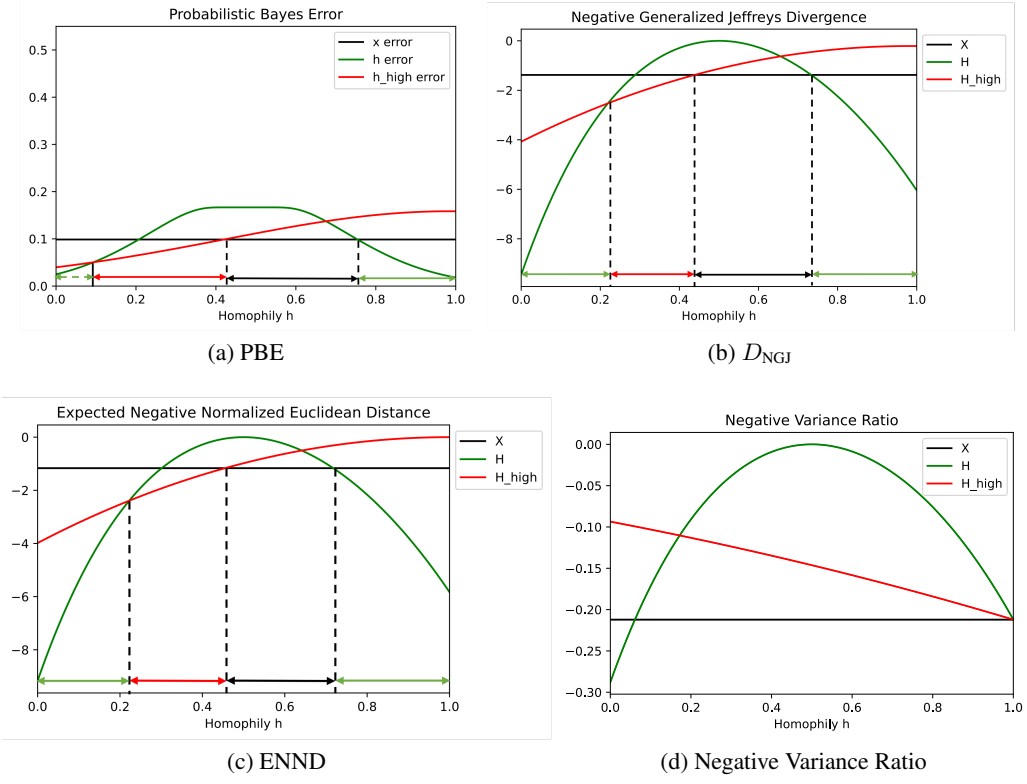

Figure 8: Comparison of CSBM-H with $n_0 = 500, n_1 = 100$.

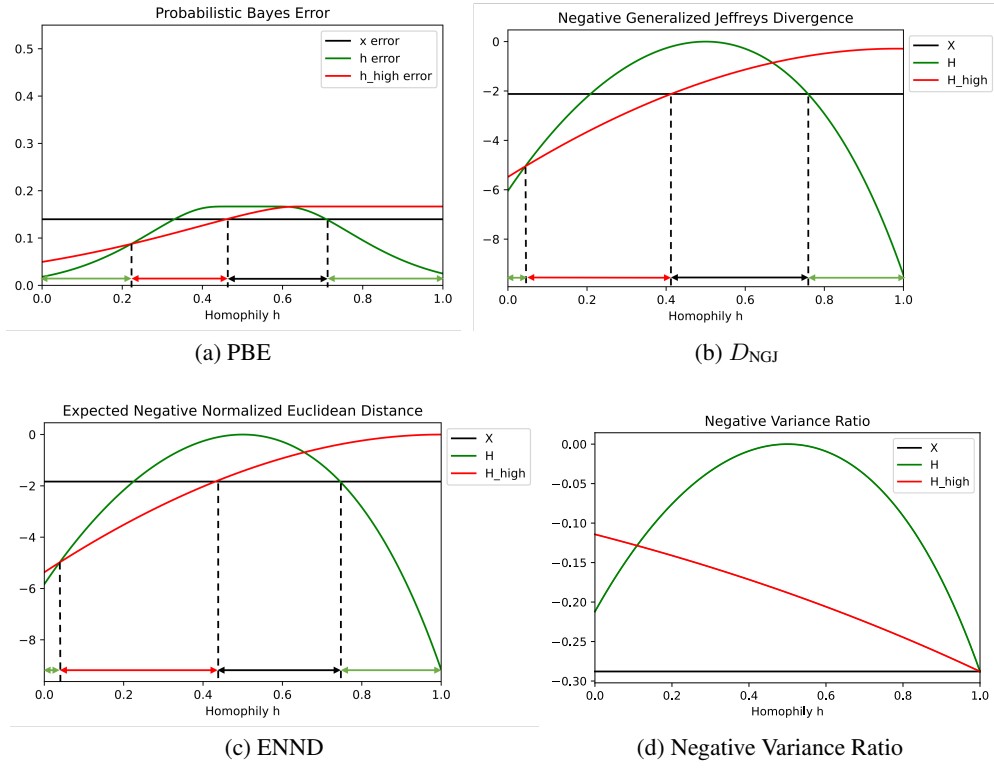

Figure 9: Comparison of CSBM-H with $n_0 = 100, n_1 = 500$.

However, we can also observe that the changes of PBE and $D_{\mathrm{NGJ}}$ curves show inconsistent results. As the size of the low-variation class increases, the LP regime shrinks and HP regime expands in PBE, while the LP regime expands and HP regime shrinks in $D_{\mathrm{NGJ}}$. In Figure 9, we observe the similar inconsistency between PBE and $D_{\mathrm{NGJ}}$ curves. This discrepancy reminds us that the performance of LP and HP filters on imbalanced datasets might be under-explored. We do not have a conclusion for this challenge in this paper and we encourage more researchers to study the connection among the prior distribution, the performance of LP and HP filters and ND.

## E   More About CSBM-H

### E.1   Directed or Undirected Graphs?

**Question**   Why not generated an undirected graph.

**Answer**   If we impose undirected assumption in CSBM-H, we have to not only discuss the node degree from intra-class edges, but also discuss degree from inter-class edges and control their relations with the corresponding homophily level. This will inevitably add more parameters to CSBM-H and make the model much more complicated. However, we find that this complication does not bring us extra benefit for understanding the effect of homophily, which deviate the main goal of our paper. And we guess this might be one of the reasons that the existing work mainly keep the discussion within the directed setting [38].

Actually, when CSMB-H was firstly designed, we would like to only have one "free parameter" $h$ in it to make it simple. Because in this way, we are able to show the whole picture of the effect of homophily from 0 to 1, like the figures in Section 3.4.

### E.2   Extend CSBM-H to More General Settings

The CSBM-H can be extended to more general settings: 1. The two-normal setting can be expanded to multi-class classification problems with different sets of $(\boldsymbol{\mu}, \sigma^2 \boldsymbol{I}, d)$ parameters for each class; 2. The degrees of nodes can be generalized to different degree distributions; 3. The scalar homophily parameter $h$ can be generalized to matrix $H \in \mathbb{R}^{C \times C}$, where $H_{c_1, c_2}$ represents the probability of nodes in class $\mathcal{C}_1$ connecting to nodes in class $\mathcal{C}_2$, which is the compatibility matrix used in [56]. Furthermore, we can also define the local homophily value $H_{v,c}$ for each node, where $H_{v,c}$ indicates the proportion of neighbors of $v$ that connect to nodes from class $\mathcal{C}_c$. To demonstrate and visualize the effect of homophily intuitively and easily, we use the CSBM-H settings in this paper as stated above. Although the settings are simple, insightful results can still be obtained. The more complicated variants will be left for future work.

### E.3   Two More Metrics of ND: Negative Squared Wasserstein Distance and Hellinger Distance

**Original Definition**   In general, the Wasserstein distance between two Gaussians is $d = W_2\left(N\left(\boldsymbol{\mu}_0, \Sigma_0\right); N\left(\boldsymbol{\mu_1}, \Sigma_1\right)\right)$ and we have [17, 28, 44, 13]

$$d^2 = \|\boldsymbol{\mu}_0 - \boldsymbol{\mu}_1\|_2^2 + \mathrm{Tr}\left(\Sigma_0 + \Sigma_1 - 2\left(\Sigma_0^{1/2}\Sigma_1\Sigma_0^{1/2}\right)^{1/2}\right)$$

The squared Hellinger distance between two Gaussians is [45]

$$H^2(N\left(\boldsymbol{\mu}_0, \Sigma_0\right); N\left(\boldsymbol{\mu_1}, \Sigma_1\right)) = 1 - \frac{\det(\Sigma_0)^{1/4}\det(\Sigma_1)^{1/4}}{\det\left(\frac{\Sigma_0 + \Sigma_1}{2}\right)^{1/2}} \exp\left\{-\frac{1}{8}\left(\boldsymbol{\mu}_0 - \boldsymbol{\mu}_1\right)^{\top}\left(\frac{\Sigma_0 + \Sigma_1}{2}\right)^{-1}\left(\boldsymbol{\mu}_0 - \boldsymbol{\mu}_1\right)\right\}$$

Wasserstein distance is a distance function defined between probability distributions and Hellinger distance is a type of $f$-divergence which is used to quantify the similarity between two probability distributions [48, 24]. These two metrics can be used to study the ND of CSBM-H and we will introduce them in the following subsection.

**Calculation for CSBM-H**   The negative squared Wasserstein distance (NSWD) for CSBM-H is

$$\mathrm{NSWD} = -\|\boldsymbol{\mu}_0 - \boldsymbol{\mu}_1\|_2^2 - F_h(\sigma_0 - \sigma_1)^2$$

The negative squared Hellinger distance (NSHD) for CSBM-H is

$$
\begin{aligned}
\text{NSHD} &= -1 + \frac{\det\left(\sigma_0^2 I\right)^{1/4} \det\left(\sigma_1^2 I\right)^{1/4}}{\det\left(\frac{\sigma_0^2 I + \sigma_1^2 I}{2}\right)^{1/2}} \exp\left\{-\frac{1}{8}\left(\boldsymbol{\mu}_0 - \boldsymbol{\mu}_1\right)^\top \left(\frac{\sigma_0^2 + \sigma_1^2}{2}\right)^{-1} \left(\boldsymbol{\mu}_0 - \boldsymbol{\mu}_1\right)\right\} \\
&= -1 + \frac{\sigma_0^{F_h/2} \sigma_1^{F_h/2}}{\left(\frac{\sigma_0^2 + \sigma_1^2}{2}\right)^{F_h/2}} \exp\left\{-\frac{1}{8}\left(\boldsymbol{\mu}_0 - \boldsymbol{\mu}_1\right)^\top \left(\frac{\sigma_0^2 + \sigma_1^2}{2}\right)^{-1} \left(\boldsymbol{\mu}_0 - \boldsymbol{\mu}_1\right)\right\} \\
&= -1 + \left(\frac{2}{\rho^2 + 1/\rho^2}\right)^{F_h/2} \exp\left\{-\frac{d_X^2}{4\left(\sigma_0^2 + \sigma_1^2\right)}\right\}
\end{aligned}
$$

Although defining the distance in different ways, both NSWD and NSHD indicate that the ND of CSBM-H depends on both intra- and inter-class ND, which is consistent with our conclusions in main paper. Besides, NSWD and NSHD provide analytic expressions for ND, which can be good tools for future research.

# F  More Figures of CSBM-H

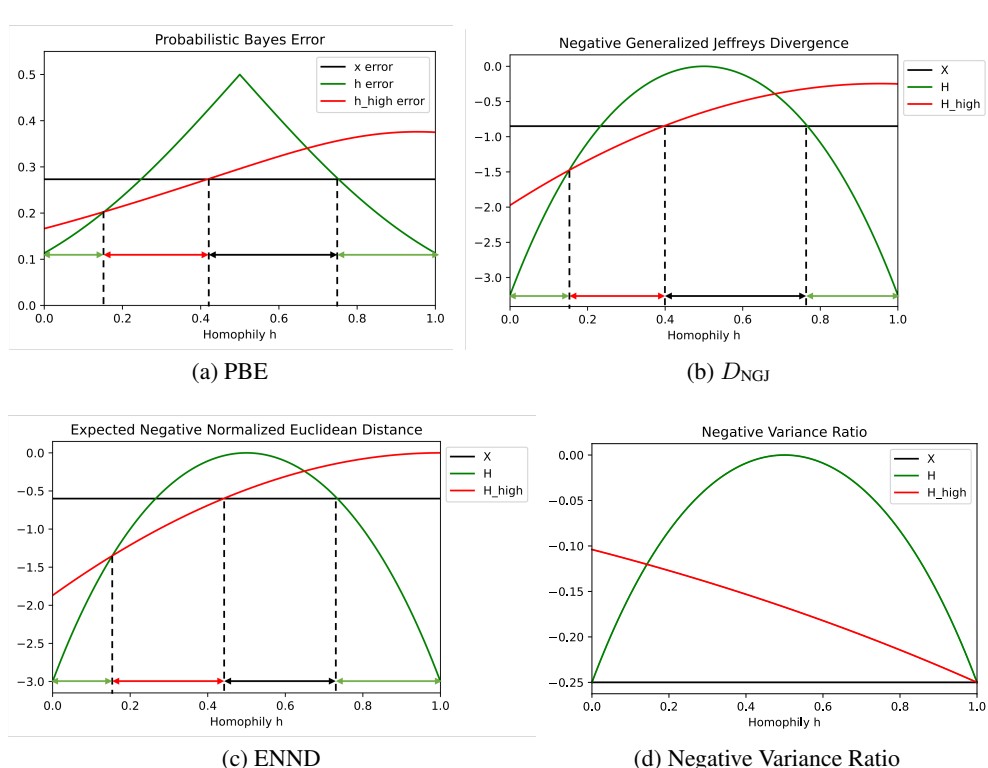

(a) PBE

(b) $D_{\text{NGJ}}$

(c) ENND

(d) Negative Variance Ratio

Figure 10: Comparison of CSBM-H with $\sigma_0^2 = 2.5, \sigma_1^2 = 5$.

# G  Proof of Theorem 2

To prove theorem 2, we need the following two lemmas.

**Lemma 1.** Let $\boldsymbol{x}_i = X_{i,:}$ and suppose each dimension of $\boldsymbol{x}_i$ are independent, then for $\boldsymbol{x}_i, \boldsymbol{h}_i = (H_{i,:})^\top, \boldsymbol{h}_i^{\mathrm{HP}} = (H_{i,:}^{\mathrm{HP}})^\top$ we have

$$\mathbb{P}\left(\|\mathbf{x}_i - \mathbf{x}_j\|_2 \geq t\right) \leq \sum_{k=1}^{F_h} \mathbb{P}\left(|\mathbf{x}_{i,k} - \mathbf{x}_{j,k}| \geq \frac{t}{\sqrt{F_h}}\right)$$

$$\mathbb{P}\left(\|\mathbf{h}_i - \mathbf{h}_j\|_2 \geq t\right) \leq \sum_{k=1}^{F_h} \mathbb{P}\left(|\mathbf{h}_{i,k} - \mathbf{h}_{j,k}| \geq \frac{t}{\sqrt{F_h}}\right)$$

$$\mathbb{P}\left(\|\boldsymbol{h}_i^{\mathrm{HP}} - \boldsymbol{h}_j^{\mathrm{HP}}\|_2 \geq t\right) \leq \sum_{k=1}^{F_h} \mathbb{P}\left(|\mathbf{h}_{i,k}^{\mathrm{HP}} - \mathbf{h}_{j,k}^{\mathrm{HP}}| \geq \frac{t}{\sqrt{F_h}}\right)$$

*Proof.* If $\|\mathbf{x}_i - \mathbf{x}_j\|_2 \geq t$, then at least for one $k \in \{1, \ldots, F_h\}$, the inequality $|\mathbf{x}_{i,k} - \mathbf{x}_{j,k}| \geq \frac{t}{\sqrt{F_h}}$ holds. Therefore, we have

$$\mathbb{P}\left(\|\mathbf{x}_i - \mathbf{x}_j\|_2 \geq t\right) \leq \mathbb{P}\left(\bigcup_{k=1}^{F_h}\left\{|\mathbf{x}_{i,k} - \mathbf{x}_{j,k}| \geq \frac{t}{\sqrt{F_h}}\right\}\right)$$

$$\leq \sum_{k=1}^{F_h} \mathbb{P}\left(|\mathbf{x}_{i,k} - \mathbf{x}_{j,k}| \geq \frac{t}{\sqrt{F_h}}\right)$$

The results for $\boldsymbol{h}_i, \boldsymbol{h}_i^{\mathrm{HP}}$ can be proved by analogy. $\qquad\square$

**Lemma 2.** (Heoffding Lemma) Let $X$ be any real-valued random variable such that $a \leq X \leq b$ almost surely, *i.e.,* with probability one. Then, for all $\lambda \in \mathbb{R}$,

$$\mathbb{E}\left[e^{\lambda X}\right] \leq \exp\left(\lambda \mathbb{E}[X] + \frac{\lambda^2 (b-a)^2}{8}\right)$$

*Proof.* See [40] $\qquad\square$

**Theorem 2.** For nodes $i, j, v \in \mathcal{V}$, suppose $z_i \neq z_j$ and $z_i = z_v$, then for constants $t_x, t_h, t_{\mathrm{HP}}$ that satisfy $t_x \geq \sqrt{F_h} D_x(i,j)$, $t_h \geq \sqrt{F_h} D_h(i,j)$, $t_{\mathrm{HP}} \geq \sqrt{F_h} D_{\mathrm{HP}}(i,j)$ we have

$$\mathbb{P}\left(\|\boldsymbol{x}_i - \boldsymbol{x}_j\|_2 \geq \|\boldsymbol{x}_i - \boldsymbol{x}_v\|_2 + t_x\right) \leq 2F_h \exp\left(-\frac{\left(D_x(i,j) - \frac{t_x}{\sqrt{F_h}}\right)^2}{V_x(i,j)}\right),$$

$$\mathbb{P}(\|\boldsymbol{h}_i - \boldsymbol{h}_j\|_2 \geq \|\boldsymbol{h}_i - \boldsymbol{h}_v\|_2 + t_h) \leq 2F_h \exp\left(-\frac{\left(D_h(i,j) - \frac{t_h}{\sqrt{F_h}}\right)^2}{V_h(i,j)}\right), \qquad (7)$$

$$\mathbb{P}(\|\boldsymbol{h}_i^{\mathrm{HP}} - \boldsymbol{h}_j^{\mathrm{HP}}\|_2 \geq \|\boldsymbol{h}_i^{\mathrm{HP}} - \boldsymbol{h}_v^{\mathrm{HP}}\|_2 + t_{\mathrm{HP}}) \leq 2F_h \exp\left(-\frac{\left(D_{\mathrm{HP}}(i,j) - \frac{t_{\mathrm{HP}}}{\sqrt{F_h}}\right)^2}{V_{\mathrm{HP}}(i,j)}\right),$$

where

$$D_x(i,j) = \left\|\boldsymbol{\mu}_{z_i} - \boldsymbol{\mu}_{z_j}\right\|_2, \ V_x(i,j) = (b-a)^2,$$

$$D_h(i,j) = \left\|\tilde{\boldsymbol{\mu}}_{z_i} - \tilde{\boldsymbol{\mu}}_{z_j}\right\|_2, V_h(i,j) = \left(\frac{1}{2d_i} + \frac{1}{2d_j}\right)(b-a)^2,$$

$$D_{\mathrm{HP}}(i,j) = \left\|\boldsymbol{\mu}_{z_i} - \tilde{\boldsymbol{\mu}}_{z_i} - \left(\boldsymbol{\mu}_{z_j} - \tilde{\boldsymbol{\mu}}_{z_j}\right)\right\|_2,$$

$$V_{\mathrm{HP}}(i,j) = \left(1 + \frac{1}{2d_i} + \frac{1}{2d_j}\right)(b-a)^2,$$

$$\tilde{\boldsymbol{\mu}}_{z_i} = \sum_{v \in \mathcal{N}(i)} \mathbb{E}_{\substack{z_v \sim \mathcal{D}_{z_i}, \\ \mathbf{x}_v \sim \mathcal{F}_{z_v}}} \left[\frac{1}{d_i} \boldsymbol{x}_v\right].$$

The proof of Theorem 2 will be splitted into three parts for $\boldsymbol{x}_i, \boldsymbol{h}_i$ and $\boldsymbol{h}_i^{\mathrm{HP}}$, respectively.

## G.1 Proof for Original (Full-pass Filtered) Features

*Proof.* Since we have

$$\|\boldsymbol{x}_i - \boldsymbol{x}_j\|_2 - \|\boldsymbol{x}_i - \boldsymbol{x}_v\|_2 \leq \|\boldsymbol{x}_i - \boldsymbol{x}_j - (\boldsymbol{x}_i - \boldsymbol{x}_v)\|_2 = \|\boldsymbol{x}_v - \boldsymbol{x}_j\|_2$$

then

$$\mathbb{P}\left(\|\boldsymbol{x}_i - \boldsymbol{x}_j\|_2 \geq \|\boldsymbol{x}_i - \boldsymbol{x}_v\|_2 + t_x\right) = \mathbb{P}\left(\|\boldsymbol{x}_i - \boldsymbol{x}_j\|_2 - \|\boldsymbol{x}_i - \boldsymbol{x}_v\|_2 \geq t_x\right)$$
$$\leq \mathbb{P}\left(\|\boldsymbol{x}_i - \boldsymbol{x}_j - (\boldsymbol{x}_i - \boldsymbol{x}_v)\|_2 \geq t_x\right) = \mathbb{P}\left(\|\boldsymbol{x}_v - \boldsymbol{x}_j\|_2 \geq t_x\right)$$

We will calculate the upper bound of $\mathbb{P}\left(\|\boldsymbol{x}_v - \boldsymbol{x}_j\|_2 \geq t_x\right)$ in the following part. To do this, we first compute the upper bound of $\mathbb{P}\left(\boldsymbol{x}_{v,k} - \boldsymbol{x}_{j,k} \geq t\right)$. For $t \geq \left\|\boldsymbol{\mu}_{z_v} - \boldsymbol{\mu}_{z_j}\right\|$ and any $s \geq 0$, we have

$$\mathbb{P}\left(\boldsymbol{x}_{v,k} - \boldsymbol{x}_{j,k} \geq t\right) = \mathbb{P}\left(\exp\left(s(\boldsymbol{x}_{v,k} - \boldsymbol{x}_{j,k})\right) \geq \exp\left(st\right)\right)$$
$$\leq \exp\left(-st\right)\mathbb{E}\left[\exp\left(s(\boldsymbol{x}_{v,k} - \boldsymbol{x}_{j,k})\right)\right] \text{ (Markov Inequality)}$$
$$= \exp\left(-st\right)\mathbb{E}\left[\exp\left(s\boldsymbol{x}_{v,k}\right)\right]\mathbb{E}\left[\exp\left(-s\boldsymbol{x}_{j,k}\right)\right] \text{ (Independency)}$$
$$\leq \exp\left(-st\right)\exp\left(s\mathbb{E}\left[\boldsymbol{x}_{v,k}\right] + \frac{(b-a)^2 s^2}{8}\right) \times \exp\left(-s\mathbb{E}\left[\boldsymbol{x}_{j,k}\right] + \frac{(b-a)^2 s^2}{8}\right) \text{ (Hoeffding's lemma)}$$
$$= \exp\left(\frac{(b-a)^2}{4}s^2 + (\boldsymbol{\mu}_{z_v,k} - \boldsymbol{\mu}_{z_j,k} - t)s\right)$$
$$\leq \exp\left(\frac{(b-a)^2}{4}s^2 + (\left|\boldsymbol{\mu}_{z_v,k} - \boldsymbol{\mu}_{z_j,k}\right| - t)s\right)$$

Since $t \geq \left\|\boldsymbol{\mu}_{z_v} - \boldsymbol{\mu}_{z_j}\right\| \geq \left|\boldsymbol{\mu}_{z_v,k} - \boldsymbol{\mu}_{z_j,k}\right|$ for any $k$, so when $s = -\frac{\left(\left|\boldsymbol{\mu}_{z_v,k} - \boldsymbol{\mu}_{z_j,k}\right| - t\right)}{\frac{(b-a)^2}{2}} \geq 0$, we get the tightest bound of the above inequality and

$$\exp\left(-\frac{\left(\left|\boldsymbol{\mu}_{z_v,k} - \boldsymbol{\mu}_{z_j,k}\right| - t\right)^2}{(b-a)^2}\right) \leq \exp\left(-\frac{\left(\left\|\boldsymbol{\mu}_{z_v} - \boldsymbol{\mu}_{z_j}\right\|_2 - t\right)^2}{(b-a)^2}\right)$$

With the same steps, we have

$$\mathbb{P}\left(\boldsymbol{x}_{v,k} - \boldsymbol{x}_{j,k} \leq -t\right) = \mathbb{P}\left(\boldsymbol{x}_{j,k} - \boldsymbol{x}_{v,k} \geq t\right) \leq \exp\left(-\frac{\left(\left\|\boldsymbol{\mu}_{z_v} - \boldsymbol{\mu}_{z_j}\right\|_2 - t\right)^2}{(b-a)^2}\right)$$

Combined together we have

$$\mathbb{P}\left(\left|\boldsymbol{x}_{v,k} - \boldsymbol{x}_{j,k}\right| \geq t\right) \leq 2\exp\left(-\frac{\left(\left\|\boldsymbol{\mu}_{z_v} - \boldsymbol{\mu}_{z_j}\right\|_2 - t\right)^2}{(b-a)^2}\right)$$

Since $\frac{t_x}{\sqrt{F_h}} \geq \left\|\boldsymbol{\mu}_{z_v} - \boldsymbol{\mu}_{z_j}\right\|$, then from Lemma 1 we have

$$\mathbb{P}\left(\|\boldsymbol{x}_v - \boldsymbol{x}_j\|_2 \geq t_x\right) \leq \sum_{k=1}^{F_h}\mathbb{P}\left(\left|\mathbf{x}_{v,k} - \mathbf{x}_{j,k}\right| \geq \frac{t_x}{\sqrt{F_h}}\right) \leq 2F_h \exp\left(-\frac{\left(\left\|\boldsymbol{\mu}_{z_v} - \boldsymbol{\mu}_{z_j}\right\|_2 - \frac{t_x}{\sqrt{F_h}}\right)^2}{(b-a)^2}\right)$$
(8)
$\square$

## G.2 Proof for Low-pass Filter

*Proof.* Part for LP filter:

Let $\boldsymbol{h}_{i,k} = \frac{1}{d_i}\sum\limits_{\substack{u \in \mathcal{N}(i), \\ z_u \sim \mathcal{D}_{z_i}, \\ \mathbf{x}_{u,k} \sim \mathcal{F}_{z_u,k}}} \boldsymbol{x}_{u,k}$ and $\tilde{\boldsymbol{\mu}}_{z_i,k} = \mathbb{E}\left[\boldsymbol{h}_{i,k}\right] = \mathbb{E}\left[\frac{1}{d_i}\sum \mathbf{x}_{u,k}\right]$. Since we have

$$\|\boldsymbol{h}_i - \boldsymbol{h}_j\|_2 - \|\boldsymbol{h}_i - \boldsymbol{h}_v\|_2 \leq \|\boldsymbol{h}_i - \boldsymbol{h}_j - (\boldsymbol{h}_i - \boldsymbol{h}_v)\|_2 = \|\boldsymbol{h}_v - \boldsymbol{h}_j\|_2$$

then

$$\mathbb{P}\left(\|\boldsymbol{h}_i - \boldsymbol{h}_j\|_2 \geq \|\boldsymbol{h}_i - \boldsymbol{h}_v\|_2 + t_h\right) = \mathbb{P}\left(\|\boldsymbol{h}_i - \boldsymbol{h}_j\|_2 - \|\boldsymbol{h}_i - \boldsymbol{h}_v\|_2 \geq t_h\right)$$
$$\leq \mathbb{P}\left(\|\boldsymbol{h}_i - \boldsymbol{h}_j - (\boldsymbol{h}_i - \boldsymbol{h}_v)\|_2 \geq t_h\right) = \mathbb{P}\left(\|(\boldsymbol{h}_v - \boldsymbol{h}_j)\|_2 \geq t_h\right)$$

We will calculate the upper bound of $\mathbb{P}\left(\|\boldsymbol{h}_v - \boldsymbol{h}_j\|_2 \geq t_h\right)$ in the following part. To do this, we first compute the upper bound of $\mathbb{P}\left(\boldsymbol{h}_{v,k} - \boldsymbol{h}_{j,k} \geq t\right)$. For $t \geq \left\|\tilde{\boldsymbol{\mu}}_{z_v} - \tilde{\boldsymbol{\mu}}_{z_j}\right\|$ and any $s \geq 0$, we have

$$\mathbb{P}\left(\boldsymbol{h}_{v,k} - \boldsymbol{h}_{j,k} \geq t\right) = \mathbb{P}\left(\exp\left(s(\boldsymbol{h}_{v,k} - \boldsymbol{h}_{j,k})\right) \geq \exp\left(st\right)\right)$$
$$\leq \exp\left(-st\right)\mathbb{E}\left[\exp\left(s(\boldsymbol{h}_{v,k} - \boldsymbol{h}_{j,k})\right)\right] \text{ (Markov Inequality)}$$

$$= \exp\left(-st\right)\mathbb{E}\left[\exp\left(\frac{s}{d_v}\sum_{\substack{u\in\mathcal{N}(v),\\ z_u\sim\mathcal{D}_{z_v},\\ \mathbf{x}_{u,k}\sim\mathcal{F}_{z_u,k}}}\boldsymbol{x}_{u,k}\right)\right]\mathbb{E}\left[\exp\left(\frac{-s}{d_j}\sum_{\substack{u\in\mathcal{N}(j),\\ z_u\sim\mathcal{D}_{z_j},\\ \mathbf{x}_{u,k}\sim\mathcal{F}_{z_u,k}}}\boldsymbol{x}_{u,k}\right)\right] \text{ (Independency)}$$

$$= \exp\left(-st\right)\prod_{\substack{u\in\mathcal{N}(v),\\ z_u\sim\mathcal{D}_{z_v},\\ \mathbf{x}_{u,k}\sim\mathcal{F}_{z_u,k}}}\mathbb{E}\left[\exp\left(\frac{s}{d_v}\boldsymbol{x}_{u,k}\right)\right]\prod_{\substack{u\in\mathcal{N}(j),\\ z_u\sim\mathcal{D}_{z_j},\\ \mathbf{x}_{u,k}\sim\mathcal{F}_{z_u,k}}}\mathbb{E}\left[\exp\left(\frac{-s}{d_j}\boldsymbol{x}_{u,k}\right)\right] \text{ (Independency)}$$

$$\leq \exp\left(-st\right)\prod_{\substack{u\in\mathcal{N}(v),\\ z_u\sim\mathcal{D}_{z_v},\\ \mathbf{x}_{u,k}\sim\mathcal{F}_{z_u,k}}}\exp\left(\frac{s}{d_v}\mathbb{E}\left[\boldsymbol{x}_{u,k}\right] + \frac{(b-a)^2s^2}{8d_v^2}\right)$$

$$\times \prod_{\substack{u\in\mathcal{N}(j),\\ z_u\sim\mathcal{D}_{z_j},\\ \mathbf{x}_{u,k}\sim\mathcal{F}_{z_u,k}}}\exp\left(\frac{-s}{d_j}\mathbb{E}\left[\boldsymbol{x}_{u,k}\right] + \frac{(b-a)^2s^2}{8d_j^2}\right) \text{ (Hoeffding's lemma)}$$

$$= \exp\left(-st\right)\exp\left(\frac{(b-a)^2s^2}{8d_v}\right)\exp\left(s\mathbb{E}\left[\frac{1}{d_v}\sum_{\substack{u\in\mathcal{N}(v),\\ z_u\sim\mathcal{D}_{z_v},\\ \mathbf{x}_{u,k}\sim\mathcal{F}_{z_u,k}}}\boldsymbol{x}_{u,k}\right]\right)$$

$$\times \exp\left(\frac{(b-a)^2s^2}{8d_j}\right)\exp\left(-s\mathbb{E}\left[\frac{1}{d_j}\sum_{\substack{u\in\mathcal{N}(j),\\ z_u\sim\mathcal{D}_{z_j},\\ \mathbf{x}_{u,k}\sim\mathcal{F}_{z_u,k}}}\boldsymbol{x}_{u,k}\right]\right)$$

$$= \exp\left(\left(\frac{(b-a)^2}{8d_v} + \frac{(b-a)^2}{8d_j}\right)s^2 + (\tilde{\boldsymbol{\mu}}_{z_v,k} - \tilde{\boldsymbol{\mu}}_{z_j,k} - t)s\right)$$

$$\leq \exp\left(\left(\frac{(b-a)^2}{8d_v} + \frac{(b-a)^2}{8d_j}\right)s^2 + (|\tilde{\boldsymbol{\mu}}_{z_v,k} - \tilde{\boldsymbol{\mu}}_{z_j,k}| - t)s\right)$$

Since $t \geq \left\|\tilde{\boldsymbol{\mu}}_{z_v} - \tilde{\boldsymbol{\mu}}_{z_j}\right\| \geq \left|\tilde{\boldsymbol{\mu}}_{z_v,k} - \tilde{\boldsymbol{\mu}}_{z_j,k}\right|$ for any $k$, so when $s = -\frac{(|\tilde{\boldsymbol{\mu}}_{z_v,k} - \tilde{\boldsymbol{\mu}}_{z_j,k}| - t)}{\frac{(b-a)^2}{4d_v} + \frac{(b-a)^2}{4d_j}} \geq 0$, we get the tightest bound of the above inequality and

$$\exp\left(-\frac{(|\tilde{\boldsymbol{\mu}}_{z_v,k} - \tilde{\boldsymbol{\mu}}_{z_j,k}| - t)^2}{\frac{(b-a)^2}{2d_v} + \frac{(b-a)^2}{2d_j}}\right) \leq \exp\left(-\frac{(\|\tilde{\boldsymbol{\mu}}_{z_v} - \tilde{\boldsymbol{\mu}}_{z_j}\|_2 - t)^2}{\frac{(b-a)^2}{2d_v} + \frac{(b-a)^2}{2d_j}}\right)$$

With the same steps, we have

$$\mathbb{P}\left(\boldsymbol{h}_{v,k} - \boldsymbol{h}_{j,k} \leq -t\right) = \mathbb{P}\left(\boldsymbol{h}_{j,k} - \boldsymbol{h}_{v,k} \geq t\right) \leq \exp\left(-\frac{\left(\left\|\tilde{\boldsymbol{\mu}}_{z_v} - \tilde{\boldsymbol{\mu}}_{z_j}\right\|_2 - t\right)^2}{\frac{(b-a)^2}{2d_v} + \frac{(b-a)^2}{2d_j}}\right)$$

Combined together we have

$$\mathbb{P}\left(\left|\boldsymbol{h}_{v,k} - \boldsymbol{h}_{j,k}\right| \geq t\right) \leq 2\exp\left(-\frac{\left(\left\|\tilde{\boldsymbol{\mu}}_{z_v} - \tilde{\boldsymbol{\mu}}_{z_j}\right\|_2 - t\right)^2}{\frac{(b-a)^2}{2d_v} + \frac{(b-a)^2}{2d_j}}\right)$$

Since $\frac{t_h}{\sqrt{F_h}} \geq \left\|\tilde{\boldsymbol{\mu}}_{z_v} - \tilde{\boldsymbol{\mu}}_{z_j}\right\|$, then from Lemma 1 we have

$$\mathbb{P}\left(\left\|\boldsymbol{h}_v - \boldsymbol{h}_j\right\|_2 \geq t_h\right) \leq \sum_{k=1}^{F_h} \mathbb{P}\left(\left|\mathbf{h}_{v,k} - \mathbf{h}_{j,k}\right| \geq \frac{t_h}{\sqrt{F_h}}\right) \leq 2F_h \exp\left(-\frac{\left(\left\|\tilde{\boldsymbol{\mu}}_{z_v} - \tilde{\boldsymbol{\mu}}_{z_j}\right\|_2 - \frac{t_h}{\sqrt{F_h}}\right)^2}{\frac{(b-a)^2}{2d_v} + \frac{(b-a)^2}{2d_j}}\right)$$

$$(9)$$

$\square$

### G.3 Theoretical Results for High-pass Filter

*Proof.* The proof for HP filter is similar to that of LP filter.

Let $\boldsymbol{h}_i^{\mathrm{HP}} = \boldsymbol{x}_i - \boldsymbol{h}_i$, which is the HP filtered signal. Since we have

$$\left\|\boldsymbol{h}_i^{\mathrm{HP}} - \boldsymbol{h}_j^{\mathrm{HP}}\right\|_2 - \left\|\boldsymbol{h}_i^{\mathrm{HP}} - \boldsymbol{h}_v^{\mathrm{HP}}\right\|_2 \leq \left\|\boldsymbol{h}_i^{\mathrm{HP}} - \boldsymbol{h}_j^{\mathrm{HP}} - (\boldsymbol{h}_i^{\mathrm{HP}} - \boldsymbol{h}_v^{\mathrm{HP}})\right\|_2 = \left\|\boldsymbol{h}_v^{\mathrm{HP}} - \boldsymbol{h}_j^{\mathrm{HP}}\right\|_2$$

then

$$\mathbb{P}\left(\left\|\boldsymbol{h}_i^{\mathrm{HP}} - \boldsymbol{h}_j^{\mathrm{HP}}\right\|_2 \geq \left\|\boldsymbol{h}_i^{\mathrm{HP}} - \boldsymbol{h}_v^{\mathrm{HP}}\right\|_2 + t_{\mathrm{HP}}\right) = \mathbb{P}\left(\left\|\boldsymbol{h}_i^{\mathrm{HP}} - \boldsymbol{h}_j^{\mathrm{HP}}\right\|_2 - \left\|\boldsymbol{h}_i^{\mathrm{HP}} - \boldsymbol{h}_v^{\mathrm{HP}}\right\|_2 \geq t_{\mathrm{HP}}\right)$$
$$\leq \mathbb{P}\left(\left\|\boldsymbol{h}_i^{\mathrm{HP}} - \boldsymbol{h}_j^{\mathrm{HP}} - (\boldsymbol{h}_i^{\mathrm{HP}} - \boldsymbol{h}_v^{\mathrm{HP}})\right\|_2 \geq t_{\mathrm{HP}}\right) = \mathbb{P}\left(\left\|\boldsymbol{h}_v^{\mathrm{HP}} - \boldsymbol{h}_j^{\mathrm{HP}})\right\|_2 \geq t_{\mathrm{HP}}\right)$$

We will calculate the upper bound of $\mathbb{P}\left(\left\|\boldsymbol{h}_v^{\mathrm{HP}} - \boldsymbol{h}_j^{\mathrm{HP}}\right\|_2 \geq t\right)$ in the following part. To do this, we first compute the upper bound of $\mathbb{P}\left(\boldsymbol{h}_{v,k}^{\mathrm{HP}} - \boldsymbol{h}_{j,k}^{\mathrm{HP}} \geq t\right)$.

For $t \geq \left\|\boldsymbol{\mu}_v - \tilde{\boldsymbol{\mu}}_v - (\boldsymbol{\mu}_j - \tilde{\boldsymbol{\mu}}_j)\right\|_2$ and $s \geq 0$, we have

$$\mathbb{P}\left(\boldsymbol{h}_{v,k}^{\mathrm{HP}} - \boldsymbol{h}_{j,k}^{\mathrm{HP}} \geq t\right) = \mathbb{P}\left(\boldsymbol{x}_{v,k} - \boldsymbol{h}_{v,k} - \boldsymbol{x}_{j,k} + \boldsymbol{h}_{j,k} \geq t\right)$$
$$= \mathbb{P}\left(\exp\left(s(\boldsymbol{x}_{v,k} - \boldsymbol{h}_{v,k} - \boldsymbol{x}_{j,k} + \boldsymbol{h}_{j,k})\right) \geq \exp\left(st\right)\right)$$
$$\leq \exp\left(-st\right)\mathbb{E}\left[\exp\left(s(\boldsymbol{x}_{v,k} - \boldsymbol{h}_{v,k} - \boldsymbol{x}_{j,k} + \boldsymbol{h}_{j,k})\right)\right] \text{ (Markov Inequality)}$$
$$= \exp\left(-st\right) \times \mathbb{E}\left[\exp\left(s\boldsymbol{x}_{v,k}\right)\right] \times \mathbb{E}\left[\exp\left(-s\boldsymbol{x}_{j,k}\right)\right] \times$$

$$\mathbb{E}\left[\exp\left(-\frac{s}{d_v}\sum_{\substack{u\in\mathcal{N}(v),\\ z_u\sim\mathcal{D}_{z_v},\\ \mathbf{x}_{u,k}\sim\mathcal{F}_{z_u,k}}}\boldsymbol{x}_{u,k}\right)\right]\mathbb{E}\left[\exp\left(\frac{s}{d_j}\sum_{\substack{u\in\mathcal{N}(j),\\ z_u\sim\mathcal{D}_{z_j},\\ \mathbf{x}_{u,k}\sim\mathcal{F}_{z_u,k}}}\boldsymbol{x}_{u,k}\right)\right] \text{ (Independency)}$$

$$\leq \exp\left(-st\right)\mathbb{E}\left[\exp\left(s\boldsymbol{x}_{v,k}\right)\right]\mathbb{E}\left[\exp\left(-s\boldsymbol{x}_{j,k}\right)\right]$$

$$\prod_{\substack{u\in\mathcal{N}(v),\\ z_u\sim\mathcal{D}_{z_v},\\ \mathbf{x}_{u,k}\sim\mathcal{F}_{z_u,k}}}\mathbb{E}\left[\exp\left(\frac{-s}{d_v}\boldsymbol{x}_{u,k}\right)\right]\prod_{\substack{u\in\mathcal{N}(j),\\ z_u\sim\mathcal{D}_{z_j},\\ \mathbf{x}_{u,k}\sim\mathcal{F}_{z_u,k}}}\mathbb{E}\left[\exp\left(\frac{s}{d_j}\boldsymbol{x}_{u,k}\right)\right]$$

$$\leq \exp\left(-st\right)\exp\left(s\boldsymbol{\mu}_{v,k} + \frac{(b-a)^2s^2}{8}\right)\exp\left(-s\boldsymbol{\mu}_{j,k} + \frac{(b-a)^2s^2}{8}\right)$$

$$\prod\exp\left(\frac{-s}{d_v}\mathbb{E}[\boldsymbol{x}_{u,k}] + \frac{(b-a)^2s^2}{8d_v^2}\right)\prod\exp\left(\frac{s}{d_j}\mathbb{E}[\boldsymbol{x}_{u,k}] + \frac{(b-a)^2s^2}{8d_j^2}\right) \text{ (Hoeffding's lemma)}$$

$$= \exp\left((\frac{(b-a)^2}{4} + \frac{(b-a)^2}{8d_v} + \frac{(b-a)^2}{8d_j})s^2 + \left(\boldsymbol{\mu}_{v,k} - \boldsymbol{\mu}_{j,k} - (\tilde{\boldsymbol{\mu}}_{v,k} - \tilde{\boldsymbol{\mu}}_{j,k}) - t\right)s\right)$$

$$\leq \exp\left((\frac{(b-a)^2}{4} + \frac{(b-a)^2}{8d_v} + \frac{(b-a)^2}{8d_j})s^2 + \left(\left|\boldsymbol{\mu}_{v,k} - \boldsymbol{\mu}_{j,k} - (\tilde{\boldsymbol{\mu}}_{v,k} - \tilde{\boldsymbol{\mu}}_{j,k})\right| - t\right)s\right)$$

Since $t \geq \left\|\boldsymbol{\mu}_v - \tilde{\boldsymbol{\mu}}_v - \left(\boldsymbol{\mu}_j - \tilde{\boldsymbol{\mu}}_j\right)\right\|_2$, then when $s = -\frac{\left|\boldsymbol{\mu}_{v,k} - \boldsymbol{\mu}_{j,k} - (\tilde{\boldsymbol{\mu}}_{v,k} - \tilde{\boldsymbol{\mu}}_{j,k})\right| - t}{\left(\frac{(b-a)^2}{2} + \frac{(b-a)^2}{4d_v} + \frac{(b-a)^2}{4d_j}\right)} > 0$, we get the tightest bound and

$$\exp\left(-\frac{\left(\left|\boldsymbol{\mu}_{v,k} - \boldsymbol{\mu}_{j,k} - (\tilde{\boldsymbol{\mu}}_{v,k} - \tilde{\boldsymbol{\mu}}_{j,k})\right| - t\right)^2}{(1 + \frac{1}{2d_v} + \frac{1}{2d_j})(b-a)^2}\right) \leq \exp\left(-\frac{\left(\left\|\boldsymbol{\mu}_v - \tilde{\boldsymbol{\mu}}_v - \left(\boldsymbol{\mu}_j - \tilde{\boldsymbol{\mu}}_j\right)\right\|_2 - t\right)^2}{(1 + \frac{1}{2d_v} + \frac{1}{2d_j})(b-a)^2}\right)$$

Then

$$\mathbb{P}\left(\left|\boldsymbol{h}_{v,k}^{\mathrm{HP}} - \boldsymbol{h}_{j,k}^{\mathrm{HP}}\right| \geq t\right) \leq 2\exp\left(-\frac{\left(\left\|\boldsymbol{\mu}_v - \tilde{\boldsymbol{\mu}}_v - \left(\boldsymbol{\mu}_j - \tilde{\boldsymbol{\mu}}_j\right)\right\|_2 - t\right)^2}{(1 + \frac{1}{2d_v} + \frac{1}{2d_j})(b-a)^2}\right)$$

Since $\frac{t_{\mathrm{HP}}}{\sqrt{F_h}} \geq \left\|\boldsymbol{\mu}_v - \tilde{\boldsymbol{\mu}}_v - \left(\boldsymbol{\mu}_j - \tilde{\boldsymbol{\mu}}_j\right)\right\|_2$, then from Lemma 1, we have

$$\mathbb{P}\left(\left\|\boldsymbol{h}_v^{\mathrm{HP}} - \boldsymbol{h}_j^{\mathrm{HP}}\right\|_2 \geq t_{\mathrm{HP}}\right) \leq \sum_{k=1}^{F_h}\mathbb{P}\left(\left|\mathbf{h}_{v,k}^{\mathrm{HP}} - \boldsymbol{h}_{j,k}^{\mathrm{HP}}\right| \geq \frac{t_{\mathrm{HP}}}{\sqrt{F_h}}\right)$$

$$\leq 2F_h\exp\left(-\frac{\left(\left\|\boldsymbol{\mu}_v - \tilde{\boldsymbol{\mu}}_v - \left(\boldsymbol{\mu}_j - \tilde{\boldsymbol{\mu}}_j\right)\right\|_2 - \frac{t_{\mathrm{HP}}}{\sqrt{F_h}}\right)^2}{(1 + \frac{1}{2d_v} + \frac{1}{2d_j})(b-a)^2}\right)$$
(10)

$\left\|\boldsymbol{\mu}_v - \tilde{\boldsymbol{\mu}}_v - \left(\boldsymbol{\mu}_j - \tilde{\boldsymbol{\mu}}_j\right)\right\|_2$ is essentially the relative center movement.

$\square$

# H   Detailed Discussion of Performance Metrics and More Experimental Results

## H.1   Hypothesis Testing of ACM-GNNs vs. GNNs

To more comprehensively validate if "intra-class embedding distance is smaller than the inter-class embedding distance" closely correlates to the superiority of a given model versus another model, we choose the SOTA model ACM-GNNs and conduct the following hypothesis testing of ACM-GNNs [35] versus GNNs and ACM-GNNs versus MLPs. From the results in table 3 we can see that the above statements hold except in ACM-SGC-1 vs. SGC-1 on Squirrel and ACM-GCN vs. GCN on CiteSeer. This again verifies that the relationship between intra- and inter-class embedding distance strongly relates to the model performance.

| | | Cornell | Wisconsin | Texas | Film | Chameleon | Squirrel | Cora | CiteSeer | PubMed |
|---|---|---|---|---|---|---|---|---|---|---|
| ACM-SGC-1 v.s. SGC-1 | p-value | 1.00 | 1.00 | 1.00 | 1.00 | 0.19 | 1.00 | 1.00 | 1.00 | 1.00 |
| | ACC ACM-SGC-1 | 93.77 ± 1.91 | 93.25 ± 2.92 | 93.61 ± 1.55 | 39.33 ± 1.25 | 63.68 ± 1.62 | 46.4 ± 1.13 | 86.63 ± 1.13 | 80.96 ± 0.93 | 87.75 ± 0.88 |
| | ACC SGC-1 | 70.98 ± 8.39 | 70.38 ± 2.85 | 83.28 ± 5.43 | 25.26 ± 1.18 | 64.86 ± 1.81 | 47.62 ± 1.27 | 85.12 ± 1.64 | 79.66 ± 0.75 | 85.5 ± 0.76 |
| | **Diff Acc** | 22.79 | 22.87 | 10.33 | 14.07 | -1.18 | -1.22 | 1.51 | 1.30 | 2.25 |
| ACM-GCN v.s. GCN | p-value | 1.00 | 1.00 | 1.00 | 1.00 | 1.00 | 1.00 | 0.41 | 0.00 | 1.00 |
| | ACC ACM-GCN | 94.75 ± 3.8 | 95.75 ± 2.03 | 94.92 ± 2.88 | 41.62 ± 1.15 | 69.04 ± 1.74 | 58.02 ± 1.86 | 88.62 ± 1.22 | 81.68 ± 0.97 | 90.66 ± 0.47 |
| | ACC GCN | 82.46 ± 3.11 | 75.5 ± 2.92 | 83.11 ± 3.2 | 35.51 ± 0.99 | 64.18 ± 2.62 | 44.76 ± 1.39 | 87.78 ± 0.96 | 81.39 ± 1.23 | 88.9 ± 0.32 |
| | **Diff Acc** | 12.29 | 20.25 | 11.81 | 6.11 | 4.86 | 13.26 | 0.84 | 0.29 | 1.76 |
| ACM-SGC-1 v.s. MLP-1 | p-value | 0.10 | 0.00 | 0.50 | 1.00 | 1.00 | 1.00 | 1.00 | 1.00 | 0.42 |
| | ACC ACM-SGC-1 | 93.77 ± 1.91 | 93.25 ± 2.92 | 93.61 ± 1.55 | 39.33 ± 1.25 | 63.68 ± 1.62 | 46.4 ± 1.13 | 86.63 ± 1.13 | 80.96 ± 0.93 | 87.75 ± 0.88 |
| | ACC MLP-1 | 93.77 ± 3.34 | 93.87 ± 3.33 | 93.77 ± 3.34 | 34.53 ± 1.48 | 45.01 ± 1.58 | 29.17 ± 1.46 | 74.3 ± 1.27 | 75.51 ± 1.35 | 86.23 ± 0.54 |
| | **Diff Acc** | 0.00 | -0.62 | -0.16 | 4.80 | 18.67 | 17.23 | 12.33 | 5.45 | 1.52 |
| ACM-GCN v.s. MLP-2 | p-value | 0.94 | 1.00 | 1.00 | 1.00 | 1.00 | 1.00 | 1.00 | 1.00 | 1.00 |
| | ACC ACM-GCN | 94.75 ± 3.8 | 95.75 ± 2.03 | 94.92 ± 2.88 | 41.62 ± 1.15 | 69.04 ± 1.74 | 58.02 ± 1.86 | 88.62 ± 1.22 | 81.68 ± 0.97 | 90.66 ± 0.47 |
| | ACC MLP-2 | 91.30 ± 0.70 | 93.87 ± 3.33 | 92.26 ± 0.71 | 38.58 ± 0.25 | 46.72 ± 0.46 | 31.28 ± 0.27 | 76.44 ± 0.30 | 76.25 ± 0.28 | 86.43 ± 0.13 |
| | **Diff Acc** | 3.45 | 1.88 | 2.66 | 3.04 | 22.32 | 26.74 | 12.18 | 5.43 | 4.23 |

Table 3: Hypothesis testing results of ACM-GNNs v.s. GNNs: The cells marked by orange are the cases that the p-values significantly indicate the opposite direction as the trained results (ground truth).

## H.2   Implementation Details of KR and GNB

Classifier-based performance metrics: we measure the quality of aggregated features based on the performance of a "training-free" classifier. In this paper, we take use of kernel regression and naive Bayes classifiers.

**Kernel Regression**   Kernel method utilizes a pairwise similarity function $K(\boldsymbol{x}_i, \boldsymbol{x}_j)$ to measure how closely related two node embeddings are, without the need for any training process [29, 22, 43]. A higher value of $K(\boldsymbol{x}_i, \boldsymbol{x}_j)$ indicates a smaller distance between the embeddings of nodes $\boldsymbol{x}_i$ and $\boldsymbol{x}_j$ and vice versa.

---

**Algorithm 1** Pseudo code for kernel regression

---

**Require:** $X, \hat{A}, Z, N, N_S, N_{\text{epochs}}$           $\triangleright$ $N$ is the number of nodes, $N_S$ is the number of samples
    **for** $i$ in $N_{\text{epochs}}$ **do**
        $S \leftarrow \text{sample}(N, N_S)$
        Get $K_S^X, K_S^H, Z_S$                 $\triangleright$ $K_S^X, K_S^H$ are the kernels for $X$ and $H$ for the sampled nodes
        $S_{\text{train}}, S_{\text{test}} \leftarrow \text{sample}(S, 0.6N_S, 0.4N_S)$
        $f_K^X \leftarrow \left(K_S^X[S_{\text{test}}, :][:, S_{\text{train}}]\right)\left(K_S^X[S_{\text{train}}, :][:, S_{\text{train}}]\right)^{-1} Z_S[S_{\text{train}}, :]$
        $f_K^H \leftarrow \left(K_S^H[S_{\text{test}}, :][:, S_{\text{train}}]\right)\left(K_S^H[S_{\text{train}}, :][:, S_{\text{train}}]\right)^{-1} Z_S[S_{\text{train}}, :]$
        Compute $\text{ACC}_i^X, \text{ACC}_i^H \leftarrow \text{Accuracy}(f_K^X, Z_S[S_{\text{test}}, :]), \text{ Accuracy}(f_K^H, Z_S[S_{\text{test}}, :])$
    **end for**
    p-value $\leftarrow \text{ttest}(\text{ACC}^X, \text{ACC}^H)$

---

To capture the **feature-based non-linear node similarity**, we use Neural Network Gaussian Process (NNGP) [30, 2, 16, 41]. Specifically, we consider the activation function $\phi(x) = \text{ReLU}(x)$ and have

$$K_{\mathrm{NL}}(\boldsymbol{x}_i, \boldsymbol{x}_j) = \frac{1}{2\pi} \left( \boldsymbol{x}_i^\top \boldsymbol{x}_j \left( \pi - \tilde{\phi} \left( \frac{\boldsymbol{x}_i^\top \boldsymbol{x}_j}{\|\boldsymbol{x}_i\|_2 \|\boldsymbol{x}_j\|_2} \right) \right) + \sqrt{\|\boldsymbol{x}_i\|_2^2 \|\boldsymbol{x}_j\|_2^2 - \left( \boldsymbol{x}_i^\top \boldsymbol{x}_j \right)^2} \right)$$

where $\tilde{\phi}(x) = \arccos(x)$ is the dual activation function of ReLU. [13]

Furthermore, we observe that there exist some datasets where linear G-aware models do not have the same performance disparities compared to their coupled G-agnostic models as non-linear G-aware models, *e.g.*, as the results on PubMed shown in table 1, SGC-1 underperforms MLPs while GCN outperforms MLP-2. This implies that relying on a single non-linear metric to assess whether G-aware models will surpass their coupled G-agnostic models is not enough, we need a linear metric as well. Thus, we choose the following linear kernel (inner product) for regression

$$K_{\mathrm{L}}(\boldsymbol{x}_i, \boldsymbol{x}_j) = \frac{\boldsymbol{x}_i^\top \boldsymbol{x}_j}{\|\boldsymbol{x}_i\|_2 \|\boldsymbol{x}_j\|_2}$$

.

For Gaussian Naïve Bayes (GNB), we just use the features and aggregated features of the sampled training nodes to fit two separate classifiers and get the predicted accuracy for the test nodes. Note that Gaussian Naïve Bayes is just a linear classifier.

**Threshold Values** Typically, the threshold for homophily and heterophily graphs is set at 0.5 [57, 55, 54, 35] . For classifier-based performance metrics, we establish two benchmark thresholds as below,

- Normal Threshold 0.5 (NT0.5): Although not indicating statistical significance, we are still comfortable to set 0.5 as a loose threshold. A value exceeding 0.5 suggests that the G-aware model is not very likely to underperform their coupled G-agnostic model on the tested graph and vice versa.

- Statistical Significant Threshold 0.05 (SST0.05): Instead of offering an ambiguous statistical interpretation, SST0.05 will offer a clear statistical meaning. A value smaller than 0.05 implies that the G-aware model significantly underperforms their coupled G-agnostic model and a value greater than 0.95 suggests a high likelihood of G-aware model outperforming their coupled G-agnostic model. Besides, a value ranging from 0.05 to 0.95 indicates no significant performance distinction between G-aware model and its G-agnostic model.

We show the results of $\mathrm{KR_L}$, $\mathrm{KR_{NL}}$ and GNB in section H.3 . Cells marked by grey are errors according to NT0.5 and results marked by red are incorrect according to SST 0.05. The comparisons with the existing homophily metrics are shown in section H.4. We can see that, no matter on small- (table 4, 6) or large-scale (table 5, 7) datasets, the classifier-based performance metrics (CPMs) are significantly better than the existing homophily metrics on revealing the advantages and disadvantages of GNNs, decreasing the overall error rate from at least 0.34 to 0.13 (table 8). The running time of CPM is short (table 9), only taking several minutes [14] even on large-scale datasets such as pokec and snap-patents, which contains millons of nodes and tens of millions of edges [32].

## H.3 Results on Small-scale and Large-scale Datasets

---

[13]Note that when $\phi(x) = \exp(ix)$, we have $K(\boldsymbol{x}_i, \boldsymbol{x}_j) = \exp\left(-\frac{1}{2}\|\boldsymbol{x}_i - \boldsymbol{x}_j\|_2^2\right)$, which is closely related to the Euclidean distance of node embeddings tested in section 4.1, further emphasizing the strong relationship between embedding distances and kernel similarities.

[14]1 NVIDIA V100 GPU with 16G memory, 8-core CPU with 16G memory

| | | Cornell | Wisconsin | Texas | Film | Chameleon | Squirrel | Cora | CiteSeer | PubMed |
|---|---|---|---|---|---|---|---|---|---|---|
| Baseline Homophily Metrics | $H_{edge}$ | 0.5669 | 0.4480 | 0.4106 | 0.3750 | 0.2795 | 0.2416 | 0.8100 | 0.7362 | 0.8024 |
| | $H_{node}$ | 0.3855 | 0.1498 | 0.0968 | 0.2210 | 0.2470 | 0.2156 | 0.8252 | 0.7175 | 0.7924 |
| | $H_{class}$ | 0.0468 | 0.0941 | 0.0013 | 0.0110 | 0.0620 | 0.0254 | 0.7657 | 0.6270 | 0.6641 |
| | $H_{agg}$ | 0.8032 | 0.7768 | 0.6940 | 0.6822 | 0.61 | 0.3566 | 0.9904 | 0.9826 | 0.9432 |
| | $H_{GE}$ | 0.31 | 0.34 | 0.35 | 0.16 | 0.0152 | 0.0157 | 0.1700 | 0.1900 | 0.27 |
| | $H_{adj}$ | 0.1889 | 0.0826 | 0.0258 | 0.1272 | 0.0663 | 0.0196 | 0.8178 | 0.7588 | 0.7431 |
| | LI | 0.0169 | 0.1311 | 0.1923 | 0.0002 | 0.048 | 0.0015 | 0.5904 | 0.4508 | 0.4093 |
| Classifier-based Performance Metrics | $KR_L$ | 1.39 | 0.00 | 0.00 | 0.7834 | 1.00 | 1.00 | 1.00 | 1.00 | 0.8026 |
| | GNB | 0.00 | 0.00 | 0.00 | 0.00 | 1.00 | 1.00 | 1.00 | 1.00 | 1.0000 |
| SGC v.s. MLP1 | ACC SGC | 70.98 ± 8.39 | 70.38 ± 2.85 | 83.28 ± 5.43 | 25.26 ± 1.18 | 64.86 ± 1.81 | 47.62 ± 1.27 | 85.12 ± 1.64 | 79.66 ± 0.75 | 85.5 ± 0.76 |
| | ACC MLP-1 | 93.77 ± 3.34 | 93.87 ± 3.33 | 93.77 ± 3.34 | 34.53 ± 1.48 | 45.01 ± 1.58 | 29.17 ± 1.46 | 74.3 ± 1.27 | 75.51 ± 1.35 | 86.23 ± 0.54 |
| | **Diff Acc** | -22.79 | -23.49 | -10.49 | -9.27 | 19.85 | 18.45 | 10.82 | 4.15 | -0.73 |
| Baseline Homophily Metrics | $H_{edge}$ | 0.5669 | 0.4480 | 0.4106 | 0.3750 | 0.2795 | 0.2416 | 0.8100 | 0.7362 | 0.8024 |
| | $H_{node}$ | 0.3855 | 0.1498 | 0.0968 | 0.2210 | 0.2470 | 0.2156 | 0.8252 | 0.7175 | 0.7924 |
| | $H_{class}$ | 0.0468 | 0.0941 | 0.0013 | 0.0110 | 0.0620 | 0.0254 | 0.7657 | 0.6270 | 0.6641 |
| | $H_{agg}$ | 0.8032 | 0.7768 | 0.6940 | 0.6822 | 0.61 | 0.3566 | 0.9904 | 0.9826 | 0.9432 |
| | $H_{GE}$ | 0.31 | 0.34 | 0.35 | 0.16 | 0.0152 | 0.0157 | 0.1700 | 0.1900 | 0.2700 |
| | $H_{adj}$ | 0.1889 | 0.0826 | 0.0258 | 0.1272 | 0.0663 | 0.0196 | 0.8178 | 0.7588 | 0.7431 |
| | LI | 0.0169 | 0.1311 | 0.1923 | 0.0002 | 0.048 | 0.0015 | 0.5904 | 0.4508 | 0.4093 |
| Classifier-based Performance Metrics | $KR_{NL}$ | 0.00 | 0.00 | 0.00 | 0.00 | 1.00 | 1.00 | 1.00 | 1.00 | 1.00 |
| | GNB | 0.00 | 0.00 | 0.00 | 0.00 | 1.00 | 1.00 | 1.00 | 1.00 | 1.00 |
| GCN v.s. MLP2 | ACC GCN | 82.46 ± 3.11 | 75.5 ± 2.92 | 83.11 ± 3.2 | 35.51 ± 0.99 | 64.18 ± 2.62 | 44.76 ± 1.39 | 87.78 ± 0.96 | 81.39 ± 1.23 | 88.9 ± 0.32 |
| | ACC MLP-2 | 91.30 ± 0.70 | 93.87 ± 3.33 | 92.26 ± 0.71 | 38.58 ± 0.25 | 46.72 ± 0.46 | 31.28 ± 0.27 | 76.44 ± 0.30 | 76.25 ± 0.28 | 86.43 ± 0.13 |
| | **Diff Acc** | -8.84 | -18.37 | -9.15 | -3.07 | 17.46 | 13.48 | 11.34 | 5.14 | 2.47 |

Table 4: Comparison on small datasets

| | | Penn94 | pokec | arXiv-year | snap-patents | genius | twitch-gamers | Deezer-Europe |
|---|---|---|---|---|---|---|---|---|
| Baseline Homophily Metrics | $H_{edge}$ | 0.4700 | 0.4450 | 0.2220 | 0.0730 | 0.6180 | 0.5450 | 0.5250 |
| | $H_{node}$ | 0.4828 | 0.4283 | 0.2893 | 0.2206 | 0.5087 | 0.5564 | 0.5299 |
| | $H_{class}$ | 0.0460 | 0.0000 | 0.2720 | 0.1000 | 0.0800 | 0.0900 | 0.0300 |
| | $H_{agg}$ | 0.2712 | 0.0807 | 0.7066 | 0.6170 | 0.7823 | 0.4172 | 0.5580 |
| | $H_{GE}$ | 0.3734 | 0.9222 | 0.8388 | 0.6064 | 0.6655 | 0.2865 | 0.0378 |
| | $H_{adj}$ | 0.0366 | -0.1132 | 0.0729 | 0.0907 | 0.1432 | 0.1010 | 0.1586 |
| | LI | 0.0851 | 0.0172 | 0.0407 | 0.0243 | 0.0025 | 0.0058 | 0.0007 |
| Classifier-based Performance Metrics | $KR_L$ | 0.00 | 0.03 | 0.98 | 0.19 | 0.00 | 0.25 | 0.00 |
| | GBN | 0.00 | 0.00 | 1.00 | 1.00 | 0.00 | 1.00 | 0.00 |
| SGC vs MLP1 | ACC SGC | 67.06 ± 0.19 | 52.88 ± 0.64 | 35.58 ± 0.22 | 29.65 ± 0.04 | 82.31 ± 0.45 | 57.9 ± 0.18 | 61.63 ± 0.25 |
| | ACC MLP-1 | 73.72 ± 0.5 | 59.89 ± 0.11 | 34.11 ± 0.17 | 30.59 ± 0.02 | 86.48 ± 0.11 | 59.45 ± 0.16 | 63.14 ± 0.41 |
| | **Diff Acc** | -6.66 | -7.01 | 1.47 | -0.94 | -4.17 | -1.55 | -1.51 |
| Baseline Homophily Metrics | $H_{edge}$ | 0.4700 | 0.4450 | 0.2220 | 0.0730 | 0.6180 | 0.5450 | 0.5250 |
| | $H_{node}$ | 0.4828 | 0.4283 | 0.2893 | 0.2206 | 0.5087 | 0.5564 | 0.5299 |
| | $H_{class}$ | 0.0460 | 0.0000 | 0.2720 | 0.1000 | 0.0800 | 0.0900 | 0.0300 |
| | $H_{agg}$ | 0.2712 | 0.0807 | 0.7066 | 0.6170 | 0.7823 | 0.4172 | 0.5580 |
| | $H_{GE}$ | 0.3734 | 0.9222 | 0.8388 | 0.6064 | 0.6655 | 0.2865 | 0.0378 |
| | $H_{adj}$ | 0.0366 | -0.1132 | 0.0729 | 0.0907 | 0.1432 | 0.1010 | 0.1586 |
| | LI | 0.0851 | 0.0172 | 0.0407 | 0.0243 | 0.0025 | 0.0058 | 0.0007 |
| Classifier-based Performance Metrics | $KR_{NL}$ | 0.00 | 0.57 | 1.00 | 0.4083 | 0.00 | 1.00 | 0.00 |
| | GNB | 0.00 | 0.00 | 1.00 | 1.00 | 0.00 | 1.00 | 0.00 |
| GCN vs MLP2 | ACC GCN | 82.08 ± 0.31 | 70.3 ± 0.1 | 40 ± 0.26 | 35.8 ± 0.05 | 83.26 ± 0.14 | 62.33 ± 0.23 | 60.16 ± 0.51 |
| | ACC MLP-2 | 74.68 ± 0.28 | 62.13 ± 0.1 | 36.36 ± 0.23 | 31.43 ± 0.04 | 86.62 ± 0.08 | 60.9 ± 0.11 | 64.25 ± 0.41 |
| | **Diff Acc** | 7.40 | 8.17 | 3.64 | 4.37 | -3.36 | 1.43 | -4.09 |

Table 5: Comparison on large-scale datasets

## H.4 Statistics and Comparisons

**Discrepancy Between Linear and Non-linear Models** From the experimental results on large-scale datasets reported in Table 5, we observe that, for linear and non-linear G-aware models, there exists inconsistency between their comparison with their coupled G-agnostics models. For example, on *Penn94, pokec, snap-patents* and *twitch-gamers*, SGC-1 underperforms MLP-1 but GCN outperforms MLP-2. In fact, PubMed in Table 4 also belongs to this family of datasets. We do not have a proved theory to explain this phenomenon for now. But there is obviously a synergy between homophily/heterophily and non-linearity that cause this discrepancy together. And we think, on this special subset of heterophilic graphs, we should develop theoretical analysis to discuss the interplay between graph structure and feature non-linearity, and how they affect node distinguishability together.

The current homophily values (including the proposed metrics) are not able to explain the phenomenon associated with this group of datasets. We keep it as an open question and encourage people from the GNN community to study it in the future.

|  | Total Error | Error Rate |
|---|---|---|
| $H_{edge}$ | 7 | 0.39 |
| $H_{node}$ | 5 | 0.28 |
| $H_{class}$ | 5 | 0.28 |
| $H_{agg}$ | 11 | 0.61 |
| $H_{GE}$ | 9 | 0.50 |
| $H_{adj}$ | 5 | 0.28 |
| LI | 7 | 0.39 |
| KR (NT0.5) | 2 | 0.11 |
| KR (SST0.05) | 1 | 0.06 |
| GNB (NT0.5) | 1 | 0.06 |
| GNB (SST0.05) | 1 | 0.06 |

Table 6: Statistics on small-scale datasets

|  | Total Error | Error Rate |
|---|---|---|
| $H_{edge}$ | 9 | 0.64 |
| $H_{node}$ | 9 | 0.64 |
| $H_{class}$ | 6 | 0.43 |
| $H_{agg}$ | 8 | 0.57 |
| $H_{GE}$ | 6 | 0.43 |
| $H_{adj}$ | 6 | 0.43 |
| LI | 6 | 0.43 |
| KR (NT0.5) | 2 | 0.14 |
| KR (SST0.05) | 5 | 0.36 |
| GNB (NT0.5) | 4 | 0.29 |
| GNB (SST0.05) | 4 | 0.29 |

Table 7: Statistics on large-scale datasets

|  | Total Error | Error Rate |
|---|---|---|
| $H_{edge}$ | 16 | 0.50 |
| $H_{node}$ | 14 | 0.44 |
| $H_{class}$ | 11 | 0.34 |
| $H_{agg}$ | 19 | 0.59 |
| $H_{GE}$ | 15 | 0.47 |
| $H_{adj}$ | 11 | 0.34 |
| LI | 13 | 0.41 |
| KR (NT0.5) | 4 | 0.13 |
| KR (SST0.05) | 6 | 0.19 |
| GNB (NT0.5) | 5 | 0.16 |
| GNB (SST0.05) | 5 | 0.16 |

Table 8: Overall statistics on small- and large-scale datasets

| | KR$_L$ | KR$_{NL}$ | GNB |
|---|---|---|---|
| Cornell | 0.58 | 0.67 | 1.39 |
| Wisconsin | 0.78 | 0.87 | 1.72 |
| Texas | 0.59 | 0.67 | 1.41 |
| Film | 5.29 | 5.41 | 2.72 |
| Chameleon | 3.97 | 3.95 | 3.81 |
| Squirrel | 5.39 | 5.36 | 4.15 |
| Cora | 3.94 | 4.10 | 3.08 |
| CiteSeer | 4.85 | 5.05 | 6.55 |
| PubMed | 9.35 | 9.41 | 5.27 |
| Penn94 | 18.57 | 18.68 | 12.43 |
| pokec | 84.47 | 86.08 | 50.03 |
| arXiv-year | 7.77 | 7.82 | 4.56 |
| snap-patents | 304.06 | 296.21 | 163.84 |
| genius | 8.20 | 8.12 | 5.30 |
| twitch-gamers | 9.34 | 9.24 | 4.17 |
| Deezer-Europe | 37.41 | 39.49 | 59.84 |

Table 9: Total running time (seconds/100 samples) of KR$_L$, KR$_{NL}$ and GNB

## H.5 Results for Symmetric Renormalized Affinity Matrix

To evaluate if the benefits of classifier-based performance metrics can be maintained for different aggregation operators, we replace the random walk renormalized affinity matrix with synmmetric renormalized affinity matrix in SGC-1, GCN, KR$_L$, KR$_{NL}$ and GNB and report the results and comparisons as belows.

It is observed that the superiority holds on both small- (table 10, 12) and large-scale datasets (table 11, 13), reducing the overall error rate from at least 0.31 to 0.13 (table 14).

| | | Cornell | Wisconsin | Texas | Film | Chameleon | Squirrel | Cora | CiteSeer | PubMed |
|---|---|---|---|---|---|---|---|---|---|---|
| Baseline Homophily Metrics | H$_{edge}$ | 0.5669 | 0.4480 | 0.4106 | 0.3750 | 0.2795 | 0.2416 | 0.8100 | 0.7362 | 0.8024 |
| | H$_{node}$ | 0.3855 | 0.1498 | 0.0968 | 0.2210 | 0.2470 | 0.2156 | 0.8252 | 0.7175 | 0.7924 |
| | H$_{class}$ | 0.0468 | 0.0941 | 0.0013 | 0.0110 | 0.0620 | 0.0254 | 0.7657 | 0.6270 | 0.6641 |
| | H$_{agg}$ | 0.8032 | 0.7768 | 0.6940 | 0.6822 | 0.61 | 0.3566 | 0.9904 | 0.9826 | 0.9432 |
| | H$_{GE}$ | 0.31 | 0.34 | 0.35 | 0.16 | 0.0152 | 0.0157 | 0.1700 | 0.1900 | 0.2700 |
| | H$_{adj}$ | 0.1889 | 0.0826 | 0.0258 | 0.1272 | 0.0663 | 0.0196 | 0.8178 | 0.7588 | 0.7431 |
| | LI | 0.0169 | 0.1311 | 0.1923 | 0.0002 | 0.048 | 0.0015 | 0.5904 | 0.4508 | 0.4093 |
| Classifier-based Performance Metrics | KR$_L$ | 0.00 | 0.00 | 0.00 | 0.9304 | 1.00 | 1.00 | 1.00 | 1.00 | 0.0003 |
| | GNB | 0.00 | 0.00 | 0.00 | 0.00 | 1.00 | 1.00 | 1.00 | 1.00 | 1.00 |
| SGC vs MLP1 | ACC SGC | 51.64 ± 12.27 | 39.63 ± 5.39 | 30.82 ± 4.96 | 27.02 ± 1 | 63.26 ± 1.98 | 46.03 ± 1.74 | 84.38 ± 1.5 | 79.51 ± 1.04 | 87.24 ± 0.44 |
| | ACC MLP-1 | 93.77 ± 3.34 | 93.87 ± 3.33 | 93.77 ± 3.34 | 34.53 ± 1.48 | 45.01 ± 1.58 | 29.17 ± 1.46 | 74.3 ± 1.27 | 75.51 ± 1.35 | 86.23 ± 0.54 |
| | **Diff Acc** | -42.13 | -54.24 | -62.95 | -7.51 | 18.25 | 16.86 | 10.08 | 4.00 | 1.01 |
| Baseline Homophily Metrics | H$_{edge}$ | 0.5669 | 0.4480 | 0.4106 | 0.3750 | 0.2795 | 0.2416 | 0.8100 | 0.7362 | 0.8024 |
| | H$_{node}$ | 0.3855 | 0.1498 | 0.0968 | 0.2210 | 0.2470 | 0.2156 | 0.8252 | 0.7175 | 0.7924 |
| | H$_{class}$ | 0.0468 | 0.0941 | 0.0013 | 0.0110 | 0.0620 | 0.0254 | 0.7657 | 0.6270 | 0.6641 |
| | H$_{agg}$ | 0.8032 | 0.7768 | 0.6940 | 0.6822 | 0.61 | 0.3566 | 0.9904 | 0.9826 | 0.9432 |
| | H$_{GE}$ | 0.31 | 0.34 | 0.35 | 0.16 | 0.0152 | 0.0157 | 0.1700 | 0.1900 | 0.2700 |
| | H$_{adj}$ | 0.1889 | 0.0826 | 0.0258 | 0.1272 | 0.0663 | 0.0196 | 0.8178 | 0.7588 | 0.7431 |
| | LI | 0.0169 | 0.1311 | 0.1923 | 0.0002 | 0.048 | 0.0015 | 0.5904 | 0.4508 | 0.4093 |
| Classifier-based Performance Metrics | KR$_{NL}$ | 0.00 | 0.00 | 0.00 | 0.00 | 1.00 | 1.00 | 1.00 | 1.00 | 0.98 |
| | GNB | 0.00 | 0.00 | 0.00 | 0.00 | 1.00 | 1.00 | 1.00 | 1.00 | 1.00 |
| GCN vs MLP2 | ACC GCN | 82.62 ± 3.04 | 70.38 ± 3.16 | 82.46 ± 2.94 | 35.79 ± 1.09 | 68.95 ± 1.09 | 52.98 ± 0.85 | 87.87 ± 0.99 | 81.79 ± 1.09 | 89.47 ± 0.27 |
| | ACC MLP-2 | 91.30 ± 0.70 | 93.87 ± 3.33 | 92.26 ± 0.71 | 38.58 ± 0.25 | 46.72 ± 0.46 | 31.28 ± 0.27 | 76.44 ± 0.30 | 76.25 ± 0.28 | 86.43 ± 0.13 |
| | **Diff Acc** | -8.68 | -23.49 | -9.80 | -2.79 | 22.23 | 21.70 | 11.43 | 5.54 | 3.04 |

Table 10: Results for symmetric renormalized affinity matrix on small-scale datasets

| | | Penn94 | pokec | arXiv-year | snap-patents | genius | twitch-gamers | Deezer-Europe |
|---|---|---|---|---|---|---|---|---|
| | $H_{edge}$ | 0.4700 | 0.4450 | 0.2220 | 0.0730 | 0.6180 | 0.5450 | 0.5250 |
| | $H_{node}$ | 0.4828 | 0.4283 | 0.2893 | 0.2206 | 0.5087 | 0.5564 | 0.5299 |
| Baseline | $H_{class}$ | 0.0460 | 0.0000 | 0.2720 | 0.1000 | 0.0800 | 0.0900 | 0.0300 |
| Homophily | $H_{agg}$ | 0.2712 | 0.0807 | 0.7066 | 0.6170 | 0.7823 | 0.4172 | 0.5580 |
| Metrics | $H_{GE}$ | 0.3734 | 0.9222 | 0.8388 | 0.6064 | 0.6655 | 0.2865 | 0.0378 |
| | $H_{adj}$ | 0.0366 | -0.1132 | 0.0729 | 0.0907 | 0.1432 | 0.1010 | 0.1586 |
| | LI | 0.0851 | 0.0172 | 0.0407 | 0.0243 | 0.0025 | 0.0058 | 0.0007 |
| Classifier-based | $KR_L$ | 0.00 | 0.02 | 0.32 | 0.46 | 0.00 | 0.00 | 0.00 |
| Performance Metrics | GNB | 0.00 | 0.03 | 1.00 | 1.00 | 0.00 | 0.97 | 0.00 |
| | ACC SGC | 64.63 ± 0.15 | 51.97 ± 0.38 | 35.24 ± 0.14 | 30.32 ± 0.05 | 81.66 ± 0.58 | 58.77 ± 0.18 | 60.2 ± 0.47 |
| SGC vs MLP1 | ACC MLP-1 | 73.72 ± 0.5 | 59.89 ± 0.11 | 34.11 ± 0.17 | 30.59 ± 0.02 | 86.48 ± 0.11 | 59.45 ± 0.16 | 63.14 ± 0.41 |
| | **Diff Acc** | -9.09 | -7.92 | 1.13 | -0.27 | -4.82 | -0.68 | -2.94 |
| | $H_{edge}$ | 0.4700 | 0.4450 | 0.2220 | 0.0730 | 0.6180 | 0.5450 | 0.5250 |
| | $H_{node}$ | 0.4828 | 0.4283 | 0.2893 | 0.2206 | 0.5087 | 0.5564 | 0.5299 |
| Baseline | $H_{class}$ | 0.0460 | 0.0000 | 0.2720 | 0.1000 | 0.0800 | 0.0900 | 0.0300 |
| Homophily | $H_{agg}$ | 0.2712 | 0.0807 | 0.7066 | 0.6170 | 0.7823 | 0.4172 | 0.5580 |
| Metrics | $H_{GE}$ | 0.3734 | 0.9222 | 0.8388 | 0.6064 | 0.6655 | 0.2865 | 0.0378 |
| | $H_{adj}$ | 0.0366 | -0.1132 | 0.0729 | 0.0907 | 0.1432 | 0.1010 | 0.1586 |
| | LI | 0.0851 | 0.0172 | 0.0407 | 0.0243 | 0.0025 | 0.0058 | 0.0007 |
| Classifier-based | $KR_{NL}$ | 0.00 | 0.32 | 0.99 | 0.99 | 0.00 | 1.00 | 0.00 |
| Performance Metrics | GNB | 0.00 | 0.29 | 1.00 | 1.00 | 0.00 | 0.97 | 0.00 |
| | ACC GCN | 81.45 ± 0.29 | 69.55 ± 0.1 | 40.02 ± 0.19 | 35.4 ± 0.04 | 83.02 ± 0.14 | 62.59 ± 0.14 | 62.32 ± 0.44 |
| GCN vs MLP2 | ACC MLP-2 | 74.68 ± 0.28 | 62.13 ± 0.1 | 36.36 ± 0.23 | 31.43 ± 0.04 | 86.62 ± 0.08 | 60.9 ± 0.11 | 64.25 ± 0.41 |
| | **Diff Acc** | 6.77 | 7.42 | 3.66 | 3.97 | -3.60 | 1.69 | -1.93 |

Table 11: Results for symmetric renormalized affinity matrix on large-scale datasets

| | Total Error | Error Rate |
|---|---|---|
| $H_{edge}$ | 6 | 0.33 |
| $H_{node}$ | 4 | 0.22 |
| $H_{class}$ | 4 | 0.22 |
| $H_{agg}$ | 10 | 0.56 |
| $H_{GE}$ | 10 | 0.56 |
| $H_{adj}$ | 4 | 0.22 |
| LI | 8 | 0.44 |
| KR (NT0.5) | 2 | 0.11 |
| KR (SST0.05) | 2 | 0.11 |
| GNB (NT0.5) | 0 | 0.00 |
| GNB (SST0.05) | 0 | 0.00 |

Table 12: Statistics for symmetric renormalized affinity matrix on small-scale datasets

| | Total Error | Error Rate |
|---|---|---|
| $H_{edge}$ | 10 | 0.71 |
| $H_{node}$ | 10 | 0.71 |
| $H_{class}$ | 6 | 0.43 |
| $H_{agg}$ | 8 | 0.57 |
| $H_{GE}$ | 6 | 0.43 |
| $H_{adj}$ | 6 | 0.43 |
| LI | 6 | 0.43 |
| KR (NT0.5) | 3 | 0.21 |
| KR (SST0.05) | 4 | 0.29 |
| GNB (NT0.5) | 4 | 0.29 |
| GNB (SST0.05) | 4 | 0.29 |

Table 13: Statistics for symmetric renormalized affinity matrix on large-scale datasets

|  | Total Error | Error Rate |
|---|---|---|
| $H_{edge}$ | 16 | 0.50 |
| $H_{node}$ | 14 | 0.44 |
| $H_{class}$ | 10 | 0.31 |
| $H_{agg}$ | 18 | 0.56 |
| $H_{GE}$ | 16 | 0.50 |
| $H_{adj}$ | 10 | 0.31 |
| LI | 14 | 0.44 |
| $KR_{NNGP}$ (NT0.5) | 5 | 0.16 |
| $KR_{NNGP}$ (SST0.05) | 6 | 0.19 |
| GNB (NT0.5) | 4 | 0.13 |
| GNB (SST0.05) | 4 | 0.13 |

Table 14: Overall statistics for symmetric renormalized affinity matrix on small- and large-scale datasets

## H.6 Experiments on Synthetic Graphs

To comprehensively investigate and corroborate the correlation between CPMs and the performance of G-aware models versus their corresponding G-agnostic models across the entire spectrum of homophily levels, we conduct experiments with the synthetic graphs. The data generation process is similar to [35].

**Data Generation & Experimental Setup** We generated a total of 280 graphs with 28 different levels of edge homophily, ranging from 0.005 to 0.95, and generated 10 graphs for each homophily level. Each graph consisted of 5 classes, with 400 nodes in each class. For nodes in each class, we randomly generated 4000 intra-class edges and $[\frac{4000}{H_{edge}(\mathcal{G})} - 4000]$ inter-class edges, and assigned features to the nodes using the *Cora, CiteSeer, PubMed, Chameleon, Squirrel, Film* datasets. We then randomly split the nodes into train/validation/test sets in a 60%/20%/20% ratio. We trained GCN, SGC-1, MLP-2, and MLP-1 models on the synthetic graphs with fine-tuned hyperparameters as [35]. For each edge homophily level $H_{edge}(\mathcal{G})$, we computed the average test accuracy of the 4 models, as well as $KR_L$, $KR_{NL}$ and other homophily metrics. The comparisons of $KR_L$, $KR_{NL}$ and the performance of GCN vs. MLP-2, SGC-1 vs. MLP-1 according to edge homophily were shown in Figure 11.

It can be observed in Figure 11 that the points where $KR_L$ intersects NT0.5 or SST0.05 (green) and the intersections of SGC-1 and MLP-1 performance (red) are perfectly matched and the curve of $KR_L$ share the similar U-shape as SGC-1, so do $KR_{NL}$ curve (blue) and GCN and MLP-2 performance curves (black) [15]. This indicates that the advantages and disadvantages of G-aware models over G-agnostic models can be better revealed by CPMs at different homophily levels than the baseline homophily metrics shown in Figure 12.

In Figure 12, we can see that the curves of node homophily (orange), class homophily (pink), generalized edge homophily (yellow) and adjusted homophily (blue) are almost linear increasing, which does not reflect the U-shaped performance curve of GNNs' performance. Although the curves for aggregation homophily (purple) and label informativeness (grey) have a rebound in low homophily area, they are unable to provide a suitable threshold value and fails to capture the intersection points.

Since the values of CPMs are either (very close to) 0 or (very close to) 1 and there do not exist enough intermediate values between 0 and 1, we do not plot the relationship between GNNs performance and CPMs as Figure 2 in [35].

---

[15]We only draw the vertical dot lines for the intersection of $KR_L$ and NT0.5 in order to keep the figures clear. The corresponding x-values for other intersections can be observed from the figures.

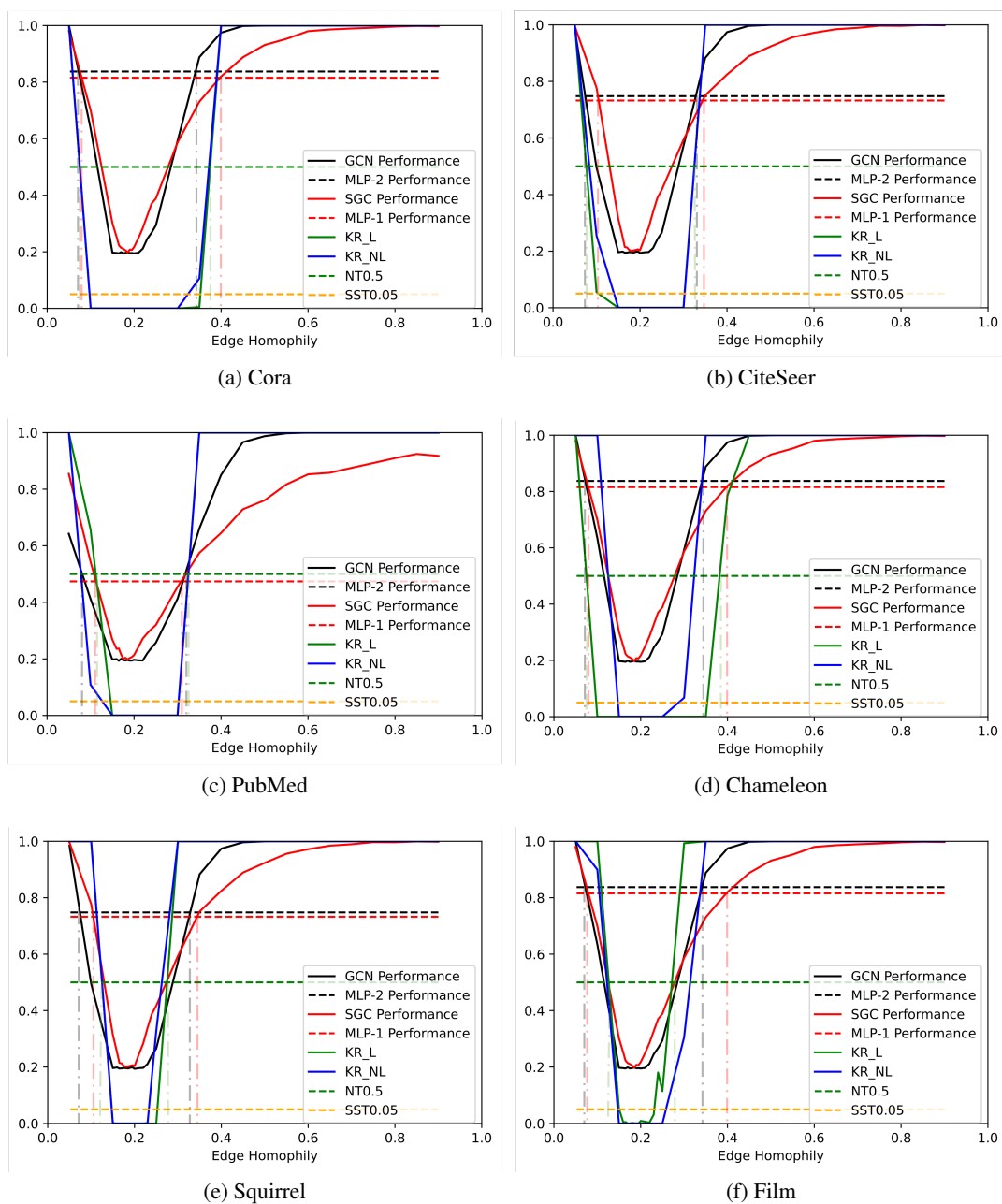

Figure 11: Results and comparisons of $KR_L$ and $KR_{NL}$ on synthetic graphs

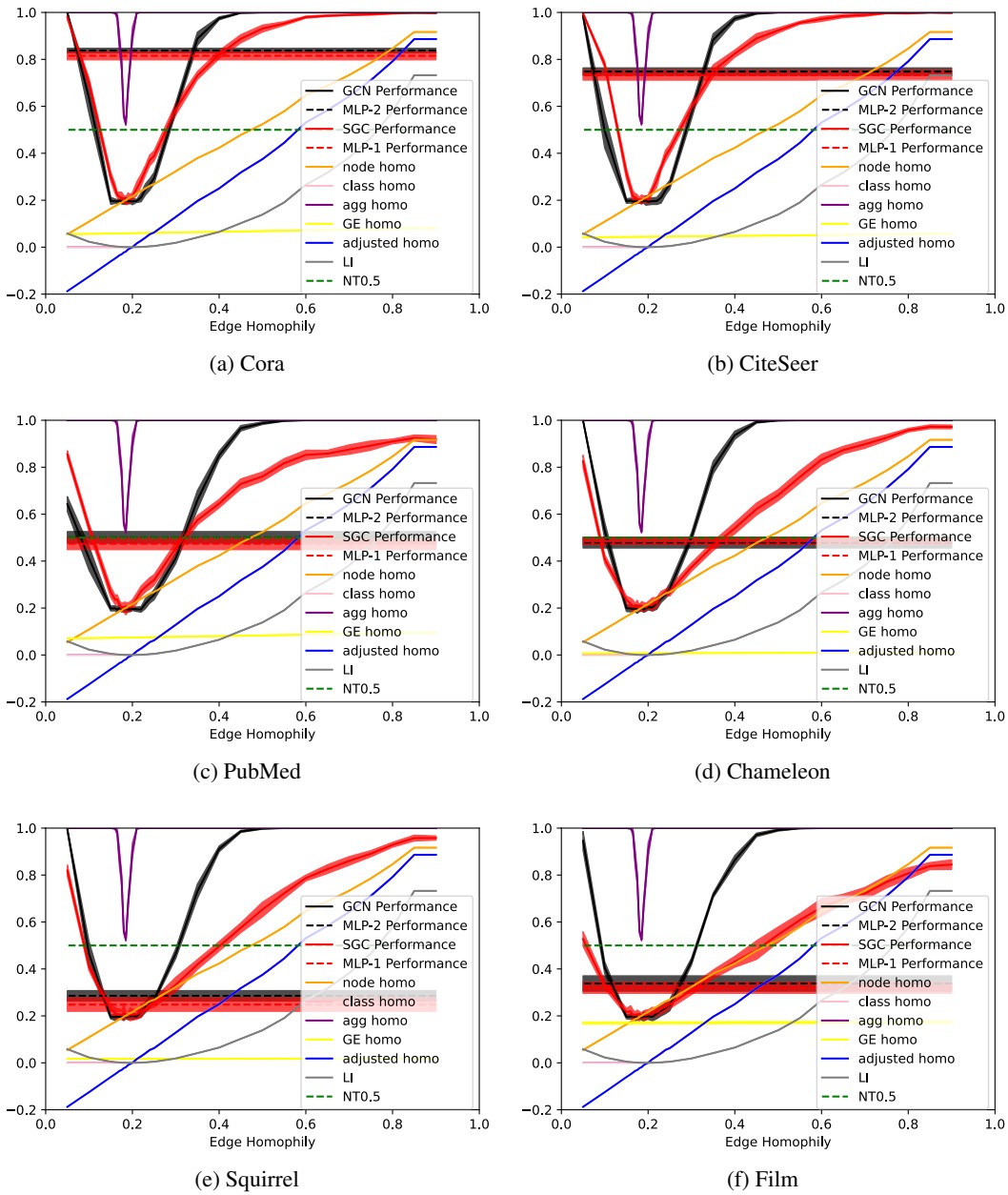

Figure 12: Results on Synthetic Graphs