# OpenReview forum: "When Do Graph Neural Networks Help with Node Classification? Investigating the Homophily Principle on Node Distinguishability"
_NeurIPS.cc/2023/Conference — NeurIPS 2023 poster_

### Official Review · Reviewer_7wh2 · 2023-06-30

**Soundness:** 3 good
**Presentation:** 2 fair
**Contribution:** 2 fair
**Rating:** 5
**Confidence:** 3

**Summary:**

This paper addresses the question: when do graph-aware models outperform graph-agnostic ones on node classification tasks? First, theoretical analysis is conducted. Most of the analysis assumes the two-class CSBM-H model – a generalization of the standard stochastic block model, where node features are sampled from normal distributions (with parameters depending on class labels), and parameter $h$ controls the homophily level. Then, Probabilistic Bayes Error and negative generalized Jeffreys divergence are used to quantify node distinguishability for several node representations: the original node features, aggregated node features (one step of random walk-based aggregation), and high-pass filtered features (after one step of the corresponding aggregation). The analysis shows that different representations are beneficial for different homophily regimes.

In the experiments, it is first shown that if a model has more distinguishable representation, then it usually performs better (Section 4.1). In Section 4.2, it is proposed to predict whether graph-aware models are better than graph-agnostic ones by training a simple classifier on original and aggregated features.

**Strengths:**

The theoretical analysis of PBE and Jeffreys divergence for the CSBM-H and the discussion in Section 3.4 are intuitive and allow one to understand why different aggregation types are helpful for different homophily regimes.

**Weaknesses:**

- The analysis assumes a particular aggregation (random walk), and it is not clear whether conclusions may change for other aggregations.
- The paper is limited to heterophilous datasets known to have certain drawbacks [1,2].
- The proposed measure (Section 4.2) is quite straightforward: it suggests predicting the relative performance of a GNN by training its simplified variant.
- The paper is hard to follow in several places.

I also have the following comments and questions.

1. On the definition of CSBM-H. According to the definition, $h \cdot d_0$ and $h \cdot d_1$ have to be integer. Also, possible values of $d_0$ and $d_1$ depend on the class sizes (the number of outgoing edges should be equal for two classes). This would not be necessary if the graphs are assumed to be directed, but this is not the case according to Section 2.

2. There is another simple measure called Label Informativeness [3] that is known to better agree with GNN performance than homophily [3].

3. The explanation in lines 168-172 was unclear to me; a more detailed explanation would help.

Minor:
- Definition of $h_k$ in (2): $Z_{v,k}$ does not depend on $u$, so it can be moved to the subscript of the sum.
- L112: it is written that all homophily measures are feature-independent, but $H_{GE}$ depends on features.
- Definition 1: notation $P(CL_{Bayes}(x))$ is not clear.

Some typos:
- L48: “Literatures”
- L100: inconsistency of singular and plural
- L111: “imply” → “implying”
- L120 and below: quotes are typed incorrectly, it should be ``like this'' in latex
- L149: “the the”
- L170: “the fixed the classifier”

[1] Lim D. et al. Large scale learning on non-homophilous graphs: New benchmarks and strong simple methods. NeurIPS 2021.

[2] Platonov O. et al. A critical look at the evaluation of GNNs under heterophily: Are we really making progress? ICLR 2023.

[3] Platonov O. et al. Characterizing graph datasets for node classification: Beyond homophily-heterophily dichotomy. ArXiv:2209.06177, 2022.

**Questions:**

The pipeline in Section 4.1 seems overcomplicated. There are two statistical tests, and one is used to define Prop(GCN). It seems that here one can compute the proportion of nodes whose average intra-class node distance is smaller than the inter-class node distance. Would this work or statistical test on this step is necessary?

**Limitations:**

Limitations are not discussed.

---

> ### Author Rebuttal · Authors · 2023-08-09
>
> ### Q1:
> The paper is limited to heterophilous datasets known to have certain drawbacks [1,2].
>
> ### R1:
> Although our paper studies the effect of heterophily, our analysis and experiments are not limited to heterophilous datasets. In fact, the analysis in Section 3 investigate the impact of graph structures under different homophily levels (from high to low, or $h$ from 1 to 0) comprehensively. For the experimental part in Section 4, the real-world benchmark datasets include both homophilic ($\textit{Cora, CiteSeer, PubMed}$) and heterophilic ($\textit{Cornell, Wisconsin, Texas, Film, Chameleon, Squirrel}$) graphs. Additionally in Appendix G.4, we test performance metrics on large-scale benchmark datasets, which also cover both homophilic and heterophilic datasets.
>
> ### Q2:
> The proposed measure (Section 4.2) is quite straightforward: it suggests predicting the relative performance of a GNN by training its simplified variant.
>
> ### R2:
> We would like to clarify and restate the contributions of the proposed metric: 1. As mentioned in Section 4.2, instead of "training its simplified variant", we emphasized that the qualified classifier should not require "training"; 2. The main contribution of the proposed metric is not only using an 'untrained' classifier, but also leveraging the proposed principle "intra-class embedding distance is smaller than the inter-class embedding distance" to construct it. The new metrics based on the above principle can provide accurate statistical threshold value to predict the superiority of G-aware models and is verified to be effective. In addition, we are the first to introduce hypothesis testing to construct performance metric and also the first to show that it is a feasible way besides homophily metrics to derive better metrics. We believe this new direction can help the community to develop metrics with better properties in the future.
>
> ### Q3:
> On the definition of CSBM-H. According to the definition, $hd_0$ and $hd_1$ have to be integer. Also, possible values of $d_0$ and $d_1$ depend on the class sizes (the number of outgoing edges should be equal for two classes). This would not be necessary if the graphs are assumed to be directed, but this is not the case according to Section 2.
>
> ### R3:
> In practice, $hd_0$ and $hd_1$ need to be interger values.
> But we relax them to be continuous values in figures in Section 3.4 because we want to make the curves more readable and intuitive, especially to show the intersections of the 3 zones. We will add comments to clarify this point for CSBM-H to avoid unnecessary confusion in the revised version.
>
> Thanks for bringing up the discussion of the "possible values of $d_0$ and $d_1$", here are our thoughts: If we impose undirected assumption in CSBM-H, we have to not only discuss
> the node degree from intra-class edges, but also discuss degree from inter-class edges (as you said) and control their relations with the corresponding homophily level. This will inevitably add more parameters to CSBM-H and make the model much more complicated. However,
> we find that this complication does not bring us extra benefit for understanding the effect of homophily, which deviate the main goal of our paper. And we guess this might be one of the reasons that the exisiting work mainly keep the discussion within the directed setting [1].
>
> Actually, when CSMB-H was first designed, we would like to only have one "free parameter" $h$ in it to make it simple, because in this way, we are able to show the whole piciture of the effect of homophily from 0 to 1 like the figures in Section 3.4.
>
> All in all, we find your suggestion interesting. We will add discussion in the revised version and encourage the GNN community to think about more CSMB-H variants with more complicated  assumptions, e.g. with different class homophily, different node local homophily distributions, different node degree and class variance distributions and the undirected assuption as you proposed, etc..
>
> ### Q4:
> The explanation in lines 168-172 was unclear to me; a more detailed explanation would help.
>
> ### R4:
> The decision boundary in [1] is defined as $P= \\{ x|w^Tx-w^T(\mu_0+\mu_1)/2 \\}$ where $w= (\mu_0-\mu_1)/||\mu_0-\mu_1||$ is a fixed parameter. This classifier only depends on $\mu_0, \mu_1$ and is fixed for different homophily levels $h$. However, as $h$ changes, the distributions of the two normal distributions will change and the "seperability" of the two normals will be different as well. Thus, the fixed classifier used in [1] is not qualified to quantify the node distinguishability of CSBM-H or any two-normal model with homophily.
> Hope this explanation is helpful.
>
> ### Q5:
> The analysis assumes a particular aggregation (random walk), and it is not clear whether conclusions may change for other aggregations.
>
> ### R5:
> In Appendix G.6 (in supplementary material), we report the results for classifier-based performance metrics with symmetric renormalized affinity matrix and compare them with the existing metrics. We observed similar performance advantages as the random walks renormalized matrix.
>
> The analysis in Section 3.2 and Section 3.3 will be the same. And for the ablation study in 3.4, we re-draw the figures for symmetric renormalized affinity matrix. We observed the similar results for the 3 curves and 3 zones, and thus the conclusions will remain the same. We will add the results and discussion for symmetric renormalized affinity matrix to the revised version.
>
> Besides, as mentioned in line 94-95, random walk aggregation is commenly used in GNN community to study heterophily [1] and we keep consistent with the current literature in our paper.
>
> ### Q6:
> ... Label Informativeness ...
>
> ### R6:
> Thanks for point it out. Please see the Author Rebuttal box for the results and comparisons of Label Informativeness and adjusted homophily.
>
> [1] Is Homophily a Necessity for Graph Neural Networks?. In International Conference on Learning Representations.

---

> > ### Author Response · Authors · 2023-08-10
> > **Response to Reviewer 7wh2 Part (2/2)**
> >
> > ### Q7:
> > The pipeline in Section 4.1 seems overcomplicated. There are two statistical tests, and one is used to define Prop(GCN). It seems that here one can compute the proportion of nodes whose average intra-class node distance is smaller than the inter-class node distance. Would this work or statistical test on this step is necessary?
> >
> > ### R7:
> > Thanks for carefully going through this detail. The statistical test in computing Prop(GCN) is necessary to avoid noisy nodes. In practice, for lots of nodes, we observed that (intra-class node distance):(inter-class node distance) is approximately 1:1, especially when the labels are sparse when we use sampling method. This will not only cause instability of the outputs, but also result in false results sometimes. Thus, we don't want to take account these "marginal nodes" into the comparison of Prop values and we found that using another hypothesis test would help a lot. This is also consistent with our goal to  test the "proportion of nodes whose intra-class node distance is $\textbf{significantly smaller}$ than inter-class node distance" as mentioned in line 299-300.
> >
> > ### Q8:
> > Definition of $h_k$ in (2): $Z_{v,k}$ does not depend on $u$, so it can be moved to the subscript of the sum.
> >
> > ### R8:
> > Thanks for your suggestion. We will modify it in the revised version.
> >
> > ### Q9:
> > L112: it is written that all homophily measures are feature-independent, but $H_\text{GE}$  depends on features.
> >
> > ### R9:
> > Thanks for your comment. We will modify it to "...almost all homophily metrics..."  in the revised version.
> >
> > ### Q10:
> > Definition 1: notation $P(\textup{CL}_{\textup{Bayes}}(\textbf{x}))$ is not clear.
> >
> > ### R10:
> > Thanks for your suggestion. We will modify it to $P(\textup{CL}_{\textup{Bayes}}(\textbf{x}) | \textbf{x})$ in the revised version. Hope this notation can clear up your confusion.
> >
> >
> > ### Q11:
> > The paper is hard to follow in several places.
> >
> > ### R11:
> > Could you please point out several places so that we can address your confusion directly?

---

> > > ### Comment · Reviewer_7wh2 · 2023-08-14
> > >
> > > Thank you for your detailed response and new results with LI and adjusted homophily, I've updated my score.
> > >
> > > Minor comment regarding Q10: as I understand, $CL_{Bayes}(x)$ is a value, while for probability we need an event (like $CL_{Bayes}(x)=0$).

---

> > > > ### Author Response · Authors · 2023-08-15
> > > > **Thank you**
> > > >
> > > > Thanks so much for your time to re-evaluate our paper. Your comments and suggestions really help us to improve our paper.
> > > >
> > > > For Q10, we will modify it to $P(\textup{CL}_{\textup{Bayes}}(\textbf{x}) = z | \textbf{x})$ in the revised version.

---

### Official Review · Reviewer_NHN6 · 2023-07-05

**Soundness:** 4 excellent
**Presentation:** 3 good
**Contribution:** 3 good
**Rating:** 6
**Confidence:** 4

**Summary:**

This paper studies when Graph Neural Networks (GNNs) can help with node classification tasks. The authors first focus on a variant of the Contextual Stochastic Block Model (CSBM) and propose new metrics for node distinguishability based on this model. The authors carry out comprehensive experiments and empirically characterize the regimes when original, low-pass filtered and high-pass filtered features are more helpful. In addition, the authors propose two additional hypothesis testing based metrics for deciding if GNNs are helpful, and empirically verify the effectiveness of the new metrics over real data. The experiments show that the new metrics are more indicative of whether GNNs can lead to better classification accuracy over baseline graph-agnostic models (i.e. MLPs).

**Strengths:**

- The effect of graph homophily/heterophily to the performance of GNNs is an important topic which has received a lot of interests. This paper provides a more comprehensive perspective to the subject.

- The experiments are carefully designed and reasonably comprehensive. The empirical results provide a good support to the main claims of the paper.

- Overall, the paper is well organized.

**Weaknesses:**

- I think the overall writing can be improved. I did not feel excited when reading the paper up to and including page 5. This is just my feeling and it does not mean the paper is not good. Moreover, I find myself sometimes lost focus when reading the first 5 pages. Maybe adding 1-2 sentences describing the emphasis of each section at the beginning of the section can be helpful.

- It seems to me that Section 3 and Section 4 study completely different things and under different settings. Although they are related. I think the results of Section 3 and Section 4 can be considered as two separate contributions of the paper. It might be a good idea to make this clear from the beginning. Currently, this gap between the two sections is not clear from neither the abstract nor the introduction.

- There are a few typos in the paper. For example:
  - Line 18: it significantly -> it is significantly
  - Line 112, the authors claim that "the current homophily metrics are all ... feature independent", however, the generalized edge homophily is feature dependent.
  - Line 148: inner-class ND -> intra-class ND. Better be consistent since you used intra-class everywhere else.
  - Most figures involving PBE, e.g. Figure 2a, 3a, have a different naming in the legend than others. Better to have a consistent naming in figure legend.


**Questions:**

- For Figure 2, since d_0 = d_1 = 5, how did you obtain the plot at h=0.5? It's fine to apply some smooth interpolation but the middle point is important since it is the peak for LP. Maybe d_0 = d_1 = 10 is a better choice for the base setting.

**Limitations:**

I could not find where the authors addressed the limitations of this work.

---

> ### Author Rebuttal · Authors · 2023-08-09
>
> ### Q1:
> I think the overall writing can be improved. I did not feel excited when reading the paper up to and including page 5. This is just my feeling and it does not mean the paper is not good. Moreover, I find myself sometimes lost focus when reading the first 5 pages. Maybe adding 1-2 sentences describing the emphasis of each section at the beginning of the section can be helpful.
>
> ### R1:
> Thanks for your helpful suggestions. We will modify our paper to make it more reader-friendly.
>
> ### Q2:
> It seems to me that Section 3 and Section 4 study completely different things and under different settings. Although they are related. I think the results of Section 3 and Section 4 can be considered as two separate contributions of the paper. It might be a good idea to make this clear from the beginning. Currently, this gap between the two sections is not clear from neither the abstract nor the introduction.
>
> ### R2:
> Thanks for your suggestion. This is a very important and implementable feedback. We will elaborate the relation between the contributions of Section 3 and Section 4 in abstract and introduction.
>
> Both Section 3 and 4 are motivated by the same principle that node distinguishability is related to "intra-class 'distance v.s. inter- class 'distance'". Section 3 directly studies this principle with CSBM-H, which is a toy example. Section 4 verifies whether this principle really relates to the performance of GNNs and derives new performance metrics based on it.
>
> ### Q3:
> There are a few typos in the paper.
>
> ### R3:
> Thanks for carefully going through our paper and point out those typos. We have corrected them in the revised version.
>
> ### Q4:
> Line 112, the authors claim that "the current homophily metrics are all ... feature independent", however, the generalized edge homophily is feature dependent.
>
> ### R4:
> Thanks for your suggestion. We will modify it to "...almost all homophily metrics..."  in the revised version.
>
>
> ### Q5:
> For Figure 2, since d_0 = d_1 = 5, how did you obtain the plot at h=0.5? It's fine to apply some smooth interpolation but the middle point is important since it is the peak for LP. Maybe d_0 = d_1 = 10 is a better choice for the base setting.
>
> ### R5:
> Thanks for your suggestion. We relax $hd_0$ and $hd_1$ to be continuous values so that the curves in figures in Section 3.4 are more readable and intuitive, especially to show the intersections of the 3 zones. We will add comments to clarify this point for CSBM-H to avoid unnecessary confusion. We will try $d_0 = d_1 = 10$ setting as you suggest in the revised version.

---

> > ### Comment · Reviewer_NHN6 · 2023-08-15
> >
> > Thank you for the responses. My questions have been addressed. Overall I think this is a good paper.

---

> > > ### Author Response · Authors · 2023-08-15
> > > **Thank you**
> > >
> > > We appreciate your recognition and strong support to our paper. Your constructive suggestions make our paper better.

---

### Official Review · Reviewer_GCuM · 2023-07-06

**Soundness:** 3 good
**Presentation:** 4 excellent
**Contribution:** 2 fair
**Rating:** 3
**Confidence:** 2

**Summary:**

The authors analyze the node distinguishability in attributed graphs under the prism of homophily. They consider the binary CSBM model. They compute the classification error rate on the original features and of low- and high-frequency filters; they study the influence of inter- and intra-class variance on their performances. The authors propose new metrics based on these variances and on the comparison between GNN and NN.


**Strengths:**

I am not very familiar with the literature needed to assess the originality of this article.

It is well written.

**Weaknesses:**

For me it seems the article mainly studies the separability of a binary Gaussian mixture. The principle it derives (l. 347, whether intra-class node embedding "distance" is smaller than inter-class node embedding "distance") just means the two Gaussians do not overlap.

**Questions:**

A few remarks:

L. 75 are the features in R or R^F? and l. 154, in F_h?

L. 157 what is FP? In eq. 3 the authors could remind what are the FP and LP filters among the many quantities.

Theorem 1: I would not call it the optimal Bayes classifier since it is restricted to the features only; it does not take in account the graph of the CSBM. Also, this is just the optimal classifier for a binary Gaussian mixture; maybe it is not worth a theorem.


**Limitations:**

The authors did not discuss possible limitations.

---

> ### Author Rebuttal · Authors · 2023-08-09
>
> ### Q1:
> For me it seems the article mainly studies the separability of a binary Gaussian mixture. The principle it derives (l. 347, whether intra-class node embedding "distance" is smaller than inter-class node embedding "distance") just means the two Gaussians do not overlap.
>
> ### R1:
> Thanks for carefully going through our paper, but we think you might oversimplify our results and would like to clarify the contributions of our paper. The main goal of Section 3 is not "studies the separability of a binary Gaussian mixture". Instead, we are actually interested in how the separability (node distinguishability) changes as homophily level $h$ changes from 0 to 1 and how to use a curve to show the whole picture. We are also interested in how the curves for different graph filters intersect with each other. Researchers are intrigued by these topics, but these problems remain underexplored and not well-understood in the current GNN community.
> Also, the simplified two-normal setting is widely used to study various tasks on graphs, including classification on heterophilic graphs [2], as mentioned at the beginning in section 3.2.
>
> In addition, rather than derived from the two-normal setting, the principle is motivated from the example in Figure 1 as stated in Section 3.1. We use CSBM-H to formulate the principle as stated in line 134.
>
> Furthermore, the two-normal setting is a toy example to study the principle intuitively and  the principle does not just mean "two Gaussians do not overlap". In fact, its importance lays in that it provides a new way to develop metrics beyond homophily values, e.g. classifier-based performance metrics, to quantify node distinguishability (ND). Those metrics are verified to be better than homophily values, however, this principle has never been studied for understanding the effect of homophily. Our paper fills the gap and we believe it can provide new tools for researchers to explore heterophily in the future. Thus, we hope you can re-evaluate the contribution and novelty of our paper.
>
> ### Q2:
> L. 75 are the features in R or R^F? and l. 154, in F_h?
>
> ### R2:
> We use $R^F$ because section 2 just gives a  general introduction of graph. But from your feedback, we found that this might cause some unnecessary confusion. Thus, we decide to use $R^{F_h}$ in the revised version.
>
> ### Q3:
> L. 157 what is FP? In eq. 3 the authors could remind what are the FP and LP filters among the many quantities.
>
> ### R3:
> FP is for Full-pass. Thanks for the suggestion and we will add explaination to FP, LP and HP before equation 3 in the revised version.
>
> ### Q4:
> Theorem 1: I would not call it the optimal Bayes classifier since it is restricted to the features only; it does not take in account the graph of the CSBM. Also, this is just the optimal classifier for a binary Gaussian mixture; maybe it is not worth a theorem.
>
> ### R4:
> Thanks for the suggestion. As mentioned in line 163-165, the theorem is about $x$ (feature-only), but the results are applicable to $h$ and $h^{HP}$ (i.e. the graph information of the CSBM-H is included in $h$ and $h^{HP}$) when the parameters are replaced according to Equation 3. Thus, the optimality will be kept for graphs with different homophily levels and with different filters. We will emphasize this sentence in case other readers miss it. About the naming problem, please refer to [1].
>
> The result from theorem 1 is important to quantify the node distinguishability of CSBM-H and is firstly proposed to study homophily. When people try to study more complicated variants of CSBM-H in the future, theorem 1 is also a crucial tool. Thus, we think it's worth a theorem.
>
>
> [1] https://en.wikipedia.org/wiki/Bayes_classifier
>
> [2] Is Homophily a Necessity for Graph Neural Networks?. In International Conference on Learning Representations.

---

> > ### Comment · Reviewer_GCuM · 2023-08-18
> >
> > I thank the authors for their detailed answers. I am still skeptical about the contribution this article brings. The presentation is easy to follow and the setting allows to derive clear and understandable conclusions; but they seem limited and too simple. Figs. 2 to 5 depict the separability of two Gaussians that are more or less mixed by different filters that depend on $h$.
> >
> > > its importance lays in that it provides a new way to develop metrics beyond homophily values, e.g. classifier-based performance metrics, to quantify node distinguishability (ND).
> >
> > If I am wright, homophily was introduced to explain why GNNs perform badly on some datasets. I may simplify too much (again, I am not very familiar with this topic): it seems that this new classifier-based metric trivially means that GNNs have difficulties where (simple) GNNs are bad.

---

> > > ### Author Response · Authors · 2023-08-18
> > > **Response to Reviewer GCuM**
> > >
> > > Thanks for your reply and here is our response to your concerns of novelty.
> > >
> > > ### Q1.
> > > I thank the authors for their detailed answers. I am still skeptical about the contribution this article brings. The presentation is easy to follow and the setting allows to derive clear and understandable conclusions; but they seem limited and too simple. Figs. 2 to 5 depict the separability of two Gaussians that are more or less mixed by different filters that depend on $h$.
> > >
> > > ### R1:
> > > As we mentioned in our reply, the two-Gaissian setting is a commonly used tool and toy example to study the complex phenomenon of homophily and heterophily, which is widely used in heterophily community. We never try to oversimplify the setting to get easier results or take any advantage of it compared with other published literature. We just try to be consistent with the tool that are developed in heterophily community.
> > >
> > > In addition, just like any other machine learning method whose direct computation is intractable or infeasible, the simplification is sometimes necessary and should not decrease its importance. The merit of two-normal setting is that its node distinguishability (ND) can be explicitly quantified. And through the ND, we discovered the 3 zones, how the 3 zones and 3 curves change as homophily level changes and how class variances and node degree impact the 3 zones. These discoveries are all important, innovative, and of high interest in heterophily community, and their importance cannot be impaired by its simplicity.
> > >
> > > Besides, in our paper, we do not limit our analysis in two-normal setting. In section 3.5, we extend the analysis to more general settings and find some similar conclusions. But unfortunately, the ND or seperability of this general setting cannot be explicitly quantified (if this is what you expect).
> > >
> > > Discussing the variants of CSBM-H with more complicated assumptions might be doable, e.g. with different class homophily, different node local homophily distributions, different node degree and class variance distributions and the undirected message passing, etc.. We will add discussion in the revised version and encourage the GNN community to think about the more complicated setting.
> > >
> > > Based on the above points, we don't think the contribution of our paper can be denied because of simplicity.
> > >
> > > ### Q2.
> > > If I am wright, homophily was introduced to explain why GNNs perform badly on some datasets. I may simplify too much (again, I am not very familiar with this topic): it seems that this new classifier-based metric trivially means that GNNs have difficulties where (simple) GNNs are bad.
> > >
> > > ### R2:
> > > Homophily was not introduced to explain "why GNNs perform badly", it talks about why GNNs are worse (compared to NNs). To clarify it, let me first introduce the history of this line of research briefly.
> > >
> > > As we stated in our paper, graph-aware models (GNNs) differ from graph-agnostic models (NNs) because it has an additional feature aggregation step in each layer. This extra aggregation step sometimes brings benefit which gives us better performance, but sometimes brings harm and damage which gives us worse performance. People want to use GNNs (with feature aggregation) on those "good" graphs and avoid GNNs on "bad" graphs and we need an easy method to categorize them.
> > >
> > > In 2020, people started looking for the reason of the harm. At that time, they believed heterophily is the answer and tried to use (edge or node) homophily values to categorize the "good" and "bad" graphs [1]. In 2021, people questioned the above conclusion and found that homophily is not a necessity for better performance and heterophily is not always worse [2]. In 2022, people tried to find out better homophily metrics to differentiate "good" and "bad" graphs [3], but that metric is linear, feature-independent and cannot provide an accurate threshold value with statistical meaning for the categorization. The classifier-based metric proposed in our paper tried to address the above problem and is verified to give more accurate threshold values with statistical significance. Finding when GNNs has advantage and disadvantage against NNs is far from trivial.
> > >
> > > I hope this explanation can clarify the importance of the proposed metric and help you better understand this line of research. We will add this explanation to the revised version if you find it necessarry.
> > >
> > > [1] Beyond homophily in graph neural networks: Current limitations and effective designs. Advances in neural information processing systems, 33, 7793-7804.
> > >
> > > [2] Is Homophily a Necessity for Graph Neural Networks? International Conference on Learning Representations. 2021.
> > >
> > > [3] Revisiting heterophily for graph neural networks. Advances in neural information processing systems, 35, 1362-1375.

---

> > > > ### Author Response · Authors · 2023-08-20
> > > >
> > > > Hi Reviewer GCuM,
> > > >
> > > > Thanks for your feedback on our paper. Since the deadline is approaching and we are not allowed to post anything after that, we could have a final round discussion before that if you think it's needed. If you find everything is ok, we hope you could reevaluate the novelty of our paper and raise your score. We will appreciate that.
> > > >
> > > > Authors

---

### Official Review · Reviewer_FryU · 2023-07-25

**Soundness:** 4 excellent
**Presentation:** 4 excellent
**Contribution:** 4 excellent
**Rating:** 8
**Confidence:** 1

**Summary:**

Recent research indicates that Graph Neural Networks (GNNs) maintain their advantage even in the absence of homophily, as long as nodes from the same class exhibit similar neighborhood patterns. This argument, however, primarily considers intra-class Node Distinguishability (ND) while overlooking inter-class ND, thus providing an incomplete understanding of homophily. Therefore, the authors in this paper propose that an ideal ND scenario entails smaller intra-class ND relative to inter-class ND. To substantiate this, the authors introduce the Contextual Stochastic Block Model for Homophily (CSBM-H) and define two ND metrics: Probabilistic Bayes Error (PBE) and negative generalized Jeffreys divergence. Experimental results reveal that the supremacy of GNNs is indeed closely tied to both intra- and inter-class ND, irrespective of homophily levels.


**Strengths:**


1. The paper addresses a highly significant issue.
2. The proposed CSBM-H and the defined metrics illustrate that the superiority of GNNs is indeed closely linked with both intra- and inter-class ND, regardless of homophily levels. The results are interesting.
3. The paper is solidly grounded in its field.


**Weaknesses:**


N/A

**Questions:**


N/A

**Limitations:**

While I do not consider myself an expert in this area, I appreciate the content of this paper. It addresses a highly significant issue, and proposes that an ideal ND scenario would have smaller intra-class ND compared to inter-class ND. To formulate this idea, the authors introduce the Contextual Stochastic Block Model for Homophily (CSBM-H) and define two ND metrics: Probabilistic Bayes Error (PBE) and negative generalized Jeffreys divergence. The proof is robust and the paper appears to be strongly grounded in the subject matter.

---

> ### Author Rebuttal · Authors · 2023-08-09
>
> Thanks so much for your nice review and strong recognition to our contributions.

---

### Official Review · Reviewer_RYrF · 2023-07-27

**Soundness:** 3 good
**Presentation:** 3 good
**Contribution:** 3 good
**Rating:** 7
**Confidence:** 3

**Summary:**

The paper delves deeper into the topic of GNNs' superiority in node classification tasks. The authors conduct experiments to show that GNNs do better due to inter and intra class node distinguishability regardless of homophily levels as suggest by current research. They propose a new metric that sheds more light on how GNNs work.

**Strengths:**

GNNs are a hot topic and studying how they work and what settings are conducive for GNNs to be more effective is important. This paper does that and proposes a new metric to that end.

**Weaknesses:**

1. Since the crux of the paper is about node distinguishability (ND), it would be nice to see inter and intra ND visualizations of the datasets used similar to Figure 1 (toy example).

2. I'm not sure the claim "superiority of GNNs is indeed closely related to both intra- and inter-class ND regardless of homophily levels" is sufficiently backed by the evidence in Table 1. For example, setting a different threshold than 0.5 for homophily metrics could make them look better.

**Questions:**

1. Anything special about the PubMed dataset causing incorrect guesses? Also, Chameleon and Squirrel datasets have more incorrect guesses than other "less homophilic" datasets. Any way to visualize ND to pin point where the difference is coming from?

2. Any comment on the time complexity trade-offs between homophily metrics and classifier based ones?

Minor:
line 18: "it is significantly" instead of "it significantly"
line 321: "split" instead of "splits"

---

> ### Author Rebuttal · Authors · 2023-08-09
>
> ### Q1:
> Since the crux of the paper is about node distinguishability (ND), it would be nice to see inter and intra ND visualizations of the datasets used similar to Figure 1 (toy example).
>
> ### R1:
> Thanks for your suggestion. Actually, we do include the option to visualize CSBM-H at different homophily levels in our code in thesupplementary material. We will add some visualization in the revised version.
>
> ### Q2:
> I'm not sure the claim "superiority of GNNs is indeed closely related to both intra- and inter-class ND regardless of homophily levels" is sufficiently backed by the evidence in Table 1. For example, setting a different threshold than 0.5 for homophily metrics could make them look better.
>
> ### R2:
> This is a good question. For the threshold problem, we would like to emphasize the following points: 1. 0.5 is a commenly used threshold value  in homophily/heterophily community to seperate homophilic and heterophilic graphs. It is not a randomly picked value and it has mathematical meaning: homophily, i.e. metric greater than 0.5, means the proportion of edges that connect nodes from the same class is larger than that from different classes in some way, and vice versa. Other threshold should have statistical or mathematical meaning and we cannot arbitrarily choose one. This property is also one of the advantages of our proposed metrics over the existing metrics. 2. We cannot manually pick different threshold values for different homophily metrics to make them "look better" on certain set of graphs. This will cause "overfitting" problem, i.e. when you use this cherry-picked threshold on other unseen datasets, it might perform badly.
>
> Thus, our claim is properly backed by the results from Tables 1. Besides, we provide additional experimental evidence in Appendix G to support our claim.
>
> ### Q3:
> Anything special about the PubMed dataset causing incorrect guesses? Also, Chameleon and Squirrel datasets have more incorrect guesses than other "less homophilic" datasets. Any way to visualize ND to pin point where the difference is coming from?
>
> ### R3:
> Thanks for going through Table 1 carefully and this is a very good question. From the experimental results on large-scale datasets reported in Table 4 in Appendix G.4, we observe that, for linear and non-linear G-aware models, there exists inconsistency between their comparison with their coupled G-agnostics models. For example, for $\textit{Penn94, pokec, snap-patents}$ and $\textit{twitch-gamers}$, SGC-1 underperforms MLP-1 but GCN outperforms MLP-2. In fact, $\textit{PubMed}$ also belongs to this family of datasets. We do not have a proved theory to explain this phenomenon for now. But there is obviously a synergy between homophily/heterophily and non-linearity that cause this discrepancy together. And we think on this spectial subset of heterophilic graphs, we shoud develop theoretical analysis to discuss the interplay between graph structure and feature non-linearity, and how they affect node distinguishability together.
> The current homophily values (including the proposed metrics) are not able to explain the phenomenon associated with this group of datasets.
> We keep it as an open question and encourage people from the GNN community to study it in the future. We will add the above discussion to the revised version of this paper.
>
>
> The visualization of ND for Chameleon and Squirrel is a good point. We will try to add this to the revised version.
>
> ### Q4:
> Any comment on the time complexity trade-offs between homophily metrics and classifier based ones?
>
> ### R4:
> In Table 9 in Appendix G.5 (in supplementary material), we provide the running time of CPMs on small- and large-scale datasets. The running time of  CPM is short - it takes several minutes with 1 NVIDIA V100 even on large-scale datasets such as pokec and snap-patents, which contain  millons of nodes and tens of millions of edges. In comparison, training GCN on these datasets will take several hours. It takes even days for SOTA models, e.g. ACM-GCNs, to train on them.

---

> > ### Comment · Reviewer_RYrF · 2023-08-13
> > **Thank you for the response**
> >
> > Thank you for the thorough response. I changed the score to 7

---

> > > ### Author Response · Authors · 2023-08-13
> > > **Thanks you**
> > >
> > > Thanks so much for your timely response and positive feedback.

---

### Author Rebuttal · Authors · 2023-08-09

In this part, we will provide experimental results of Label Informativeness (LI) and adjusted homophily ($H_\text{adj}$) on small- and large-scale datasets and compare them with the proposed metrics. Before discussing the results, we would like to introduce 2 threshold values for classifier-based performance metrics (you can find it in Appendix G.2)

Typically, the threshold for homophily and heterophily graphs is set at 0.5. For classifier-based performance metrics, we establish two benchmark thresholds as below,
 • Normal Threshold 0.5 (NT0.5): Although not indicating statistical significance, we are still comfortable to set 0.5 as a loose threshold. A value exceeding 0.5 suggests that the G-aware model is not very likely to underperform their coupled G-agnostic model on the tested graph and vice versa. (Error cases with regard to NT0.5 are marked by grey)
 • Statistical Significant Threshold 0.05 (SST0.05): Instead of offering an ambiguous statistical interpretation, SST0.05 will provide a clear statistical meaning. A value smaller than 0.05 implies that the G-aware model significantly underperforms their coupled G-agnostic model and a value greater than 0.95 suggests a high likelihood of G-aware model outperforming their coupled G-agnostic model. Besides, a value ranging from 0.05 to 0.95 indicates no significant performance distinction between G-aware model and its G-agnostic model. SST0.05 is a more strict threshold than NT0.5. Therefore, we will observe more error cases under SST0.05 than NT0.5. (Error cases with regard to SST0.05 are marked by red)

We can see that, no matter on small- or large-scale datasets, the classifier-based performance metrics (CPMs) are significantly better than the existing metrics on revealing the advantages and disadvantages of GNNs, decreasing the overall error rate from at least 0.34 to 0.13 (at most 0.19 under SST0.05).

---

### Decision · Program_Chairs · 2023-09-21

**Decision:**

Accept (poster)

**Comment:**

The paper studies the question of when graph-aware models outperform the corresponding graph-agnostic (non-GNN) models for node classification (motivated by past work observing that graph-agnostic methods outperform GNNs on certain heterophilic graphs, and a follow-up work discussing that GNNs can still perform well under heterophily).

Their main insight is that while existing work mainly considers intra-class Node Distinguishability (ND), inter-class ND also has to be jointly considered. To support this idea, they conduct theoretical analysis using the two-class CSBM-H model, a variant of contextual stochastic blockmodels parametrized by the homophily level. They then define metrics (PBE and Jeffreys divergence) to quantify node distinguishability, showing how intra- and inter-class ND relate. They also conduct an empirical study on ND, finding that intra- vs inter-class node distance is well correlated with whether the graph-aware model outperforms graph-agnostic models.

Reviewer scores are borderline to positive, with 1 negative review. The main strengths raised include the significance of the issue, and insights about intra-class and inter-class ND with good theoretical and empirical support.

The main weaknesses raised by the review (with score of 3) include concerns about the significance of the results based on the two-Gaussian setting and whether they are too limited / simple, or its relation to separability of Gaussians. The authors responded, e.g. mentioning the effect of graph filters, that the heterophily issue is significant and not well-understood in the GNN community, and how the two-Gaussian setting leads to their insights of the "3 zones" and new metrics. In my view, the paper's analysis indeed captures the effect of graph structure under different homophily levels (using the CSBM model), high/low-pass filters, and studies them using metrics proposed in the paper, providing some insights that are (in my view) not well known in the graph learning community.

Also, the author response helped to satisfactorily address some of the other raised issues, e.g. adding new baselines (Label Informativeness and adjusted homophily). Overall, the reviewers and AC overall find that the paper addresses an important issue, provide novel and interesting insights supported by convincing theoretical and empirical analysis. Hence, I recommend acceptance.

Authors are requested to note the suggested improvements / action items arising from discussion with the reviewers, such as adding the new empirical results and plots to the final manuscript, and other improvements to the paper (e.g. writing issues) arising from reviewer discussion.